# Navigating the complexity of detrital rutile provenance: Methodological insights from the Neotethys Orogen in Anatolia

Megan A. Mueller[1,2,*], Alexis Licht[1,3], Andreas Möller[4], Cailey B. Condit[1], Julie C. Fosdick[2], Faruk Ocakoğlu[5], Clay Campbell[6]

[1.] Department of Earth and Space Sciences, University of Washington, 4000 15th Avenue NE, Seattle, WA 98195, USA

[2.] Department of Earth Sciences, University of Connecticut, 354 Mansfield Road - Unit 1045, Storrs, CT 06269, USA

[3.] Aix-Marseille Université, CNRS, IRD, INRAE, Collège de France, CEREGE, Technopôle de l'Arbois-Méditerranée, BP80, 13545 Aix-en-Provence, France

[4.] Department of Geology, The University of Kansas, 1414 Naismith Drive, Lawrence, KS 66045, USA

[5.] Department of Geological Engineering, Eskişehir Osmangazi University, Büyükdere, 26040 Eskişehir, Türkiye

[6.] Department of Geosciences, University of Arizona, 1040 E 4th St, Tucson, AZ 85721, USA

[*] Now at Department of Earth and Planetary Sciences, Jackson School of Geosciences, The University of Texas at Austin, 2305 Speedway Stop C1160, Austin, TX 78712, USA

*Correspondence to*: Megan Mueller (megan.mueller@jsg.utexas.edu)

**Abstract.** Sedimentary provenance is a powerful tool for reconstructing convergent margin evolution. Yet single mineral approaches, like detrital zircon, have struggled to track sediment input from mafic and metamorphic sources. Detrital rutile complements detrital zircon datasets by offering a path forward in sedimentary provenance reconstructions where metamorphic terranes are potential source regions. However, U-Pb geochronology in rutile can be difficult due to low uranium concentrations and incorporation of common Pb, and multiple workflows are currently in use. Here, investigate U-Pb and trace element data reduction, processing, and common Pb correction workflows using new detrital rutile U-Pb geochronology and trace element geochemistry results from the Late Cretaceous to Eocene Central Sakarya and Sarıcakaya Basins in Anatolia. A significant number of analyses were rejected (54%) due to signal intensity limitations, namely low U, low Pb, anomalous signal, and inclusions. We identify this as a universal limitation of large-*n* detrital rutile studies and recommend the systematic reporting of the amount of discarded analysis and the processes for rejection in all studies using detrital rutile U-Pb geochronology. Additionally, we show that (1) the $^{208}Pb$ and $^{207}Pb$ common Pb reduction schemes produce similar age distributions and can be used indifferently; (2) The Stacey-Kramers distance is a suitable metric for quantifying U-Pb discordance but a discordance filter is not recommended; (3) Instead, filtering U-Pb data by a power law function based on corrected date uncertainty is appropriate; (4) the exclusion of low uranium concentration rutile biases date distributions and favors pelitic-derived, higher Zr-in-rutile temperature, higher U-Pb concordance grains; (5) paired U-Pb and trace elements

can be used to evaluate potential bias in U-Pb data rejection, which reveals that data rejection does not bias the provenance interpretations; (6) the signature of sediment recycling can be identified through U-Pb dates and Zr-in-rutile temperatures. To better navigate the complexity of detrital rutile datasets and to facilitate the standardization of data reporting approaches, we provide open access code as Jupyter Notebooks for data processing and analysis steps, including common Pb corrections, uncertainty filters, discordance calculations, and trace element analysis.

## 1 Introduction

Sedimentary provenance analysis is widely used to reconstruct ancient sediment dispersal networks, source-to-sink sediment budgets, sedimentary basin evolution, and to discern links between tectonics, geodynamics, paleogeography, climate, and biologic evolution (Dickinson and Suczek, 1979; Garzanti et al., 2007; Clift et al., 2008; Gehrels, 2014; Blum and Pecha, 2014). Compositional provenance methods include sediment petrologic, chemical, and heavy mineral characterizations (e.g., Gazzi, 1965; Hubert, 1971; Dickinson and Suczek, 1979; Morton, 1985; Garzanti and Andò, 2007). Over the last several decades, the rise of chronometric and geochemical techniques led to the increase in single-mineral approaches. Detrital zircon U-Pb geochronology has become the most widely used technique as zircon is refractory and is abundant in crustal rocks (e.g., Gehrels, 2014). Further, the age, thermal history, and elemental and isotopic composition of detrital zircons can quantitatively reconstruct both sedimentary provenance and geodynamic, tectonic, and magmatic processes (Carrapa, 2010; Paterson and Ducea, 2015; Tang et al., 2020; Sundell et al., 2022). However, one major limitation is that zircons predominantly form in intermediate to felsic magmas, thus detrital zircon suites generally lack information about mafic igneous and metamorphic processes and sources (Hietpas et al., 2011; Moecher et al., 2011; Gaschnig, 2019). Zircon is present in metamorphic rocks as inclusions in other minerals or as recrystallized-dissolved-reprecipitated rims on zircon cores (Kohn and Kelly, 2017). The outer growth domains of zircons can be targeted with laser ablation ICP-MS depth profiling or with spot analysis if the rims are thick enough, yet the most commonly used techniques for rapid provenance data acquisition do not routinely analyze zircon rims. Therefore, sedimentary provenance interpretations based on detrital zircon alone are incomplete. For this reason the sedimentary provenance community is increasingly turning to U-Th-Pb and trace elements in phases commonly used in petrochronology, such as detrital rutile (Zack et al., 2004a; Meinhold, 2010; Triebold et al., 2012; Bracciali et al., 2013; Rösel et al., 2014, 2019; O'Sullivan et al., 2016; Odlum et al., 2019; Pereira et al., 2020), detrital apatite (Morton and Yaxley, 2007; Chew et al., 2011; Mark et al., 2016; O'Sullivan et al., 2016, 2020), detrital monazite (Hietpas et al., 2010; Moecher et al., 2011; Gaschnig, 2019), and detrital titanite (Guo et al., 2020; Chew et al., 2020), in addition to other isotopic systems in these and other detrital minerals.

Detrital rutile is a complementary sedimentary provenance proxy to detrital zircon. Rutile forms in metamafic and metapelitic rocks across a range of P-T conditions, therefore, detrital rutile is especially advantageous when tracking sediment input from greenschist to eclogite or granulite facies sources (e.g., Meinhold, 2010; Zack and Kooijman, 2017). The geochemical composition can further distinguish between metamorphic protoliths (e.g., Triebold et al., 2007, 2012; Meinhold,

2010). However, rutile U-Pb analysis is challenging due to low U and low radiogenic Pb concentrations and due to the incorporation of initial non-radiogenic Pb. Here, we use a new detrital rutile petrochronology dataset from Anatolia to investigate data reduction, processing and analytical steps in order to support robust provenance interpretations. In a number of studies, analyses have been discarded during U-Pb data reduction due to unacceptable signal intensity (e.g. Bracciali et al., 2013; Rösel et al., 2014, 2019), and we find that discarding analyses is a limitation to large-*n* detrital rutile datasets in the literature and this study. We test the sensitivity of resulting U-Pb date spectra to Pb correction methods, uncertainty and discordance filters, and a low U cutoff threshold. Ultimately, the new dataset demonstrates that detrital rutile captures sediment input from a subduction accretion complex that is poorly resolved in the detrital zircon record. Despite the described limitations, detrital rutile petrochronology can be effectively used to reconstruct sedimentary provenance and sediment recycling, deformation, and metamorphism.

## 2 Detrital Rutile Provenance

### 2.1 Detrital Rutile Synopsis

The advantages of detrital rutile provenance are extensively documented (e.g., Zack et al., 2004a; Meinhold, 2010; Triebold et al., 2012; Bracciali, 2019; Gaschnig, 2019; Pereira et al., 2020; Pereira and Storey, 2023) so we provide only a brief overview here. Rutile is the most common $TiO_2$ polymorph, a common accessory mineral in metamorphic and igneous rocks (Meinhold, 2010; Zack and Kooijman, 2017), and an abundant heavy mineral in sedimentary rocks (Morton, 1985). Rutile is present across a range of P–T conditions: rutile is generally stable at the surface and medium- to high-grade metamorphic conditions. Rutile can readily crystallize from titanite, ilmenite and biotite during prograde metamorphism (Luvizotto et al., 2009; Meinhold, 2010; Cave et al., 2015). The breakdown of rutile to titanite occurs in prograde and retrograde environments, particularly in sub-greenschist to lower greenschist facies where titanite stability is favored (Cave et al., 2015; Zack et al., 2004b). Experimentally, rutile is stable above around 1.2–1.4 GPa in metagranitoids and hydrated basalts depending on compositional and chemical variability and in some cases can be stable down to 0.7 GPa (Xiong et al., 2005; Angiboust and Harlov, 2017). In subduction zone settings, rutile is especially abundant in eclogites (Klemme et al., 2002).

The chemical composition of rutile preserves original petrogenetic information. Rutile concentrates high field strength elements (Zr, Nb, Mo, Sn, Sb, Hf, Ta, W) through substitution with Ti that are commonly used as fingerprints of subduction zone metamorphism and crustal evolution (Foley et al., 2000; Rudnick et al., 2000). Detrital rutile geochemistry fingerprints the lithologies of sediment sources in several unique ways: rutile concentrates the vast majority of available Nb whereas Cr is non-selective and is distributed across metamorphic minerals; therefore, the Cr and Nb concentrations in rutile can discriminate between metamafic and metapelitic lithologies (Zack et al., 2004a, b; Triebold et al., 2011, 2012). Cr and Nb concentrations are attributed to different protoliths: metapelitic rutile (i.e. mica schists, paragneisses, felsic granulites) have Cr < Nb and metabasic rutile (i.e., mafic eclogites and granulites) have Cr > Nb, generally (Zack et al., 2004b)Additionally, the incorporation of Zr in rutile is largely temperature dependent (Zack et al., 2004b; Watson et al., 2006; Tomkins et al., 2007;

Ferry and Watson, 2007). Zirconium mobilizes during prograde metamorphic fluid release; the incorporation of Zr into rutile
is buffered by coexisting quartz and zircon (Zack et al., 2004b). Zr contents in rutile correlate with peak metamorphic
temperature and pressure conditions (Zack et al., 2004b; Watson et al., 2006; Tomkins et al., 2007; Kohn, 2020). Therefore,
the Zr elemental composition in rutile is a commonly used thermometer, empirically and experimentally calibrated across a
range of pressures and thermodynamic activity parameters (Zack et al., 2004b; Watson et al., 2006; Tomkins et al., 2007;
Kohn, 2020). Zircon, quartz and rutile must be equilibrium to use the Zr-in-rutile thermometer (e.g., Zack et al., 2004b), an
assumption that likely holds in pelitic rocks (Pereira et al., 2021) but may not in mafic lithologies, yet the assumption is hard
to evaluate in a detrital context. Inclusions in rutile can be used to determine whether rutile grew in equilibrium (Hart et al.,
2016, 2018; see also Pereira and Storey, 2023 and references therein). In detrital rutile, removed from the petrologic system in
which they formed, and thereby miss key thermobarometric mineral associations, the Zr-in-rutile thermometer thus provides
an estimate of the minimum peak metamorphic temperatures because the exact activity of $SiO_2$ in the original system is
unconstrained (Kooijman et al., 2012; Triebold et al., 2012; Pereira et al., 2021; see also Meinhold et al., 2008; Rösel et al.,
2019; Şengün et al., 2020; Zoleikhaei et al., 2021). For rutile of unknown source lithology, the calculated temperature is
affected by the chosen pressure estimate; Pereira and Storey (2023) demonstrate this pressure dependence in detrital grains
and recommend using the experimental and empirical calibration of Kohn (2020; their eqn. 13) at an average pressure of 13
kbar with an uncertainty of 5 kbar:
$$T \ (\degree C) = \frac{71360 + 0.378 \times P - 0.130 \times C}{130.66 - R \times \ln[C]} - 273.15$$
(*1*)
where P is the pressure in bars, C is the concentration of Zr in ppm and R is the gas constant, 8.3144 in $J \cdot mol^{-1} \cdot K^{-1}$.
Uranium is easily substituted for $Ti^{4+}$ in rutile at concentrations up to ~100 ppm U making rutile a suitable mineral
for U-Pb analysis. Rutile U-Pb analyses were first performed using thermal ionization mass spectrometry (TIMS) (Schärer et
al., 1986; Mezger et al., 1989; Möller et al., 2000; Schmitz and Bowring, 2003; Kylander-Clark et al., 2008) and have since
been collected with SHRIMP (Clark et al., 2000; Meinhold et al., 2010; Ewing et al., 2015), LA-MC-ICP-MS (Vry and Baker,
2006; Bracciali et al., 2013; Apen et al., 2020), LA-Q-ICP-MS (Storey et al., 2007; Zack et al., 2011), and LA-SC-ICP-MS
(Kooijman et al., 2010; Okay et al., 2011; Smye and Stockli, 2014). As a high-temperature thermochronometer, U-Pb dates in
rutile likely reflect mineral cooling through the closure temperature for volume diffusion of Pb (Dodson, 1973), which is
between 400–640°C in rutile. The temperature sensitivity of this partial retention zone in rutile is dependent on diffusion
kinetics, cooling rate, chemistry, and grain size (Mezger et al., 1989; Cherniak, 2000). Rutile U-Pb dates may correspond to
monotonic cooling from post-magmatic temperatures or cooling from the most recent medium to high-temperature
metamorphic event that exceeded the closure temperature (Zack et al., 2004b; Zack and Kooijman, 2017). Slow cooling rates
can produce rutile U-Pb dates significantly younger than the timing of peak metamorphism (e.g., Möller et al., 2000; Flowers
et al., 2005). Because rutile U-Pb dates record thermal history information from conditions characteristic of the middle to
lower crust (> 400 °C), U-Pb dates are ideal for inferring the timing and rate of deep seated orogenic processes (Mezger et al.,
1989; Möller et al., 2000; Flowers et al., 2005; Kylander-Clark et al., 2008; Smye et al., 2018) and of craton formation,
stabilization and cooling (Davis et al., 2003; Schmitz and Bowring, 2003; Blackburn et al., 2012). Furthermore, detrital rutile
U-Pb geochronology is regularly used in sedimentary provenance analysis to reconstruct sedimentary basin evolution,
paleoclimate and paleoenvironments, and orogen-scale deformation, exhumation, and sediment transport (Rösel et al., 2014,
2019; Mark et al., 2016; O'Sullivan et al., 2016; Pereira et al., 2020; Caracciolo et al., 2021; Clift et al., 2022).

## 2.2 Detrital Rutile U-Pb Challenge #1: Low Uranium Content

Detrital rutile U-Pb petrochronology presents unique analytical, data reduction, and interpretation challenges.
Uranium concentration in rutile varies among metamorphic protoliths: for example, rutile from mafic eclogites tend to have,
on average, 75% less U than those from metapelites (i.e., 5 ppm vs. 21 ppm; Meinhold, 2010). The low U concentrations—
from old rutile or sourced from mafic lithologies (cf. Section 6.2)—can make rutile challenging to date. To optimize data
collection, some detrital rutile methods first analyze trace elements then only collect U-Pb data on rutile above a given U
concentration threshold (ca. > 4–5 ppm; e.g., Zack et al., 2004a, 2011; Okay et al., 2011; Rösel et al., 2019). There is not a
systematic relationship between uranium concentration and common Pb concentration. However, screening low U rutile
reduces the overall length of U-Pb analytical sessions and produces a higher proportion of concordant analyses (Zack et al.,
2004a, 2011; Okay et al., 2011; Rösel et al., 2019). This protocol however introduces bias into the provenance results against
metamafic rocks (cf. Section 6.2), and is generally discouraged (Bracciali et al., 2013; Bracciali, 2019). While this low-U
screening is not necessarily common globally, it is a regional concern. There are 4 published detrital rutile U-Pb datasets from
Türkiye; 2 of the 4 (Okay et al., 2011; Şengün et al., 2020) only analyze U-Pb on detrital rutile with uranium concentrations
above ca. 4-5 ppm. The two studies that do not use a U-threshold filter but instead analyze all detrital rutile grains (Shaanan et
al., 2020; this study) must discard data due to very low uranium signals (below limit of detection; LOD). The U-threshold filter
is intended to maximize the proportion of concordant rutile analyzed. This includes rutile grains that have low incorporation
of U during growth (independent of analytical instrumentation) and rutile grains that have poorly resolved U-Pb ratios due to
low U CPS such as old rutile and mafic rutile (machine dependent). Omitting low U rutile may make sense in some settings;
however, this analytical approach likely biases provenance results as the concentration of uranium in rutile systematically
varies by metamorphic protoliths, with mafic eclogites having lower U contents than metapelites (e.g., Meinhold, 2010). This
potential bias is important to investigate as metamafic units in suture zones, presumably with low U rutile, are expected to be
a major contributor of detritus to many orogenic basins, including the northwestern Anatolian basins of this study.

## 2.3 Detrital Rutile U-Pb Challenge #2: Common Pb Incorporation

### 2.3.1 Common Pb Correction Overview

A second challenge with detrital rutile lies with data reduction and presentation. Because many detrital
geochronologists are familiar with the zircon system, here we emphasize the differences in how U-Pb data should be treated
in common Pb bearing minerals versus zircon. The U-Pb system in rutile is different from that of zircon due to the incorporation
of common Pb, thereby requiring careful methodological choices on how to treat non-radiogenic Pb and U-Pb discordance.
The zircon U-Pb system is 'simple' in the sense that zircon incorporates negligible non-radiogenic initial or 'common' Pb
during crystallization, and Pb diffuses only at extremely high temperatures and in zircon with radiation damage (e.g., Schoene,
2014 and references therein). Thus, the majority of detrital zircon U-Pb analyses tend to be close to concordia, which makes
data reduction and interpretation fairly straightforward, as even the $^{207}Pb/^{206}Pb$ dates of moderately discordant zircon are likely
to be meaningful. Unlike zircon where discordant data exceeding a specified threshold are often discarded, it is not surprising
that many rutile analyses may be discordant as rutile can incorporate a significant amount of common Pb. *In-situ* studies
mitigate this by: (1) regressing discordia lines through co-genetic analyses in Tera-Wasserburg space, where the lower intercept
of the discordia with the concordia defines the U-Pb age of Pb diffusion closure (e.g., Faure, 1986; Chew et al., 2011;
Vermeesch, 2020); or (2) applying a non-radiogenic Pb correction either by using an *ad hoc* Pb evolution model such as that
of Stacey and Kramers (1975) or by measuring the composition of non-radiogenic Pb in a co-existing phase (e.g. Zack et al.
2004b). However, by nature, co-genetic grains in detrital samples are unknown, and a model therefore has to be applied. Below
we review the common Pb correction calculations and discordance metrics for common Pb bearing detrital minerals.

**2.3.2 $^{204}Pb$ Correction**

The basis of all common Pb correction approaches—$^{204}Pb$, $^{207}Pb$ and $^{208}Pb$—is to use a Pb evolution model (e.g.,
Stacey and Kramers, 1975) to find the fraction of total $^{206}Pb$ that is common $^{206}Pb$ and, by corollary, find the radiogenic $^{206}Pb$
fraction and then calculate the corrected date (Compston et al., 1984; Williams, 1997). We did not measure $^{204}Pb$ in this study
and refer readers to other publications for $^{204}Pb$ correction details (Williams, 1997; Andersen, 2002; Storey et al., 2006; Chew
et al., 2014). The $^{204}Pb$ correction method is valuable because it uses the non-radiogenic $^{204}Pb$ isotope and does not assume
concordance, yet accurate measurement of $^{204}Pb$ is needed (in contrast, see Andersen, 2002) which can be challenging as $^{204}Pb$
is the least abundant Pb isotope. While accurate determination of the low-intensity $^{204}Pb$ peak is not a problem for TIMS or
MC-ICP-MS instruments (e.g., Simonetti et al., 2005; Gehrels et al., 2008), it can require prohibitively long dwell times in
single-collector instruments. Furthermore, the measurement of $^{204}Pb$ is complicated by the isobaric interference of $^{204}Hg$
introduced in the gas supply. In some cases, the concentration of $^{204}Hg$ can be reduced with traps or filters and back stripped
by measuring $^{201}Hg$ or $^{202}Hg$ (e.g., Storey et al., 2006).

**2.3.3 $^{208}Pb$ Correction**

The $^{208}Pb$ correction method determines the common Pb component using the $^{232}Th\text{-}^{208}Pb$ decay scheme and assumes
U-Th-Pb concordance, undisturbed Th/U, and no Pb loss. Because Pb loss is not considered, all corrected dates are (possibly)
minimum ages. The $^{208}Pb$ correction is ideal for low-Th phases (Zack et al., 2011) and is commonly used for rutile, although
not all rutile grains have low Th concentrations and Th contents are often not reported. The equations here are previously
described in Williams (1997), Chew et al. (2011), McLean et al. (2011) and as the total-Pb/U-Th scheme in Vermeesch (2020).

The proportion of $^{206}Pb_{common}$, $f_{206}$, is calculated as

$$f_{206} = \frac{\left(^{208}Pb/^{206}Pb_{measured}\right) - \left(^{208}Pb^*/^{206}Pb^*\right)}{\left(^{208}Pb/^{206}Pb_{common}\right) - \left(^{208}Pb^*/^{206}Pb^*\right)}$$

(2)

where $^{208}Pb/^{206}Pb_{measured}$ is calculated directly from the raw data. The $^{208}Pb/^{206}Pb_{common}$ ratio is calculated from the two-stage Pb evolution model of Stacey and Kramers (1975) for dates older than 3.7 Ga as

$$\left(\frac{^{206}Pb}{^{204}Pb}\right)_{common} = 7.19 \cdot \left(e^{\lambda_{238} \cdot 4.57 x 10^9} - e^{\lambda_{238} \cdot t_i}\right) + 9.307$$

(3)

and

$$\left(\frac{^{208}Pb}{^{204}Pb}\right)_{common} = 33.21 \cdot \left(e^{\lambda_{232} \cdot 4.57 x 10^9} - e^{\lambda_{232} \cdot t_i}\right) + 29.487$$

(4)

or for dates younger than 3.7 Ga as

$$\left(\frac{^{206}Pb}{^{204}Pb}\right)_{common} = 9.74 \cdot \left(e^{\lambda_{238} \cdot 3.7 x 10^9} - e^{\lambda_{238} \cdot t_i}\right) + 11.152$$

(5)

and

$$\left(\frac{^{208}Pb}{^{204}Pb}\right)_{common} = 36.84 \cdot \left(e^{\lambda_{232} \cdot 3.7 x 10^9} - e^{\lambda_{232} \cdot t_i}\right) + 31.23$$

(6)

where using $t_i$ is the uncorrected date in years, the $^{232}Th$ decay rate $\lambda_{232}$ is $4.9475 x 10^{-11}$ yr$^{-1}$, and the $^{238}U$ decay rate $\lambda_{238}$ is $1.55125 x 10^{-10}$ yr$^{-1}$ (Faure, 1986). The expected radiogenic $^{208}Pb^*/^{206}Pb^*$ ratios are calculated as

$$\frac{^{208}Pb^*}{^{206}Pb^*} = \left(\frac{^{232}Th}{^{238}U}\right) \cdot \left(\frac{e^{\lambda_{232} t_i} - 1}{e^{\lambda_{238} t_i} - 1}\right)$$

(7)

Then, the radiogenic component, the $^{206}Pb^*/^{238}U$ ratio, can be calculated as

$$^{206}Pb^*/^{238}U = (1 - f_{206}) \cdot \left(^{206}Pb/^{238}U_{measured}\right).$$

(8)

Finally, the [208]Pb-corrected date ([206]Pb*/[238]U date) is calculated by solving the age equation with the [206]Pb*/[238]U ratio:

$$t_{206} = \frac{1}{\lambda_{238}} \cdot \ln\left(\frac{^{206}Pb^*}{^{238}U} + 1\right)$$

216                          (9)

where $t_{206}$ is the corrected age in years. The final corrected date is calculated iteratively, whereby each iteration replaces $t_i$ with
the previously calculated [206]Pb*/[238]U date. The final [208]Pb-corrected date presented here is from the two hundredth iteration.
For our dataset, we varied the initial age estimate, and therefore the initial common Pb composition, from 1 Ma to 1000 Ma
and, by the fifth iteration, the resulting [208]Pb-corrected date differs by less than 0.05% for 98% of the unknowns. The
uncertainty on the date is calculated as the equivalent of the percent (propagated) uncertainty of the uncorrected [206]Pb/[238]U
ratio (Odlum et al., 2019).

### 2.3.4 [207]Pb Correction

The [207]Pb correction method is based on a linear regression of [207]Pb/[206]Pb and [238]U/[206]Pb in Tera-Wasserburg space
(Tera and Wasserburg, 1972) along a two-component mixing line between non-radiogenic and radiogenic Pb (Faure, 1986;
Figure 1). This method is most powerful for co-genetic minerals because it does not require knowing [207]Pb/[206]Pb_common. Yet,
because co-genetic analyses are inherently unknown in detrital samples, the routine used here calculates the common Pb
component of each individual analysis using the Stacey and Kramers (1975) two-stage Pb evolution model and an initial age
estimate. The [207]Pb correction method assumes U-Pb concordance and no Pb loss but, unlike the [208]Pb correction, does not
assume an undisturbed U/Th ratio. Because Pb loss is not considered, all corrected dates are (possibly) minimum ages. The
equations given here are modified for detrital samples with unknown co-genetic minerals, previously described in Faure
(1986), Williams (1997), Chew et al. (2011), and the semitotal-Pb/U scheme of Ludwig (1998) and Vermeesch (2020).
The calculation is similar to the [208]Pb correction. First, the proportion of [206]Pb_common is calculated as

$$f_{206} = \frac{\left(^{207}Pb/^{206}Pb_{measured}\right) - \left(^{207}Pb^*/^{206}Pb^*\right)}{\left(^{207}Pb/^{206}Pb_{common}\right) - \left(^{207}Pb^*/^{206}Pb^*\right)}$$

235                          (10)

where [207]Pb/[206]Pb_measured is taken directly from the raw data. The [207]Pb/[206]Pb_common ratio is based on the two-stage Pb evolution
model of Stacey and Kramers (1975), which is calculated as the ratio of Equation (3) and Equation (11) for dates older than
3.7 Ga or as the ratio of Equation (5) and Equation (12) for dates younger than 3.7 Ga:

$$\left(\frac{^{207}Pb}{^{204}Pb}\right)_{common} = \frac{7.19}{137.88} \cdot \left(e^{\lambda_{235} \cdot 4.57 x 10^9} - e^{\lambda_{235} \cdot t_i}\right) + 10.294$$

240                          (11)

or

$$\left(\frac{^{207}Pb}{^{204}Pb}\right)_{common} = \frac{9.74}{137.88} \cdot \left(e^{\lambda_{235}\cdot 3.7x10^9} - e^{\lambda_{235}\cdot t_i}\right) + 12.998$$

(*12*)
where $t_i$ is the initial date estimate in years and the $^{235}$U decay rate $\lambda_{235}$ is 9.8485x10$^{-10}$ yr$^{-1}$ (Faure, 1986). Here, for $t_i$ we use
the $^{206}$Pb/$^{238}$U date from the iolite data reduction. However, Chew et al. (2011) demonstrated that the choice of initial date
results in a < 0.05% difference in the final $^{207}$Pb-corrected date after 5 iterations. The expected radiogenic $^{207}$Pb/$^{206}$Pb* ratio is
calculated as

$$\frac{^{207}Pb^*}{^{206}Pb^*} = \left(\frac{^{235}U}{^{238}U}\right) \cdot \left(\frac{e^{\lambda_{235}t_i} - 1}{e^{\lambda_{238}t_i} - 1}\right)$$

(*13*)
where $^{235}$U/$^{238}$U is 137.88 (Steiger and Jäger, 1977). Finally, the radiogenic component, the $^{206}$Pb*/$^{238}$U ratio, can be calculated
using Equation (8) and then used to solve the age equation (Equation (9)). As with the $^{208}$Pb correction, to iteratively calculate
the date, each iteration replaces $t_i$ with the previously calculated $^{206}$Pb*/$^{238}$U date. The $^{207}$Pb-corrected date presented here is
from the two hundredth iteration. The uncertainty on the date is calculated as the equivalent of the percent (propagated)
uncertainty of the uncorrected $^{206}$Pb/$^{238}$U ratio (Odlum et al., 2019). For example, if the initial $^{206}$Pb/$^{238}$U ratio has 2%
uncertainty at 2 sigma and the corrected date is 200 Ma, then the corrected date uncertainty is ± 4 Ma (2s).
**2.3.5 Discordance**
Although there are various ways to calculate the discordance of U-Pb analyses, which are reviewed elsewhere (e.g.,
Vermeesch, 2021), it remains unclear which metric is best for common Pb bearing minerals and if a discordance threshold
should be applied. One family of discordance metrics relies on the difference between the $^{206}$Pb/$^{238}$U date and $^{207}$Pb/$^{206}$Pb date
(e.g., Gehrels, 2011). Because $^{207}$Pb and $^{208}$Pb corrections force concordance, these metrics are not applicable to common Pb
bearing minerals. Two metrics potentially relevant to common Pb-bearing minerals are the Stacey-Kramers distance and
Aitchison distance (after Vermeesch, 2021). The Stacey-Kramers distance is calculated by first using the U-Pb analysis and
$^{207}$Pb/$^{206}$Pb$_{common}$ composition (calculated during common Pb correction) to find the discordia in Tera-Wasserburg space, then
discordance is calculated as the distance between the measured $^{238}$U/$^{206}$Pb and $^{207}$Pb/$^{206}$Pb coordinates and the concordia
intersection ($\delta_2$) along the total discordia line distance ($\delta_1 + \delta_2$) (Figure 1; Vermeesch, 2021):
$$\text{Concordance} = \delta_1/(\delta_1 + \delta_2)$$
(*14*)
If a discordance threshold is applied, the Stacey-Kramers distance approach includes more young dates than old dates (> 1000
Ma) due to the change in concordia slope around 1000 Ma (Vermeesch, 2021). A second metric is the Aitchison distance
(Aitchison, 1982; Pawlowsky-Glahn et al., 2015) which calculates the Euclidean distance from the measured $^{238}$U/$^{206}$Pb and
$^{207}$Pb/$^{206}$Pb coordinates to the concordia line in log-ratio Tera-Wasserburg space (Figure 1; Vermeesch, 2021). We compare

these two metrics with our new dataset. Additionally, detrital zircon studies commonly use a discordance threshold that excludes analyses with discordance above 5-30%, typically around 10% (Spencer et al., 2016), which can induce bias (Nemchin and Cawood, 2005; Malusà et al., 2013). The application of a discordance threshold has been underexplored in detrital rutile, with most studies applying no discordance filter, perhaps due to the lack of consensus on how to define discordance in common Pb bearing minerals. Rather, a group of studies proposes to filter analyses based on the percent uncertainty of the corrected date (Mark et al., 2016; Govin et al., 2018; Chew et al., 2020; Caracciolo et al., 2021). It is noted that there is little guidance on how uncertainties are calculated and propagated during Pb correction, which ought to be investigated in future work; meanwhile, the filters should be applied with care. We explore these thresholds with our new dataset.

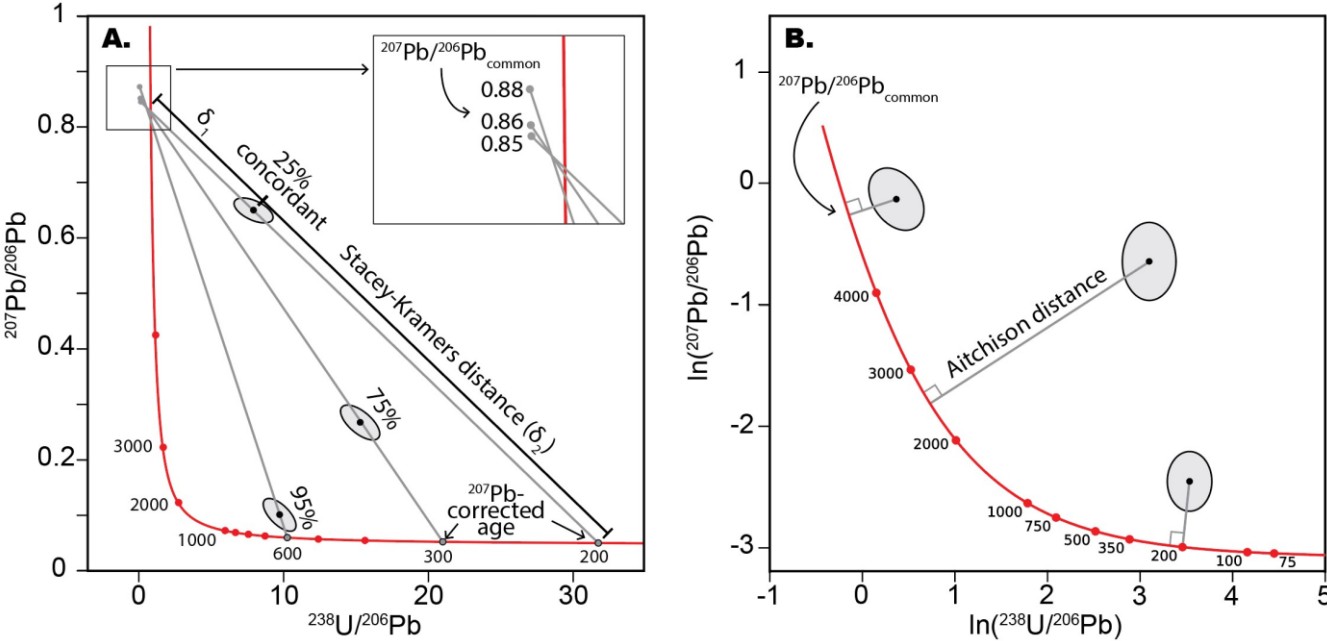

Figure 1. Conceptual schematics of the $^{207}$Pb correction and Stacey-Kramers distance (A) and Aitchison distance (B). (A) For the $^{207}$Pb correction, first, the common $^{207}$Pb/$^{206}$Pb ratio is calculated from the initial date estimate ($t_i$). Next, a discordia is fitted between $^{207}$Pb/$^{206}$Pb$_{common}$ and the data point. Then, the lower intersection of the line with the concordia marks the corrected $^{238}$U/$^{206}$Pb and $^{207}$Pb/$^{206}$Pb, which are used to calculate the $^{207}$Pb-corrected date. The Stacey-Kramers distance defines concordance as the distance along the discordia between the upper and lower intersections of the discordia with the concordia (Equation (14)). (B) The Aitchison distance calculates the Euclidean distance between the analysis and concordia curve in log-ratio space, where higher distance values are considered more discordant. Figure modified from Vermeesch (2021).

## 3. Geologic Context

Anatolia is composed of a series of subduction complexes, island arcs, and continental terranes that accreted and collided from the Late Paleozoic through Cenozoic during the progressive opening and closing of Paleotethys and Neotethys seaways (Şengör and Yilmaz, 1981). Today, northwestern Anatolia comprises, from structurally highest (north) to lowest (south), the continental Pontides, including the Cretaceous–Eocene forearc-to-foreland Central Sakarya and Sarıcakaya Basins, the Permian–Triassic Karakaya Complex, the İzmir-Ankara-Erzincan suture zone and associated Neotethys ophiolites and mélange, and the lower plate Anatolide-Trauride continental terranes (Figure 2). The Pontides basement contains Paleozoic paragneiss, schist, and amphibolite rocks intruded by Carboniferous granitoids emplaced during the Variscan orogeny (Göncüoğlu et al., 2000; Ustaömer et al., 2012). The nature of the Karakaya Complex is debated but is generally considered a subduction-accretion complex associated with the Late Paleozoic–Early Mesozoic closure of the Paleotethys along the southern margin of Eurasia (Pickett and Robertson, 1996; Okay and Göncüoglu, 2004; Federici et al., 2010; Ustaömer et al., 2016). The Karakaya Complex contains metamafic and metasedimentary rocks interpreted as seamounts of intra-oceanic basaltic composition and forearc basin and trench deposits (Pickett and Robertson, 1996) that were subsequently metamorphosed to blueschist and epidote-amphibolite with minor eclogite facies with estimated temperatures of 340–550 ± 50 °C (Okay et al., 2002; Federici et al., 2010) with phengite, glaucophane, and barroisite Ar-Ar cooling dates around 200–215 Ma (Okay et al., 2002). The youngest Karakaya Complex units are unmetamorphosed or metamorphosed to zeolite to lower greenschist facies (120–376 °C) (Federici et al., 2010) and are unconformably overlain by Jurassic platform carbonates. The Cretaceous to present closure of the Neotethys and associated suturing is recorded in the Central Sakarya and Sarıcakaya Basins located north of the suture. Stratigraphic and paleocurrent (Ocakoğlu et al., 2018), provenance (Mueller et al., 2022; Campbell et al., 2023), and mudstone geochemistry records (Açıkalın et al., 2016) show the input of suture zone derived material into the Central Sakarya Basin from the Late Cretaceous through Eocene, interpreted as progressive suture zone uplift and exhumation during accretion and continental collision (Ocakoğlu et al., 2018; Okay et al., 2020; Mueller et al., 2022; Campbell et al., 2023). Cretaceous subduction-related arc volcanism and Paleogene syn-collisional volcanic centers are located within and to the north of the basins (Kasapoğlu et al., 2016; Ersoy et al., 2017, 2023; Keskin and Tüysüz, 2018). By Eocene times, continued collision increased plate coupling which manifested as increased contractional deformation that partitioned the southern Central Sakarya Basin into the Sarıcakaya Basin along the basement-involved Tuzaklı-Gümele Thrust (also termed the Söğüt Thrust or Nallıhan Thrust) (Mueller et al., 2022). The Eocene Sarıcakaya Basin received sediment from the suture zone and Karakaya Complex to the south and basement-involved thrust sheets to the north (Mueller et al., 2019).

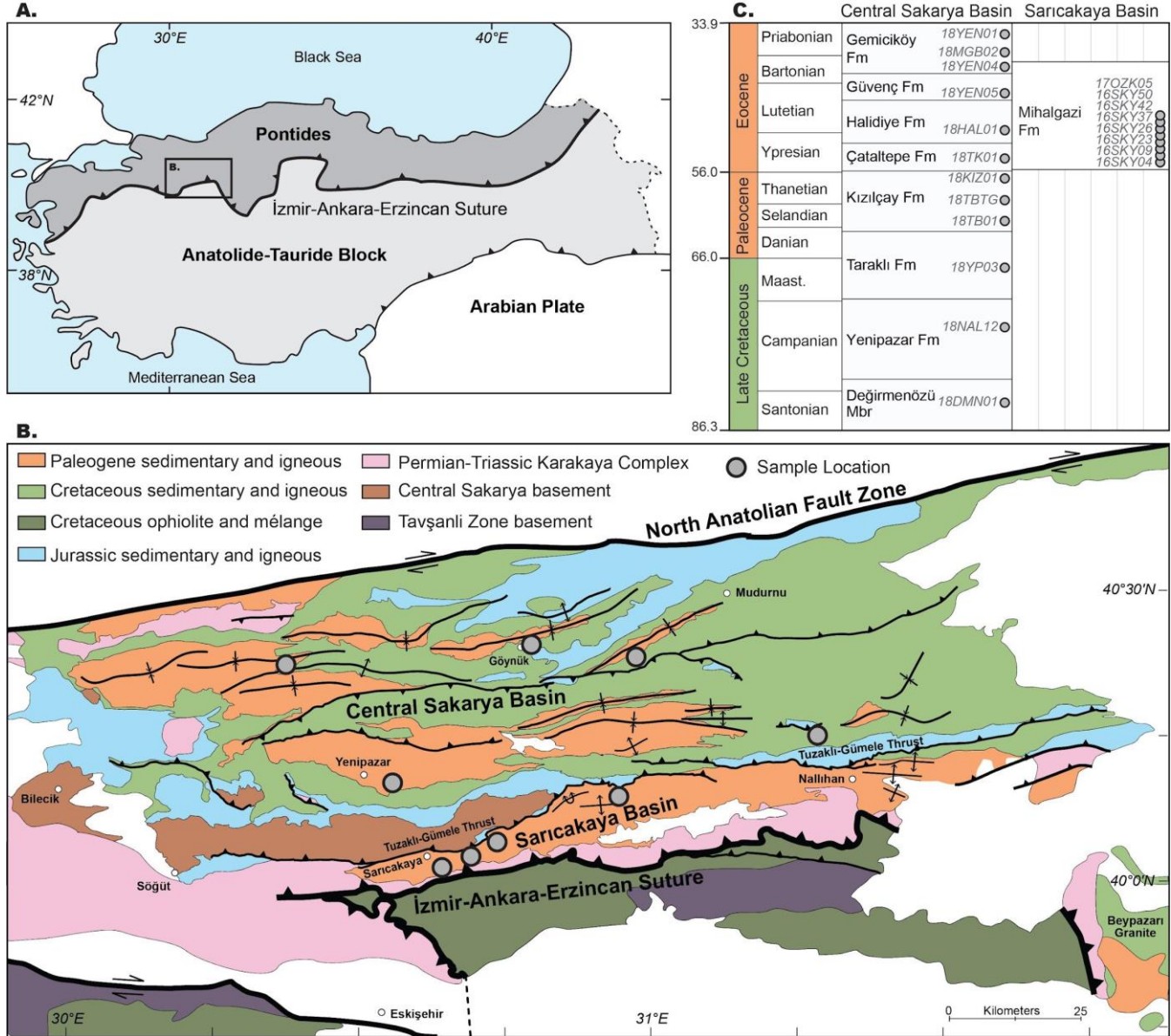


*Figure 2: (A) Simplified terrane map of Anatolia and (B) geologic map of the Central Sakarya Basin and Sarıcakaya Basin*
*region (after Aksay et al., 2002). (C) Simplified stratigraphic correlation chart and schematic sample distribution. Stratigraphy*
*after Ocakoğlu et al. (2018).*

**4 Methods**

**4.1 Sample Information**

Sedimentary rock samples were collected from Upper Cretaceous to Eocene siliciclastic sections in the Central
Sakarya Basin and Sarıcakaya Basin in western Anatolia (Figure 2; Table S1). Detrital zircon U-Pb ages and Hf isotopes from
these samples are already published (cf. Section 8; Mueller et al., 2019, 2022; Campbell et al., 2023); a set of 20 samples were
chosen for detrital rutile U-Pb dating and trace element analysis. Heavy minerals were extracted using standard heavy mineral
techniques, including crushing, water table, heavy liquid, and magnetic separation (see supporting information). Rutile grains
were handpicked from the ≥0.3 amp. magnetic fraction using a Leica M205C binocular microscope. Three samples—
16SKY26, 16SKY42 and 17OZK05—yielded hundreds of rutile grains and we handpicked 260–320 rutile grains from each
sample; for samples with smaller yield, all grains were picked. The low yield of rutile grains partially contributes to the low-$n$
date distributions of the individual samples. Rutile grains were mounted in epoxy and polished to expose the internal structure.
Rutile mounts were carbon coated and imaged with a TFS Apreo-S with Lovac SEM with an energy-dispersive detector (EDS)
to distinguish $TiO_2$ grains from other heavy minerals (Figure S1).

**4.2 U-Pb Analytical Protocol**

Detrital rutile U-Pb geochronology was conducted at the Isotope Geochemistry Lab at the University of Kansas (KU-
IGL) using a Thermo Element2 magnetic sector field ICP-MS coupled to a Photon Machines AnalyteG2 excimer laser ablation
system. The protocol was modified from Rösel et al. (2019) to optimize for low U contents (Text S1; Table S2). The ICP-MS
was manually tuned using NIST SRM 612 reference material glass to optimize for high sensitivity and low oxide production.
Grains were ablated for 25 seconds with a laser beam diameter of 50 μm, laser fluence of 3.0 J/cm$^2$, and 10 Hz repetition rate.
The U-Pb data were collected in 4 analytical sessions. The analytical protocol was modified from session to session to optimize
for the analysis of low U and Pb unknowns and high U and Pb reference materials. In the first two analytical sessions, 21RtF
and 21RtG, Pb and Th isotopes were measured with the secondary electron multiplier operating in counting detection mode,
whereas Pb and Th isotopes were measured with the secondary electron multiplier in both counting and analog modes ('both
mode') for the final two sessions, 21RtA and 21RtB. Primary and secondary reference materials were the R10 (1091.6 ± 3.5
Ma by TIMS, 2s abs.; Luvizotto et al., 2009), Wodgina (2845.8 ± 7.8 Ma by TIMS; Ewing, 2011), 9826J (381.9 ± 1.1 Ma by
TIMS; Kylander-Clark, 2008), LJ04-08 (498 ± 3 Ma by LA-ICP-MS; Apen et al., 2020), and Kragerø (1085.7 ± 7.9 Ma by
TIMS; Kellett et al., 2018). For U-Pb analyses, the analysis of 5-8 unknowns was followed by 2 standards, the primary standard
R10 and one of the secondary standards. The data were reduced in iolite 4 (Paton et al., 2011), calibrated against R10
uncorrected for initial Pb, and using the weighted linear fit drift correction which reproduced secondary standard ages and
brought their MSWDs closest to 1. The concordia ages are satisfactory for all reference materials, except for the Wodgina and
Kragerø, which did not perform well during the first two analytical sessions—likely due to [206]Pb counts per second exceeding
the limit of linear behavior in counting detection mode—and are discarded from those analytical sessions. Standard
reproducibility is discussed further in the supplemental text included in the data repository; U-Pb data are provided in the data
repository (Mueller et al., 2023).

**4.3 Trace Element Geochemistry Analytical Protocol**

Detrital rutile trace element geochemistry ([49]Ti, [51]V, [53]Cr, [66]Zn, [69]Ga, [90]Zr, [93]Nb, [95]Mo, [118]Sn, [121]Sb, [177]Hf, [181]Ta, [182]W)
was conducted at the KU-IGL using the same instrumentation and parameters, except with a 25 or 35 μm spot size. Reference
materials included USGS GSD-1G and USGS GSC-1G glasses (Jochum et al., 2011) and R10 rutile (Luvizotto et al., 2009).
For trace element analysis, the analysis of 5−10 unknowns was followed by analysis of 2 standards, the primary standard GSD-
1G and one of the secondary standards. Trace element concentrations were calculated using the Trace Element routine in iolite
4 with [49]Ti as an internal standard; for rutile unknowns, $TiO_2$ was set to be 100 mass-% (e.g., Plavsa et al., 2018; Rösel et al.,
2019). Standard reproducibility is discussed in the supporting information in the data repository (Text S2). In short, for the
secondary standard GSC-1G, all elements are within 10% of the published values except for Sn and Ga, and for the secondary
standard R10, all results are within the range of reported values. Following U-Pb and trace element analysis, mounts were
imaged in an SEM at University of Nevada Reno (Figure 3). Most grains have both U-Pb and trace element results, but some
grains have only U-Pb results due to the grains being too small for a second ablation spot or only trace element results due to
discarded U-Pb data. Detrital rutile trace element data are given in the data repository (Mueller et al., 2023).

**4.4 Additional Data Workflows**

Additional data reduction and data calculations steps were performed. Provided as a complement to this manuscript
are open access Jupyter Notebooks that contain the Python and R code used to perform these additional calculations and to
generate figures, which are briefly described here (Mueller, 2024). (1) The [208]Pb and [207]Pb corrections were performed in the
Detrital-Common-Pb-Corrections notebook using the equations detailed in Section 2 above. The notebook allows for either a
manually set number of iterations or to iterate until all analyses are below a given threshold—the percent difference in corrected
date between the current and previous iteration. Presented here are the results from the 200[th] iteration. (2) The UPb-Plotter
notebook visualizes the uncorrected U-Pb results in Tera-Wasserburg space, compares metrics for excluding analyses based
on uncertainty filters (Section 5.3), and calculates discordance using the Stacey-Kramers and Aitchison distances (Section 2;
Figure 1). (3) The Rutile-Trace-Elements notebook includes the calculations and resulting figures for exploring $TiO_2$
polymorphs, mafic and pelitic protoliths, Zr-in-rutile thermometry, and low U contents. Here, rutile grains are classified as
mafic or pelitic based on the Cr-Nb discrimination fields of Triebold et al. (2012), and Zr-in-rutile temperatures are calculated
with the Kohn (2020) formulation (Equation (1)) at 13 kbar. (4) The Detrital-PCA-R notebook performs principal component
analysis on trace element data using the pcaCoDa function in the robCompositions library, which is designed to handle
compositional data (Templ et al., 2011). Due to the variable performance of Sn and Ga in the secondary standards, these
elements were excluded from the PCA (Supplemental Text S2, Figure S6). Additionally, Mo and Sb were excluded because
grains with very low or zero concentrations influence the results to be artificially dominated by these elements. (5)
Additionally, the UPb-Timeseries notebook is provided for visualizing U-Pb timeseries data.

## 5 U-Pb Geochronology Results

### 5.1 U-Pb Data Quality

A total of 1,278 detrital rutile grains were analyzed for U-Pb geochronology. A significant number of analyses were
rejected and excluded, as discussed below. We aim to be transparent in data reporting—including the number of grains
analyzed and the criteria for rejection—in order to give precedence for this practice, which is missing in the literature, and to
explore the current limitations of large-$n$ detrital rutile studies. Even with the optimized LA-ICP-MS protocol, a significant
number of analyses did not meet quality control goals: 665 of 1,277 (54%) analyses were rejected due to anomalous (spiky)
patterns in raw signal intensity, or low U or low Pb signal intensity. Figure 3 depicts representative examples of signal intensity
in accepted and rejected analyses. Inclusions and anomalous patterns were easily spotted through monitoring $^{206}Pb$, $^{207}Pb$, $^{238}U$,
$^{232}Th$, $^{206}Pb/^{238}U$ and $^{207}Pb/^{206}Pb$ channels. In some instances, the signal of an inclusion or anomalous (spiky) pattern was short
enough that the integration window could be shortened to exclude it. In other cases, the non-inclusion signal could not be
isolated and the entire analysis was discarded. Potential causes for the abnormal signal patterns and high Pb uncertainty include
(1) elemental heterogeneity from ablating into small inclusions and/or lamellae; (2) inhomogeneities due to micro-cracks with
different element/isotope composition; (3) heterogeneous amount of common lead incorporation during rutile growth; (4)
textural and/or elemental heterogeneities due to multiple rutile growth events. Although, scenarios 3 and 4 are unlikely for Pb
because it diffuses and should not cause spikes.
The SEM images do not give a clear picture of how to better select grains that will produce acceptable signal intensity
and U-Pb concordance. Figure 3 shows SEM images of representative rutile grains after laser ablation. All grains appeared
inclusion-free before ablation, yet some analyses clearly ablated into inclusions (Figure 3b,e). The large laser spot size of 50
μm gives a higher signal, which is better for grains with potentially low U or low Pb concentrations, but the potential trade-off
is increasing the likelihood of hitting inclusions. Grains with obvious inclusion lamellae generally yielded poor data quality.

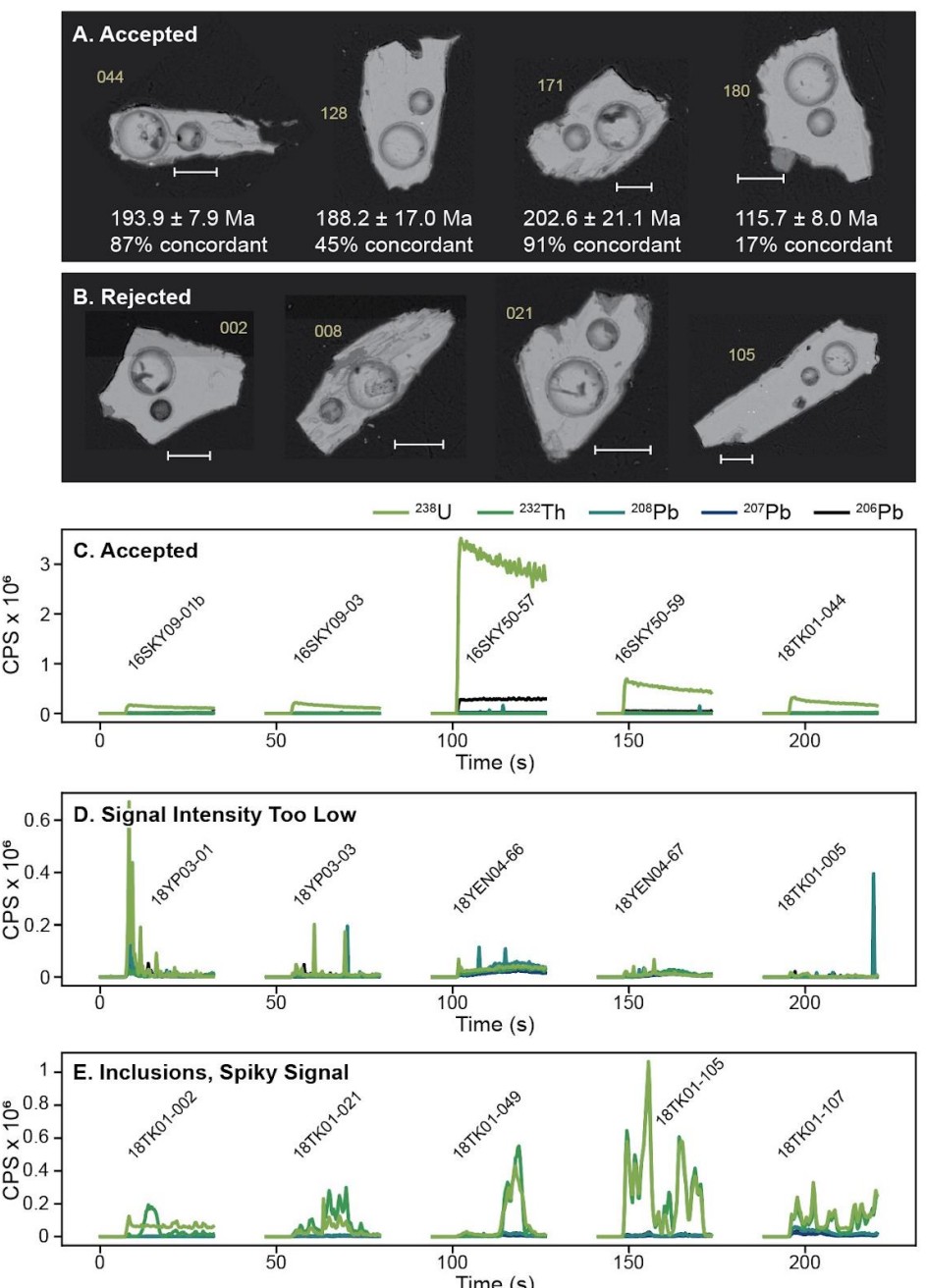

Figure 3: SEM BSE images and U-Pb signal intensities of representative rutile grains. (A) Rutile grains with acceptable U-Pb analyses across a range of concordance. U-Pb date and concordance are from the [207]Pb correction method and Stacey-Kramers metric, respectively. Ablation pits are from U-Pb analysis (larger) and trace element analysis (smaller). The scale bar is 50 µm. All grains are from sample 18TK01; the grain number is in yellow. (B) Images of rutile grains with U-Pb analyses rejected because of inclusions (18TK01-002) or spiky signal (18TK01-008, -021, -105). (C-E) Representative U-Pb raw signal intensity patterns of accepted analyses (C) and rejected analyses from too low signal intensity (D) or inclusions and/or spiky signal (E).

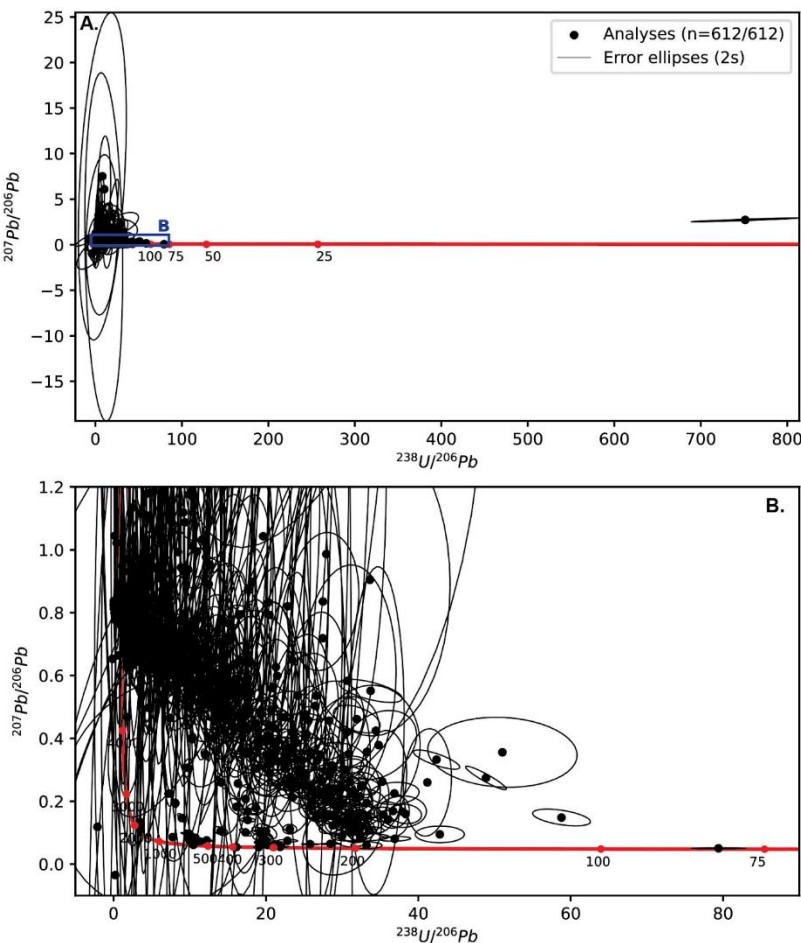

417

*Figure 4. Uncorrected detrital rutile U-Pb results displayed in Tera-Wasserburg space. Uncertainty ellipses are 2s propagated. The area displayed in (B) is highlighted by the blue box in (A).*

## 5.2 U-Pb Geochronology and Common Pb Correction Results

The uncorrected U-Pb results are displayed in Figure 4. We note that all concordia diagram figures display the uncorrected U-Pb data; common Pb corrections force concordance and the corrected data are displayed as date distributions. A number of analyses plot close to the concordia curve and many plot along the discordia trend toward common Pb values. Both $^{208}$Pb- and $^{207}$Pb-corrections were performed on the uncorrected U-Pb analyses. After 200 iterations, the $^{208}$Pb- and $^{207}$Pb-corrections resulted in 547 and 487 corrected dates between 0 Ma and 4500 Ma, respectively. These numbers differ because no corrected date is calculated when the proportion of $^{206}$Pb$_{common}$ is greater than 1, and because the common Pb corrections can yield dates younger than 0 Ma or significantly older than 4500 Ma depending on the calculated proportion of $^{206}$Pb$_{common}$ ($f_{206}$). The Pb corrected U-Pb data are shown in Figure 5 as kernel density estimates (KDEs) and cumulative KDE distributions.

The date distributions of individual samples are given in Figure S7, but due to small sample sizes, interpretations are based on
the cumulative dataset.

431       The two different Pb corrections produce similar date distributions (Figure 5). For both distributions, the main date
peak is at ca. 185 Ma with a minor peak around 297 Ma. The [207]Pb and [208]Pb distributions vary in the presence and amplitude
of minor Paleozoic and older populations. The [208]Pb correction results include more Devonian and older grains (n=131/547,
24%) than the [207]Pb correction (n=68/487, 14%).

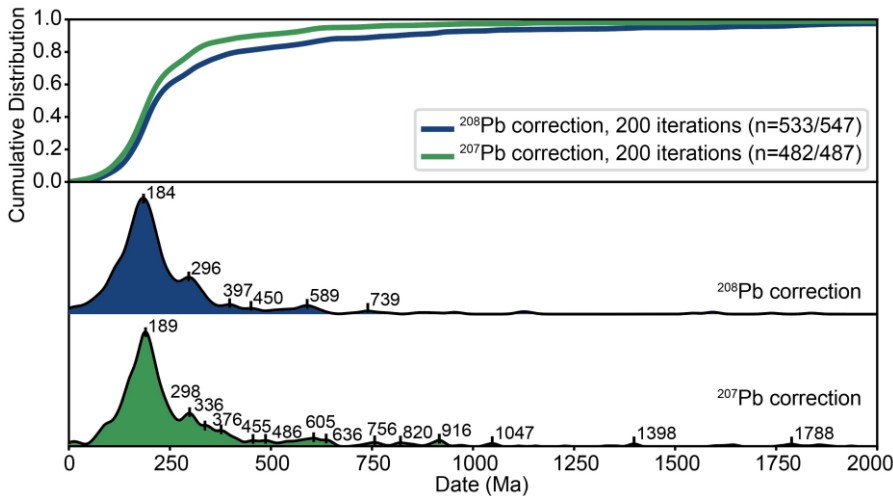

*Figure 5. The $^{208}$Pb and $^{207}$Pb corrected date distributions from 0 to 2000 Ma displayed as normalized kernel density estimates*
*and cumulative KDE distributions, visualized with detritalPy (Sharman et al., 2018). No discordance or uncertainty filter is*
*applied.*
**5.3 Uncertainty and Discordance Thresholds**

441       Detrital U-Pb data can further be filtered by U-Pb ratio uncertainty, date uncertainty, or discordance thresholds.
Because the uncertainty on the corrected date is calculated from the uncertainty on the measured $^{206}$Pb/$^{238}$U ratio (cf. Section
2), these metrics are similar. Figure 6 displays the results of three uncertainty threshold filters: (1) 20% uncertainty on
$^{238}$U/$^{206}$Pb and $^{207}$Pb/$^{206}$Pb ratios (modified from Lippert, 2014), (2) a date-dependent filter that excludes analyses with > 10%
date uncertainty for corrected dates > 100 Ma, > 20% uncertainty for dates 10–100 Ma, or > 25% uncertainty for dates < 10
Ma (after Govin et al., 2018), and (3) a power law threshold that excludes analyses if the percent uncertainty on the $^{207}$Pb
corrected date exceeds the function: (t ^ -0.65)*8 (after Chew et al., 2020). The results of these filters are displayed as
uncorrected U-Pb data in Tera-Wasserburg space and $^{207}$Pb corrected date distributions (Figure 6). From the $^{207}$Pb corrected
analyses total (n=487), the above thresholds exclude an additional 108 (22%), 191 (39%), and 46 (9%) analyses, respectively.
The power law function excludes the fewest number of analyses.

451       The three filters have similar $^{207}$Pb corrected date distributions (Figure 6). The main age modes identified in all three
filters are 183 Ma, 300 Ma and 400 Ma. Minor Devonian and older date modes are present. Only the date-dependent filter
identifies the 89 Ma date mode and it includes a 9 Ma mode that is significantly younger than the youngest sampled strata
(Bartonian–Priabonian). The U-Pb ratio and power law filters have nearly identical date peaks with the power law filter
including more grains, especially in the ~183 Ma mode.

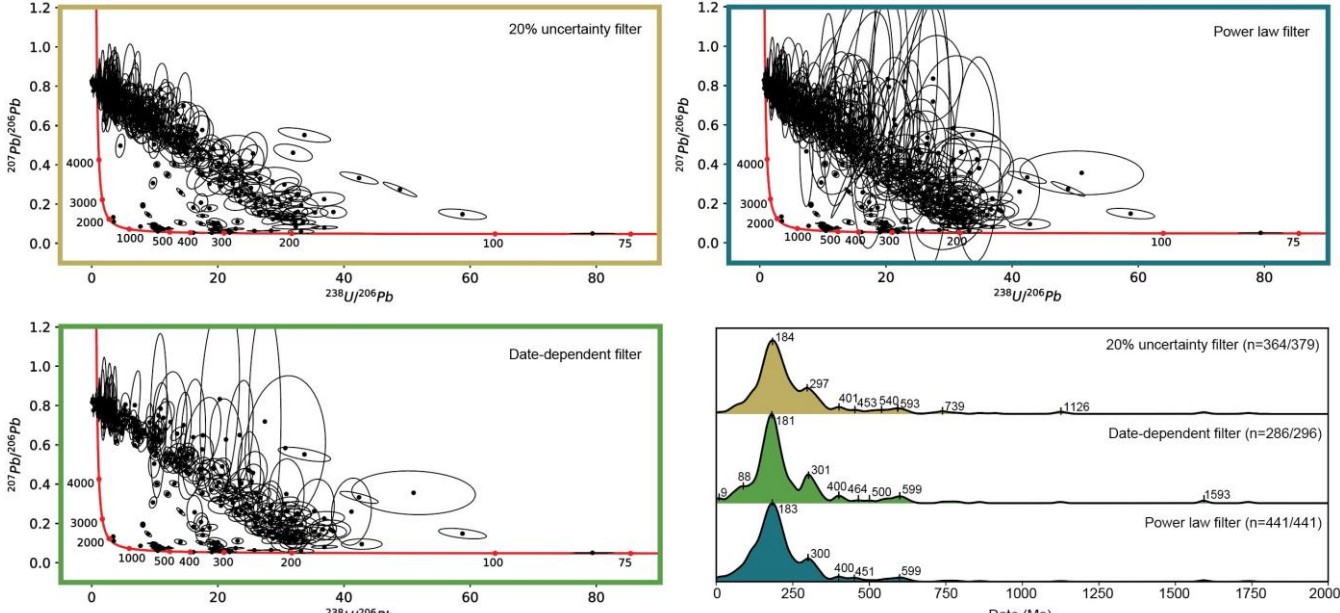


*Figure 6. Comparison of U-Pb data filters based on U-Pb ratio and date uncertainties, displayed in Tera-Wasserburg space (uncorrected) and normalized kernel density estimates ($^{207}Pb$-corrected). The U-Pb ratio uncertainty filter (yellow) excludes all analyses with $^{238}U/^{206}Pb$ and $^{207}Pb/^{206}Pb$ ratio uncertainties above 20% (modified from Lippert, 2014). The date -dependent filter (green) excludes analyses based on the $^{207}Pb$-corrected date and uncertainty (see text; after Govin et al., 2018); after Govin et al., 2018). The power law filter (blue) excludes analyses if the percent uncertainty on the $^{207}Pb$ corrected date exceeds the given power law function (see text; after Chew et al., 2020).*


To quantify discordance in common Pb bearing minerals, two metrics are considered: Aitchison and Stacey-Kramers
distances (Figure 1). The results are shown in Figure 7 in Tera-Wasserburg space with uncorrected U-Pb analyses colored by
distance (concordance). The Aitchison distance is calculated as the Euclidean distance between the analysis and concordia
curve in log-ratio space, where higher distance values are considered more discordant. The results show that analyses closest
to concordia are the least discordant (most concordant). This means that analyses close to the lower concordia curve and the
common Pb composition are considered less discordant (more concordant) whereas analyses in the middle space are considered
most discordant (Figure 7b). In the Stacey-Kramers distance formulation, discordance is calculated from the distance between
the analysis and the upper and lower intercepts (Equation (14)). In this case, analyses closest to the common Pb composition
are considered most discordant (Figure 7c). If a discordance filter were applied based on the Aitchison distance, analyses in
the middle space of the concordia diagram would be excluded, whereas a discordance filter based on the Stacey-Kramers

distance would exclude analyses closer to the common Pb composition. The Stacey-Kramers distance appears to reflect U-Pb systematics in common Pb bearing minerals and is a representative metric of discordance.

*Figure 7. Comparison of (A,B) Aitchison distance and (C) Stacey-Kramers distance as metrics for discordance in common Pb bearing minerals. For simplicity all uncorrected U-Pb data are shown as circles rather than error ellipses. Circles are color-coded by distance (concordance). The Aitchison distance results are shown in Tera-Wasserburg concordia diagrams in original (B) and log-ratio space (A). The 185 Ma isochron is displayed in both diagrams. Circles closest to the concordia have the lowest discordance (highest concordance). (C) The Stacey-Kramers distance results are shown in Tera-Wasserburg space, where the gray lines are individual discordia and light gray circles are intersection points. Uncorrected U-Pb circles are color-coded for percent distance along the total discordia distance (from common Pb composition to lower intersection point). Circles closest to the lower concordia intercept have the lowest discordance. Dark gray circles are U-Pb analyses without Stacey-Kramers distance values (no lower intersection point due to positive discordia line slope, for example) or without $^{207}Pb$-corrected dates (due to $f_{206} > 1$).*

The U-Pb dates are subdivided into bins based on their Stacey-Kramers concordance values. Figure 8 displays the
[207]Pb-corrected date distributions filtered using the power law threshold. The 100-80% concordance group has the most discrete
date modes at 189 Ma, 307 Ma, 608 Ma, and 1593 Ma. The 80-60%, 60-40% and 40-0% bins have unimodal age distributions
that are asymmetric toward older dates, and have a dominant age mode around 180 Ma. The cumulative distributions reveal
that the distribution of all grains together has a similar distribution to that of the 40-0% group (Figure 8 top). Comparison of
the whole distribution to the 100-80% concordance group reveals that, if a 20% discordance filter were applied similar to
detrital zircon U-Pb workflows, the same general date modes would be identified. However, the addition of lower concordance
grains (i.e., 80-0% concordance groups) broadens the Jurassic peak and shifts it slightly younger from 189 Ma to ~180 Ma,
decreases the amplitude of the Carboniferous and Proterozoic peaks, and increases the amplitude of the ~400-450 Ma peaks.

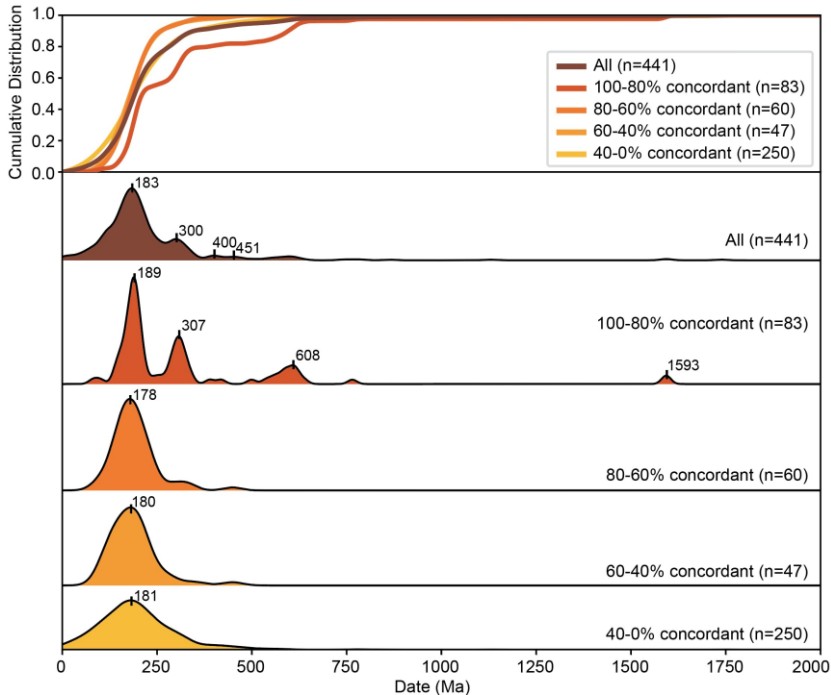

*Figure 8. Relative kernel density estimates (KDEs; bottom panels) and cumulative KDE distributions (top) of [207]Pb-corrected,*
*power law uncertainty filtered dates categorized by discordance from Stacey-Kramers distance values.*
**6 Trace Element Geochemistry Results**
**6.1 Metamorphic Protolith**
Trace element results are provided in the data repository. Discrimination diagrams using V, Cr, Zr, Fe, and Nb can
distinguish rutile from other $TiO_2$ polymorphs (Triebold et al., 2011), and all analyzed grains plot within the rutile field (Figure
S2). The Cr and Nb concentrations discriminate between metapelitic and metamafic source rocks (Zack et al., 2004a; Triebold
et al., 2011, 2012). Even though there are multiple proposed discrimination lines between metamafic and metapelitic source
lithologies (e.g., Meinhold et al., 2008; Triebold et al., 2012), the detrital rutile in this dataset plot in both the metamafic (33%)
and metapelitic (67%) fields (Figure 9a). There is no clustering of protolith by U-Pb date, with prominent date modes
containing both metamafic and metapelitic grains (Figure 9b). While some metamafic grains plot close to concordia (more
concordant), many plot close to the common Pb composition concordia intercept (more discordant).

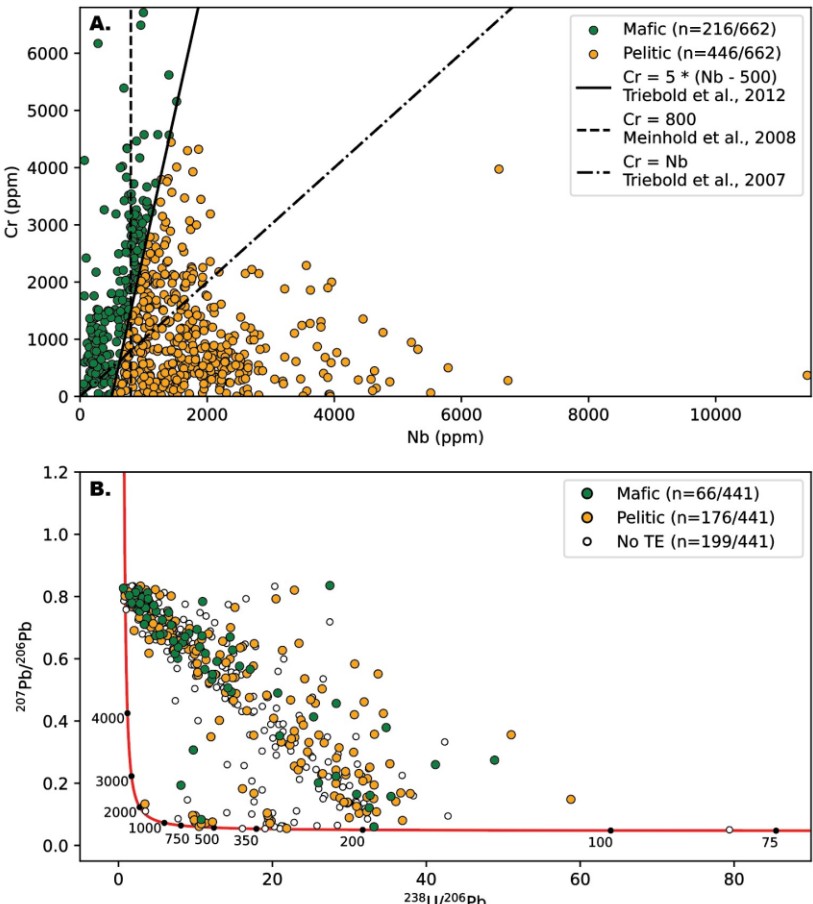


*Figure 9. (A) Protolith discrimination diagram. Grains are classified as (meta)mafic and (meta)pelitic based on the Triebold*
*et al. (2012) line, with the Triebold et al. (2007) and Meinhold et al. (2008) lines also shown. (B) Concordia diagram of*
*uncorrected U-Pb circles colored by protolith classification. The power law filter is applied. Open circles represent grains*
*with U-Pb data but no trace element data (TE). Sample size differs between plots because not all grains have both U-Pb and*
*trace element data.*

### 6.2 Zr-in-Rutile Temperature and Uranium Concentration


The Zr-in-rutile temperatures were calculated using the Kohn (2020) calibration (Equation (1)) at 13 kbar with an
uncertainty of 5 kbar; results are included in the data repository. The Zr concentrations range from 2 to 1934 ppm, yielding
source rock minimum peak temperatures from 336 ± 15 °C to 849 ± 28 °C. The Zr-in-rutile temperature results are displayed

alongside U concentration and colored by protolith (Figure 10). There is not a correlation between Zr-in-rutile temperature and protolith. The majority of grains have moderate temperatures corresponding to greenschist to blueschist facies conditions: 68% (n=147/216) of mafic and 67% (n=301/446) of pelitic grains are below 500 °C. When displayed in Tera-Wasserburg space, dominant date modes—90 Ma, 185 Ma, 300 Ma, 500–650 Ma—have fairly consistent Zr-in-rutile temperatures (Figure 11). The highest temperatures, reaching granulite facies conditions, are found in the 90 Ma date mode. The 500–650 Ma and 300 Ma rutile grains similarly preserve high temperatures, up to 700–820 °C, whereas the majority of 185 Ma grains have temperatures in greenschist to blueschist facies around 450–550 °C.

The uranium concentrations range from 0.0006 to 113 ppm. These low values are within the detection limit. The primary standard, R10, has a U concentration of 44 pm (Luvizotto et al., 2009) and, in our measurements, on average, 2.1 million CPS $^{238}$U (i.e., ~50,000 counts/ppm). The $^{238}$U baseline was about 5 CPS, therefore, the instrument set-up has a detection limit of about 0.0003 ppm $^{238}$U (calculated from 3x background). All analyses are above the detection limit, with 91% (n=555/612) of analyses at least an order of magnitude above this limit. The comparison of Zr-in-rutile temperatures with U concentration reveals that the majority of low U rutile (< 4 ppm) are within greenschist to blueschist facies conditions (68%, n=205/303 below 500 °C; Figure 10). Additionally, mafic classified grains are dominantly low U (95%, n=106/112 below 4 ppm). The majority of rutile with U contents above 4 ppm are classified as pelitic (85%, n=34/40) and generally have higher Zr contents.

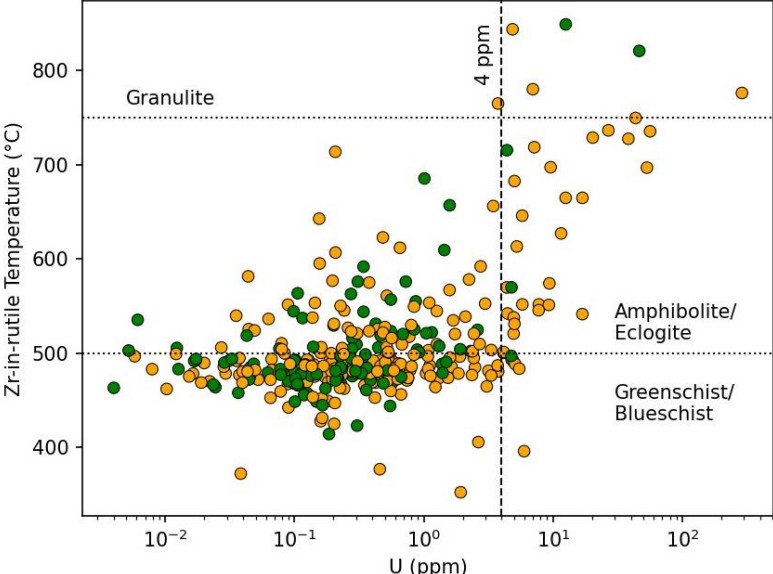

*Figure 10. Zr-in-rutile temperature versus U concentration. Mafic and pelitic discrimination is from Cr and Nb concentrations (Figure 9) mafic protoliths shown in green, pelitic in orange. The 4 ppm U line demarcates grains included/excluded by a U filter. Zr-in-rutile temperatures follow the Kohn (2020) calibration. Note that not all analyses have both U and trace element (TE) data, therefore there are fewer grains represented in this scatter plot than in Figure 9.*

545

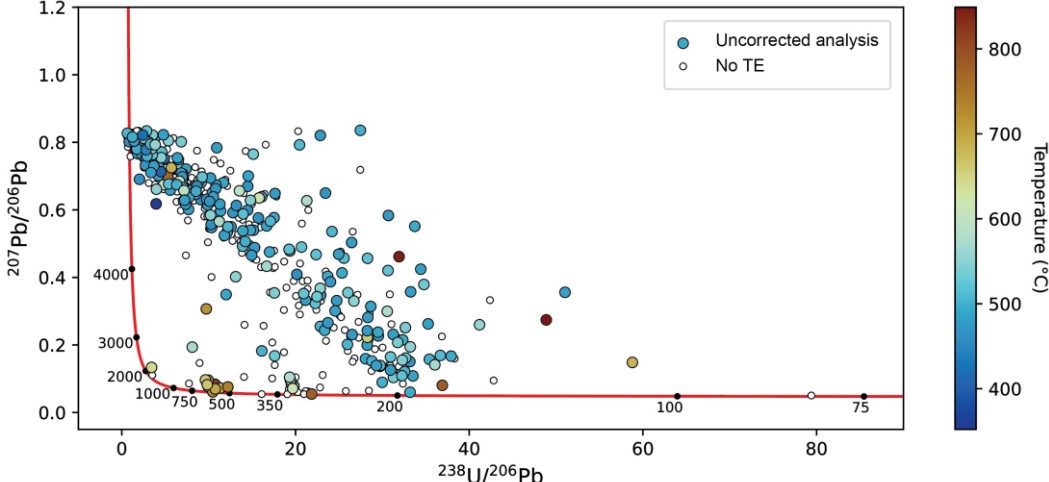

546

*Figure 11. Uncorrected rutile U-Pb results in Tera-Wasserburg space colored by Zr-in-rutile temperature calculated from the Kohn (2020) calibration. The mode centered around 95 Ma has the highest temperatures, and modes centered around 300 Ma and 500–650 Ma also contain high temperatures, whereas the 185 Ma mode is predominantly composed of moderate temperature grains. Open circles are rutile U-Pb analyses without trace element (TE) data. Colormap is from Crameri (2020).*

**6.3 Principal Component Analysis**

Principal component analysis (PCA) was conducted on the detrital rutile trace element compositions (V, Cr, Zn, Zr, Nb, Hf, Ta, W) using an in-house R code (cf. Section 4.4; Mueller, 2024) and the results are given in the data repository. PCA is a multivariate statistical procedure that identifies the variables that explain the most amount of variance within a dataset. The principal components are ranked based on the amount of variance they explain. Plots of principal component 'loadings' display the distribution of the trace element variables with respect to the principal components. The scores and loadings in Figure 12 show that the variance between rutile grains can largely be explained by Cr, Nb and Ta, and W, Zr, and Hf. Because Cr, Nb and Ta are protolith dependent (PC 2) and Hf and Zr are temperature dependent (PC 1), the variance in detrital rutile trace element chemistry is best explained by both protolith and metamorphic grade, tracking these two properties of source rocks. The protolith and temperature components capture the most important portion of the trace element results.

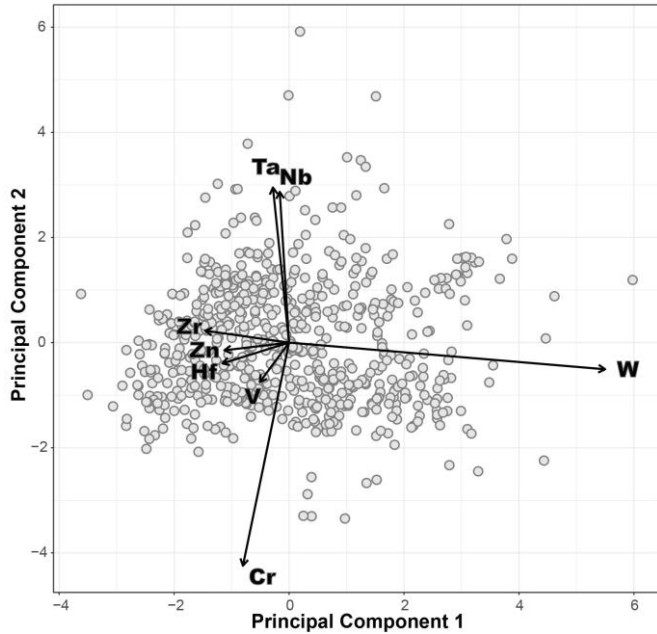

*Figure 12. PCA score and loadings plot of principal components 1 and 2, which cumulatively explain 66.6% of trace element variance. The variance in trace element chemistry is best explained by metamorphic grade (PC 1) and protolith (PC 2).*

**7 Discussion**

**7.1 Recommendations for U-Pb Data Rejection, Correction, and Filtering**

The complex, natural dataset presented here allows an examination of the current practices of data reporting and limitations of large-*n* detrital rutile studies. In this study, a large number of analyses were rejected during U-Pb data reduction, but the SEM images do not provide simple criteria (e.g., inclusions, fractures) how to better select grains that will produce acceptable signal quality or lower U-Pb discordance (Figure 3). All areas selected for analysis appeared inclusion-free before ablation, yet some analyses evidently ablated into inclusions (Figure 3b,e). Because we expected grains from mafic sources with low U or low Pb concentrations, we used a large 50 μm laser beam diameter, but this potentially increased the probability of hitting inclusions. While rejecting analyses is not ideal, low U and Pb signal intensities are not unexpected in natural samples, so some degree of data rejection is to be anticipated, especially given the predicted metamafic (very low U) protolith sources. We contend here that while the exclusion of data from interpretation is common to many detrital rutile studies (e.g., Bracciali et al., 2013; Rösel et al., 2014, 2019; Caracciolo et al., 2021), ours included. However, in most studies, the number of discarded analyses and criteria for discarding analyses during U-Pb data reduction is unclear or not mentioned, thereby limiting opportunities to evaluate data quality and navigate results in a potentially meaningful way. We recommend that these criteria be explicitly stated and discussed in all studies using detrital rutile U-Pb geochronology.

After U-Pb data reduction, additional analyses were excluded during common Pb correction and uncertainty filtering.
Here, the $^{208}$Pb and $^{207}$Pb corrections produce similar date spectra (Figure 5) as do the various uncertainty filters (Figure 6).
We tentatively favor the power law uncertainty filter as it does not appear to alter the presence or proportion of individual age
populations, and because this filter excludes the fewest analyses. Future work is needed to determine if this holds in other
datasets. We propose that the Stacey-Kramers distance is a better metric than Aitchison distance for quantifying discordance
as it reflects U-Pb systematics (Figure 7). A discordance threshold is not recommended as an exclusion criterion based on the
similarity of the date distributions across concordance bins (Figure 8). Further, most mafic-classified grains plot closer to
common Pb compositions, so a discordance filter would bias results toward pelitic and high U grains (Figure 9). Including
initially discordant data is acceptable because geologically meaningful interpretations can be made from initially discordant
data when appropriate common Pb corrections are applied. Note that common $^{208}$Pb and $^{207}$Pb corrections force concordance
so that initially discordant data are concordant after correction. U-Pb discordance in common Pb bearing minerals is well
documented in published reference materials (e.g., Chew et al., 2011, 2014). In petrochronologic applications, *in-situ* work
demonstrates that individual analyses can be nearly 100% discordant and still interpreted confidently within the population of
co-genetic grains (e.g., Poulaki et al., 2023). Although some detrital rutile U-Pb datasets are dominated by concordant analyses
(e.g., Rösel et al., 2019, Kooijman et al. 2010), many detrital datasets contain analyses across the concordance spectrum,
including highly discordant analyses, whose Pb-corrected dates are used in interpretations (Bracciali et al., 2013; Mark et al.,
2016; O'Sullivan et al., 2016; Govin et al., 2018; Ershova et al., 2024). For these reasons, we do not advocate filtering detrital
rutile U-Pb data based on discordance. Future work with large-*n* detrital datasets is needed to explore the influence of common
Pb corrections and data filters based on uncertainty and discordance, including whether these filters influence date distributions
in other datasets.
Expanding detrital rutile U-Pb applications is hindered by data rejection, as seen in this dataset and others. Caracciolo
et al. (2021) attempted to present a large-*n* detrital rutile dataset in which rutile grains were identified via Raman spectroscopy.
Their workflow using automated Raman is better suited for identifying polymorphs and reducing bias than the handpicking
and SEM-EDS workflow used here and in many other studies. However, of the 712 detrital rutile grains analyzed by Caracciolo
et al. (2021), only 347 grains remained (48%) after their data reduction and uncertainty filtering (using a modified power law
filter). Similar to our dataset, there were not enough rutile dates per sample to discuss sample-by-sample provenance
interpretations (Figure S7). Govin et al. (2018) discarded 36% (n=53/146) of detrital rutile U-Pb analyses using their date-
dependent filter. Shaanan et al. (2020) present the only other detrital rutile dataset from Anatolia that does not impose a low-
U filter; they discard 60% (n=97/163) of their data during discordance filtering. Together these studies illustrate that there is a
formidable methodological hurdle in trying to scale up detrital rutile U-Pb to large-*n* provenance applications.

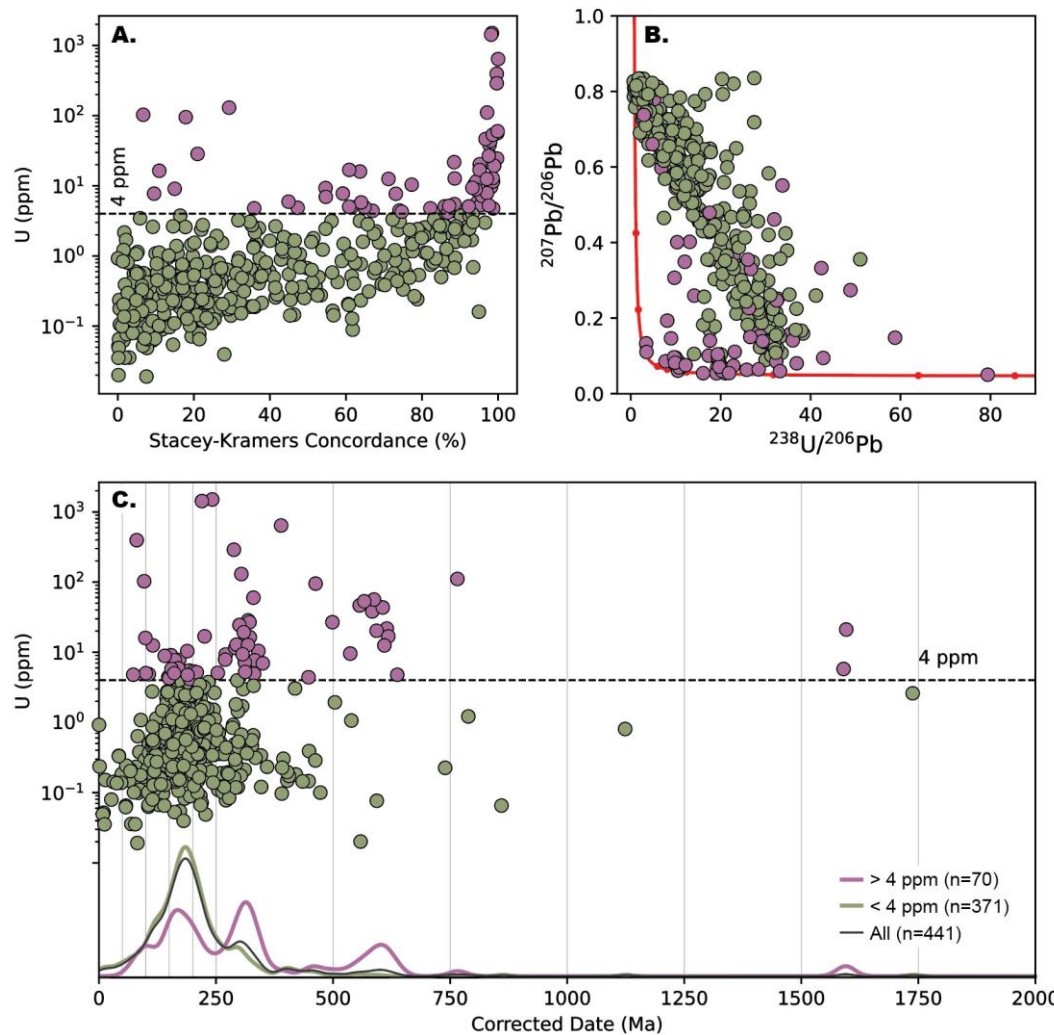

*Figure 13. Comparison of detrital rutile filtering based on U concentration or concordance. (A) Rutile U concentration versus percent concordance (Stacey-Kramers distance). The U-threshold filter groups grains greater than and less than 4 ppm U. (B) Rutile U-Pb results in Tera-Wasserburg space following the color scheme in panel A. (C) Rutile U concentration versus $^{207}Pb$-corrected U-Pb date. The relative KDEs display the date spectra from the different U concentration groups: all analyses, above 4 ppm U, below 4 ppm U. The power law filter is applied to all plots in the figure.*

## 7.2 Low Uranium Rutile

Isotopic and elemental concentrations are calculated based on the measured count rate (i.e., counts per second, CPS), which is inherently dependent on the individual mass spectrometer and laser ablation parameters (e.g., spot size, fluence). For instruments with lower sensitivity (lower CPS per ppm), the same calculated concentration (i.e., the 4 ppm threshold used in some publications) yields lower CPS and therefore higher analytical uncertainties than for instrument with higher sensitivity. In this way, the U threshold filter based on calculated concentration is instrument and parameter dependent and we do not recommend screening rutile to exclude low U concentration analyses.

Most studies no longer impose a U threshold, yet, it is a regional concern in Türkiye where two of the four detrital
rutile U-Pb datasets only analyze U-Pb on detrital rutile with uranium concentrations above 4-5 ppm (Okay et al., 2011; Şengün
et al., 2020). The two studies that do not use a U filter analyze all detrital rutile grains (Shaanan et al., 2020; this study). In this
dataset of this study, 87% of detrital rutile are below 4 ppm U (n=537/612). The majority of detrital rutile with U > 4 ppm are
classified as pelitic and generally have higher Zr contents (higher temperature), whereas low-U rutile in this study generally
correlates with lower Zr contents (lower temperature) and includes the majority of mafic-classified grains (Figure 9). Note that
there are limitations to the Zr-in-rutile thermometer in mafic rocks if the equilibrium conditions are not met. Figure 13
compares U concentration with concordance and U-Pb date. Concordance does not appear to be correlated with U
concentration (Figure 13a). Comparing the date distribution for all grains with that of the below and above 4 ppm U groups
reveals that provenance results would be biased by excluding grains below 4 ppm U (Figure 13c). The above 4 ppm U group
has age modes at 100 Ma, 165 Ma, 315 Ma, 458 Ma, and 600 Ma (Figure 13c pink) whereas the total date spectrum has peaks
at 185 Ma, 300 Ma, 400 Ma, 450 Ma and 600 Ma (Figure 13c gray). The above 4 ppm U rutile group has higher amplitude
Paleozoic peaks, a minor 100 Ma peak, and a younger, lower amplitude Mesozoic peak (165 Ma vs 185 Ma). In summary, the
U threshold filter introduces bias into the provenance results because omitting low-U rutile biases results toward metapelitic
sources, higher Zr-in-rutile temperatures, and shifts the prominent date modes and their amplitudes.

### 7.3 Source Protolith and Metamorphism

The Zr-in-rutile thermometer generally preserves the crystallization or recrystallization temperature. The Zr-in-rutile
thermometer can become uncoupled from the U-Pb age because Pb diffusion during medium- to high-temperature
metamorphic events or extended cooling periods will cause partial or complete resetting of the U-Pb system (Cherniak et al.,
2007; Luvizotto and Zack, 2009; Kooijman et al., 2012; Pereira and Storey, 2023). Because temperatures calculated for the
185 Ma population are cooler than for the older events and are not high enough to have reset the U-Pb dates, we interpret these
temperatures as primary. Furthermore, partially reset dates would smear the data along concordia from the initial crystallization
event age, not towards common Pb.
The Zr-in-rutile temperatures and protolith classification are discussed in the following section in the context of
regional provenance. The PCA results show that the first two principal components are explained by Cr, Nb and Ta, and W,
Zr, and Hf. These elements are protolith (Cr, Nb, Ta) and temperature (Zr, Hf) dependent, therefore the protolith and Zr-in-
rutile sections are already exploring the most salient aspects of the trace element dataset.

### 7.4 Evaluating Bias in Discarded U-Pb Data

To evaluate the potential bias in U-Pb data reduction and processing, the detrital rutile grains with both U-Pb and
trace element data are compared to those with only trace element data (U-Pb rejected and/or excluded by filter). Figure 14
gives a sense for what data are missing from the U-Pb results as well as the effects of the uncertainty filter. Not all detrital
rutile grains have trace element data, so the subset of grains with U-Pb analyses and without trace element data cannot be
considered. In the plots of protolith versus Zr-in-rutile temperature, grains included by the power law filter (Figure 14a) are
compared to those excluded by the power law filter or without U-Pb data (Figure 14b). Effectively this compares accepted U-
Pb analyses to those rejected from unacceptable U-Pb signal patterns or high uncertainties. About 30% of mafic-classified
grains and 35% of pelitic-classified grains are acceptable U-Pb analyses included by the power law filter (Figure 14c). The
analyses rejected by power law filtering (Figure 14c) have a similar temperature distribution, with the majority of temperatures
from 450–550 °C. Most grains with these temperatures fall within the 185 Ma date mode (Figure 11), potentially suggesting
that the detrital rutile grains with poor U-Pb precision would have ~185 Ma dates. Further, the rejected analyses group has
fewer high temperature pelitic grains (> 600 °C) and a more abundant lower temperature pelitic population (< 400 °C). These
temperature windows do not seem diagnostic of specific date populations among pelitic grains, however, about 30% of high
temperature pelitic grains fall within the 500-650 Ma population (Figure 11). The similarity in temperature distributions of
pelitic and mafic grains between the accepted and rejected U-Pb analyses suggests that there is not significant bias in the U-
Pb results due to data rejection. Consequently, the U-Pb and trace element data can be used together to interrogate potential
bias in U-Pb data rejection and filtering.

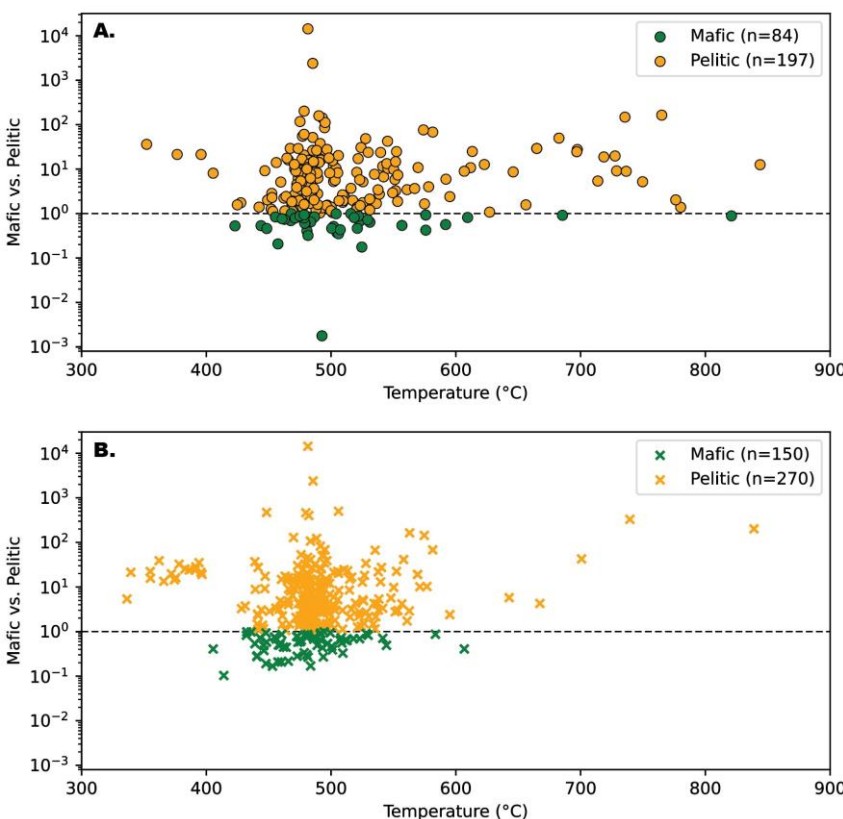


*Figure 14. (A) Protolith versus Zr-in-rutile temperature plot displays all detrital rutile analyses with trace element data*
*included in the power law filter. (B) Plot B shows both the detrital rutile analyses without U-Pb data and those excluded by*
*the power law filter in A. The y-axis values are the transformed distance from the mafic-pelitic discrimination line of Triebold*
*et al. (2012) (Figure 9).*

## 8 Anatolian Sedimentary Provenance

Sedimentary provenance is interpreted from all detrital rutile dates together, rather than by sample, due to the small number of analyses in each sample (see Figure S7 for individual sample results). The detrital rutile results are displayed along with detrital zircon dates from the same Upper Cretaceous to Eocene units in the Central Sakarya and Sarıcakaya Basins (Figure 15; data from Campbell, 2017; Ocakoğlu et al., 2018; Mueller et al., 2019, 2022; Okay and Kylander-Clark, 2022). The detrital zircon and rutile provenance results are discussed together from youngest to oldest date population. The rutile grains that (poorly) define the ca. 90 Ma population (Figure 15) include some of the highest Zr-in-rutile temperatures (Figure 11). The zircon record has abundant Late Cretaceous and Eocene populations (Figure 15) associated with magmatic flare-ups during Alpine orogeny-related subduction and syn-collisional magmatism, respectively (Harris et al., 1994; Kasapoğlu et al., 2016; Yildiz et al., 2015; Ocakoğlu et al., 2018; Mueller et al., 2022; Campbell et al., 2023). The lower plate Anatolide-Tauride terrane underwent HP/LT blueschist facies metamorphism that generally youngs from Late Cretaceous in the north to early Eocene in the south (Sherlock et al., 1999; Okay and Kelley, 1994; Candan et al., 2005; Pourteau et al., 2016). The samples are from sedimentary basins in the upper plate (Figure 15) and the detrital zircon record indicates no sediment transport across the suture zone between from the Anatolide-Tauride terranes to the Pontides in the latest Cretaceous (Okay and Kylander-Clark, 2022). Thus, we interpret the 90 Ma rutile population as either igneous or metamorphic rutile derived from Late Cretaceous magmatism and associated contact metamorphism on the Pontides.

The 185 Ma peak includes the lowest Zr-in-rutile temperatures (~450–550 °C; Figure 11), mafic and pelitic sources (Figure 9), and predominantly low U rutile (Figure 13). The age, lithology, and temperature findings support a Karakaya Complex sediment source. The Permian–Triassic Karakaya Complex contains intra-oceanic basalts and forearc deposits that were metamorphosed to blueschist and epidote-amphibolite facies (340–550 ± 50 °C; Okay et al., 2002; Federici et al., 2010) during the Triassic Cimmerian event. The rutile U-Pb dates interpreted as Karakaya Complex (broad 185 Ma peak) are younger than existing Karakaya Complex phengite, glaucophane, and barroisite Ar-Ar cooling dates (~200–215 Ma: Okay et al., 2002; Federici et al., 2010; Şengör et al., 1984). The closure temperature windows for rutile U-Pb and phengite Ar-Ar overlap, with Pb in rutile extending to lower temperature than Ar in phengitic white mica (Itaya, 2020; M. Grove, pers. comm., 2024). The younger rutile dates likely indicate protracted cooling because extended time spent in the partial retention zone would cause variable Pb loss that could lead to a younger rutile U-Pb dates than any actual heating event and/or a spread in ages (broad peak). This 185 Ma population is not prominent in the detrital zircon spectra. Detrital zircons from Karakaya Complex units have age modes at ca. 235 Ma, 315 Ma, and 400 Ma and are interpreted as sediment input to the forearc from the Pontides Triassic magmatic arc, oceanic plateau, or spreading center (e.g., Okay et al., 2015), Variscan granitoids, and crystalline basement (Ustaömer et al., 2016).

The Carboniferous peaks in the zircon and rutile record correspond to a ~330–340 Ma pulse of high-T metamorphism and ~290–320 Ma magmatism in the Pontides during the Variscan orogeny (Topuz et al., 2007, 2020; Ustaömer et al., 2012, 2013). Variscan-aged detrital rutiles were found in Jurassic sandstones in the Central Sakaraya Basin and interpreted as derived

from either primary Pontide basement or recycled sedimentary sources (Şengün et al., 2020). The Pontide basement units crop
out along the thrust fault that partitions the two sedimentary basins (Tuzaklı-Gümele Thrust; Figure 15b). Therefore, the
Variscan-aged detrital rutile present in Upper Cretaceous to Eocene units could be derived from primary basement sources or
recycled Jurassic sedimentary units. The Pontides crystalline basement contains scarce Devonian (380–400 Ma) and Silurian
(420–440 Ma) metaigneous rocks, which are exposed in the hanging wall of the Tuzaklı-Gümele Thrust (Topuz et al., 2020).
The absence of this age population in the rutile record could be due to the scarcity of outcrops, small sample size, dilution
during sediment recycling, or overprinting by the Carboniferous high temperature event. Late Ordovician–Early Silurian
metamorphism associated with the accretion of the Istanbul–Moesia–Scythian Platform (Okay et al., 2006) is not prominent
in the detrital rutile record, which could suggest the absence of major south-directed sediment transport across the Pontides
(i.e., from the Istanbul Zone to Sakarya Zone across Intra-Pontide ocean/suture) during the Late Cretaceous to Eocene. Lastly,
the 500−650 Ma Pan-African detrital rutile ages align with the detrital zircon age spectra. Gondwana-derived terranes are
characterized by Neoproterozoic–Cambrian plutonism and metamorphism from the Pan-African–Cadomian orogeny, which is
not well documented in Anatolia (Okay et al., 2006). Grains of this age could be sourced from the Pontides basement or
recycled from sedimentary units (Ustaömer et al., 2012; Mueller et al., 2019). However, if the grains of this age were first-
cycle from crystalline basement sources, we would expect them to have reset U-Pb dates from younger metamorphic reheating
events. In this interpretation, the 500–650 Ma dates are preserved because these grains must have been unaffected by any
younger high-T events. In order to have escaped metamorphic reheating, the grains had to have been already eroded from the
crystalline basement and deposited in sedimentary units. Therefore, we interpret the 500–650 Ma grains as polycyclic grains
derived from recycled sedimentary units. Together, the detrital zircon and rutile age spectra demonstrate that, from the Late
Cretaceous to Eocene, sediment was routed to the Central Sakarya and Sarıcakaya Basins from syn-depositional magmatic
centers, the Karakaya Complex within the suture zone, the Pontides crystalline basement, and recycled sedimentary units
(Figure 15).

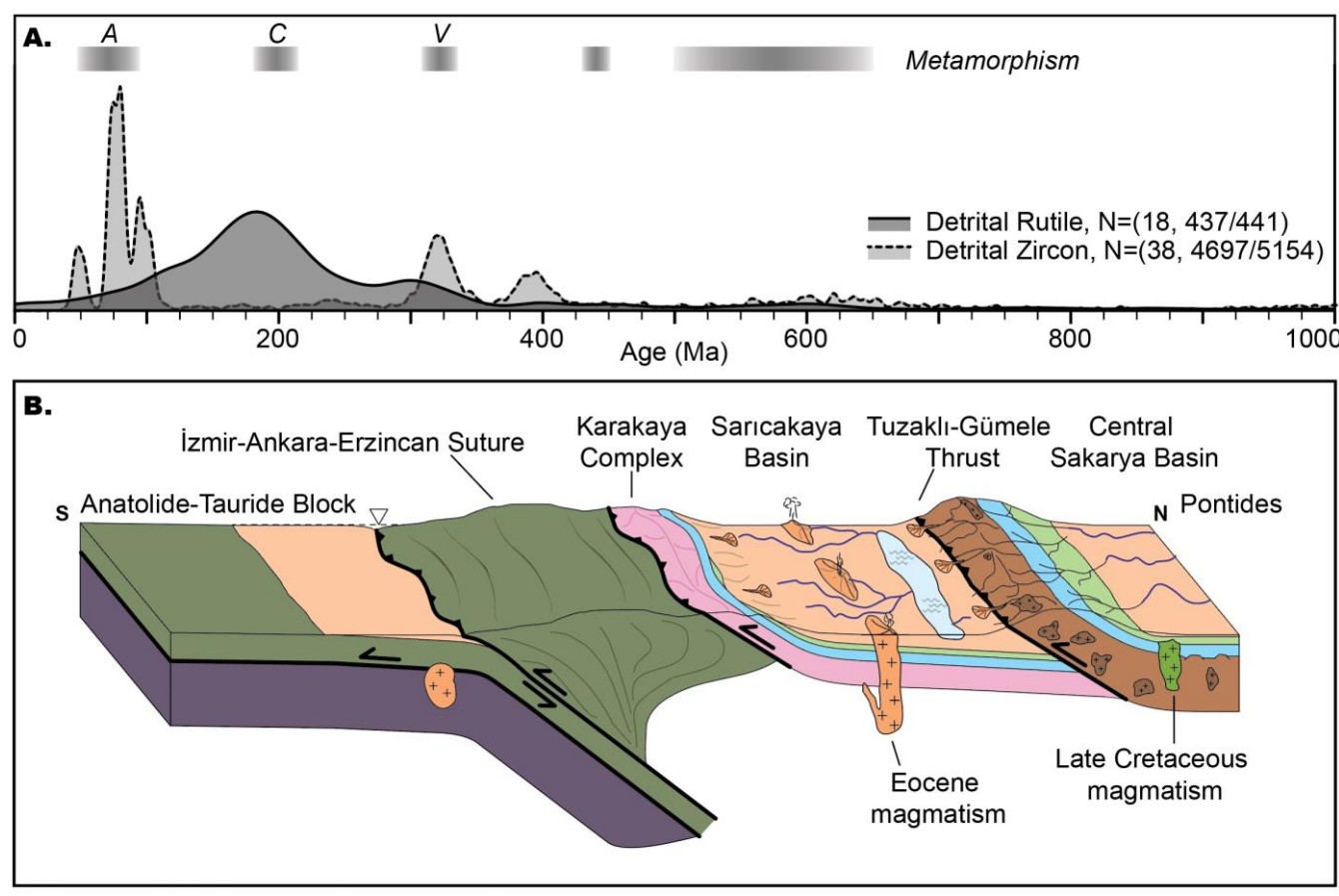

*Figure 15. (A) Kernel density estimate of all detrital rutile dates ($^{207}$Pb-corrected, power law uncertainty filtered) shown alongside a compilation of all published detrital zircon ages from Upper Cretaceous to Eocene strata in Central Sakarya and Sarıcakaya Basins. Gray bars depict periods of metamorphism in western Anatolia. (B) Schematic reconstruction of northwestern Anatolia in the Eocene during continental collision (after Mueller et al., 2019). The main sources of sediment to the basins were the Karakaya Complex exposed in the suture zone, Pontides crystalline basement exposed along the Tuzaklı-Gümele Thrust, Cretaceous-Eocene igneous units, and recycled sedimentary units. A: Alpine metamorphism, C: Cimmerian metamorphism, V: Variscan metamorphism.*

## 9 Conclusions

This work provides a systematic exploration of the data reduction and processing workflows for detrital rutile U-Pb geochronology using a new dataset from the Central Sakarya and Sarıcakaya Basins in Anatolia. Provenance interpretations are made from combining U-Pb dates and trace element geochemistry. The results have several implications for navigating workflows and interpretations in common Pb bearing detrital minerals:

(1) Natural datasets can be complex. While attempting a large-*n* provenance study, a significant number of analyses were discarded due to unacceptable U-Pb signal intensity and stability, namely low U, low Pb, and inclusions. This hurdle is evidently not unique to this dataset and should always be reported in detrital rutile U-Pb geochronology. Advances are needed

to determine the best path forward, such as analyzing more grains for achieving large-*n* detrital rutile U-Pb datasets and more rigorous data reporting and standardizing metrics used for evaluating 'acceptable' U-Pb analyses. We recommend that the criteria for data rejection be explicitly discussed in all detrital rutile studies.

(2) We provide a method for evaluating the potential bias in U-Pb data rejection and filtering by comparing the detrital rutile grains with both U-Pb and trace element data to those with only trace element data. The U-Pb rejected and filtered out grains have a similar trace element distribution in terms of Zr-in-rutile temperature and mafic-pelitic classification to those with acceptable U-Pb analyses, suggesting there is not significant bias from U-Pb data rejection and filtering.

(3) The $^{208}$Pb and $^{207}$Pb correction methods produce similar age spectra and do not change the final provenance interpretations. Similarly, the uncertainty filters—based on U-Pb ratio uncertainty and corrected date uncertainty—produce similar date spectra. The power law uncertainty filter is preferred because it does not alter the date distribution and includes the most grains.

(4) There has not been an agreed upon metric to quantify discordance in common Pb minerals. We evaluate the Stacey-Kramers and Aitchison distance metrics and recommend the Stacey-Kramers distance as a suitable metric for quantifying discordance. However, because reliable interpretations can be made from analyses with significant proportions of common Pb, we recommend not applying a discordance filter to common Pb detrital minerals.

(5) In some labs and geographic locations, only rutile above a certain uranium concentration (i.e., 4-5 ppm U) are analyzed for U-Pb. We demonstrate that excluding low-U rutile biases provenance interpretations toward grains with pelitic classification, higher Zr-in-rutile temperatures, and higher concordance, and changes the overall date distribution, especially the amplitude of date peaks.

(6) A significant challenge in provenance work is pinpointing the signature of sediment recycling. Here we use paired U-Pb dates and Zr-in-rutile temperatures to identify polycyclic detrital rutile grains. The recycled grains preserve U-Pb dates that indicate that they escaped younger metamorphic reheating events of the crystalline basement by already being eroded and deposited in sedimentary units. In this way, detrital rutile petrochronology can address problems of sediment recycling.

(7) The data processing workflows used here are provided as code in Jupyter Notebooks that can be used by future studies. The code includes common Pb corrections, uncertainty filters, discordance calculations, principal component analysis of trace element data, and other trace element plots. The provided code is one path forward to achieving the required documentation and unification of data reduction approaches.

**Data and code availability**

All of the data generated in this manuscript are publicly archived and available in an Open Science Framework data repository that can be accessed at https://doi.org/10.17605/OSF.IO/A4YE5 (Mueller et al., 2023). The data repository also includes the supporting information text. Jupyter Notebooks containing the Python and R code used for data reduction and visualization are open and available at https://zenodo.org/doi/10.5281/zenodo.10636727 (Mueller, 2024).

## Author contributions

MAM conceptualized the project; MAM and AL acquired funding; all authors were involved in the investigation; MAM and AM performed the formal data collection; all authors contributed to writing and revising the manuscript.

## Competing interests

The authors declare that they have no conflict of interest.

## Disclaimer

The software described here is provided under the Apache License, Version 2.0. It is provided "as is," without warranty of any kind, express or implied, including but not limited to the warranties of merchantability, fitness for a particular purpose, and noninfringement. In no event shall the authors or copyright holders be liable for any claim, damages, or other liability, whether in an action of contract, tort, or otherwise, arising from, out of, or in connection with the software or the use or other dealings in the software.

## Acknowledgements

We thank Çelik Ocakoğlu, Jan Westerweel, Kate Huntington, Alison Duvall, Scott Braswell, Joel DesOrmeau, Sean Mulcahy, Scott Dakins, and Eric Steig for support in the field and lab. We thank Andrew Kylander-Clark, Francisco Apen, and Peter Downes for reference materials and Stuart Thomson, Margo Odlum, Eirini Poulaki, and Drew Levy for discussions on common Pb corrections. We thank Associate Editor Pieter Vermeesch and referees David Chew, Laura Bracciali and Ines Pereira for thoughtful reviews that improved the manuscript. We thank the *iolite* team for student access.

## Financial support

This work was funded by the University of Washington Department of Earth and Space Sciences and NSF EAR-1543684 and EAR-2141115.

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
