# Peer review of "Navigating the complexity of detrital rutile provenance"

_EGUsphere, 2023_

## Referee Comment (RC2)

[referee-annotated manuscript omitted]

---

## Referee Comment (RC3)

Dear Pieter,

Please see below by review of Mueller et al. – "An expanded workflow for detrital rutile provenance studies: An application from the Neotethys Orogen in Anatolia".

This is a nice dataset which is used as a case study to argue for a new/expanded workflow in U-Pb detrital rutile provenance studies.

There are several issues with the paper as is:

The use of the phrases "expanded workflow" (title) or "new workflow" (section 7.2 heading). This revised workflow appears to be mainly not applying U concentration thresholding in an initial trace element session. The majority of labs nowadays (as far as I know) are not doing this, so that is not new. It is shown that it is inappropriate, but if it is only undertaken by a small subset of labs then is it all that important ? it certainly doesn't warrant inclusion in the title. e.g. in L26 "We present a new workflow that accounts for low-U rutile…" - I can show you lots of published papers that date all the rutile in the rock and do not undertake U thresholding, five from my lab alone extending back to 2019.

ii) The choice of common Pb composition – it is interesting to explore the difference between the 207Pb and 208Pb methods, but they ultimately do not show much of a difference.  That is new, but maybe not that significant a result.  But I like the general approach to discordance filtering.

iii) the section on the choice of initial age estimate to stick into the 207Pb correction (uncorrected age [t_initial] versus the 208Pb corrected age [t_208]) is really confusing.  I am really puzzled by the large difference between the two approaches for discordant data (Fig. 6). In a 2011 Chemical Geology paper I showed that the final 207Pb-corrected age differs by < 0.05% if an initial age estimate of 1 Ma is used instead of 1 Ga, demonstrating it is not dependent on the choice of initial age after five iterations.  As far as I can see after five iterations in a 207Pb correction, you have converged on the answer, regardless of the starting age estimate. So I cannot explain Fig. 6 unless only one iteration of the correction has been undertaken? If that is indeed the case (only one iteration of the correction has been done), then that entire section should be removed as the process has not yet converged on a solution.

iv) the PCA approach to exploring the data is interesting, but currently not much is made of it.

I recommend major revisions, with suggestions for improvements including:
   a) scaling back on the strong statements about new workflows etc. – because there are papers out there already which identify all the rutiles on a mount (SEM-EDS or Raman), analyse all grains for U-Pb and trace elements, including Cr vs Nb discrimination and Zr-in-rutile temperatures.
   b) Keep the U threshold aspect in, but shorten significantly and do not make it a key aspect of the paper as I do not think it is all that common an approach nowadays

c) Clarify the choice of initial age estimate to stick into the 207Pb correction – if an iterative approach has not been used (with at least five iterations) then I am not sure why it is included. But keep the bit on discordance filtering.

d) Make more of the PCA plot, and also of your own data.

Best wishes,

David Chew

Comments working through paper:

First 8 lines of abstract. I really think there needs to be a caveat here. Very broadly speaking (and there are exceptions), rutile is not particularly common in igneous rocks in the crust and requires reasonably high pressures to crystalize – it is mainly a metamorphic mineral (where again it requires reasonably high pressures to crystallize). It is better than zircon in recording metamorphic events in provenance studies but it too has a relatively restricted paragenesis.

L46 – I am not sure why this sentence starts with "In convergent margin settings….", as I feel it is applicable to other tectonic settings as well.

L64 – and detrital apatite. In terms of recent publications (i.e. last 5 years) I feel it is more commonly used nowadays than either detrital monazite and detrital muscovite. Please also list the geochronological system applying to the mineral – U-Pb, 40Ar/39Ar, Luf-Hf etc

Section 2.1 This section needs something on the role of pressure and composition on the stability field of igneous and metamorphic rutile, and also the well documented instability instability in the sub-greenschist to lower greenschist facies (Zack et al., 2004; Yakymchuk et al., 2017).

L124 (Challenge 1). I dispute the sentence "many detrital rutile methods first analyse trace elements then only collect U-Pb data on rutile above a given U concentration threshold (4-5 ppm).". I have reviewed quite a few studies in the last few years with detrital rutile U-Pb and trace element data in them, and I have never (as far as I remember) encountered this approach. I can see why it may have been applied historically (maybe over a decade ago), where quadrupole-ICP-MS or a slow-scanning sector field MS was used to give the trace elements and U-Pb was subsequently analysed by sector field ICP-MS. But let us talk about the last few years (i.e. what is currently happening). The amount of labs doing this now I feel is very small. A modern quadrupole such as an iCAP or Agilent 7900 can easily produce all the necessary TEs and good U-Pb data simultaneously in the same spot ablation. It may appear that I am making a big deal of this - but then L125-130 then make a big deal of this. I strongly agree it would introduce a bias and this is shown later on. But I feel that such an approach is hardly ever used nowadays and so the authors are arguing against a false premise as a rationale for this paper. I feel challenge #1 needs rewriting and the screening part removed, or convincing demonstration it is still a common approach (e.g. look at all detrital rutile studies published in the last five years and find the % that did U thresholding).

I feel this is entirely restricted to sector field labs (a subset of all data produced) and only a subset of those studies would in turn screen by U thresholding.

L184 I am confused here. "We explore using an initial date estimate from the uncorrected date (ti) and from the 208Pb-corrected date (t208)." How many iterations are you using after this initial age estimate? It doesn't really matter what the age estimate is if it eventually converges on a solution? That is what is important. Unlike for the 208Pb correction you do not specify the amount of iterations after this initial age estimate?

L188 "Note that because the correction forces intersection with the concordia, the two dates are identical". I wouldn't mention this at all – you have only one age when doing a 207Pb correction - you report a 207Pb-corrected date.

L190 I would like to see more about the choice of the Pb initial and whether it is appropriate to use the Stacey and Kramers (1975) model. It is well known that the 207Pb/206Pb initial ratio of metamorphic titanite is often significantly lower (i.e. more radiogenic) than the Stacey and Kramers (1975) crustal evolution model, reflecting incorporation of radiogenic Pb from rutile, a common titanite precursor (see Essex and Gromet, 2000). But rutile replacing titanite is also seen in bedrock samples  - have a look at Gumsley et al. (2023, Lithos). In their Figs 11a and 11b you have metamorphic rutile with a 207Pb/206Pb initial with 0.10 -0.12, which can be convincingly linked to breakdown of late Variscan titianite

L197 presumably this principle about minimum ages also applies to 208Pb corrected data?

L200 How many iterations are used following this initial age estimate.  Five was quoted for the 208Pb correction, but the number of iterations is not quoted for the 207Pb correction, and it urgently needs to be.  I found in Chew et al. (2011) that it was generally insensitive to the choice of the initial age estimate input into Stacey & Kramers after a few iterations.

Section 2.3 Why are 204Pb corrections not discussed?

L216. Please also provide pressure estimates.  The pressure dependence on the stability of rutile is not getting much attention in this manuscript.

L281. Significant error here – S&K at 1000 Ma is about 0.909?

Section 5.1 I found this section really hard to assess when it came to the 207Pb correction using a starting estimate of t_initial or t_208, as the amount of iterations in the 207Pb correction calculation (as far as I could see) was not explicitly specified earlier. I would be somewhat surprised to see any significant variation after a few iterations (say five).  It doesn't matter if there is a difference after one iteration – what matters is the variation after the iterative process has been completed.  For example, there is a surprising large age difference in Fig 6 for the low concordance grains between an initial age estimate using t_Initial vs an initial age estimate of t_208. If this is after five iterations, then that is a noteworthy result. If it is after one iteration, then it is in my opinion of no significance as you have yet to converge on the solution. Hence I am not sure if the starting age estimate issue is all that important and could be removed (e.g. if Fig.6 is based on one iteration), but

it is hard to assess without more information.  I found Fig. 6 pretty confusing to be honest and I think the figure cpation needs more information as I am not entirely sure what was being plotted.

L299-L300 "However, the similarity in the 207Pb with t208 cumulative date distribution for the 100–40% and 40–0% groups is notable,". I really didn't understand this clause – sorry.

L306-308. Exactly how common is this approach nowadays? To the best of my knowledge I have never reviewed a detrital rutile U-Pb paper that does this. I think nowadays it is a fairly (or even very) uncommon approach. For this reason alone, I am not sure section 5.2 it is worth including in the manuscript, certainly not in so much detail.

Figure 8 – not clear which lines defining fields belong to which paper (Triebold vs Meinhold) on the Cr/Nb plot without reading the text – label them.

Section 6.3 – PCA
Not quite sure what the main point this paragraph is trying to tell us – it could be expanded on. It does show the Cr vs Nb plot is useful in that Cr (+ V which has similar behaviour) pulls in an opposite direction to Nb (+ Ta which has similar geochemical behaviour).  So the Cr vs Nb plot does a good job of separating the fields. If you were to crudely put on mafic vs pelitic fields on the PCA plot (boundary between yellow vs green), then the Hf + Zr vectors would be roughly parallel, showing the mafic vs pelitic distinction is somewhat independent of temperature. But the plot is introduced without significant additional interpretation.

Sections 7.2
Some points below link back to substantive points made at the start of the review:
"the various Pb correction methods produce similar age spectra and do not change the final provenance interpretations" – so maybe that section should be scaled a bit as ultimately it does not appear to be that important.

"the 190 Ma population is poorly represented in the detrital zircon record" - but the counterpoint needs to be made that the 90 Ma population is very important in the detrital zircon record and not in the rutile record.

What exactly is the new workflow - not doing a U-threshold and analysing all grains including those identified by SEM-EDS?  There are lots of studies already doing that – I cannot see the justification for "New workflow" in the abstract text or in the heading for section 7.2. For example, Caracciolo et al. (2022) present a large U-Pb detrital rutile (n =712) dataset (along with zircon and apatite), where all rutile grains in the heavy mineral fraction determined by Raman were analysed for U-Pb and trace elements (including Cr/Nb discrimination and Zr-in-rutile temperatures).  I do not think the phrase "new workflow" is justified.

---

## Referee Comment (RC4)

[referee-annotated manuscript omitted]

---

## Author Comment (AC1)

Response to Reviewer 1's comments on manuscript egusphere-2023-1293

"An expanded workflow for detrital rutile provenance studies: An application from the Neotethys Orogen in Anatolia"

Megan A. Mueller, Alexis Licht, Andreas Möller, Cailey B. Condit, Julie C. Fosdick, Faruk Ocakoğlu, and Clay Campbell

November 15, 2023

We appreciate the thoughtful reviews of the 3 referees. First, we summarize the broad themes from the three reviews before responding in detail to Reviewer #1 below. The reviews critiqued (1) the novelty of the study, (2) the number of U-Pb analyses discarded during data reduction, (3) the potential bias of discarding data and the validity of interpreting discordant U-Pb analyses, and (4) the apparent lack of novelty and complexity in geochemical data interpretations. Regarding these points, the goal of the manuscript was to provide a transparent workflow for detrital rutile geochronology using new data from sedimentary basins in Anatolia. We acknowledge that it was confusing or misleading to present this work as a revised workflow. We intended to emphasize that we found a paucity of methods papers that provide a straightforward approach. Although we did not claim to present new workflows with the geochemical data, this was perhaps unintentionally implied with the title. We have presented as much information as we could squeeze out of our particular geochemical dataset. The revised manuscript will scale back statements on new workflows and instead refocus the title and introduction on suture zone settings and Anatolian geology while maintaining the broad overview of detrital rutile provenance and the details of our methods.

We also acknowledge that it is surprising to see the number of data discarded during data reduction. However, we contend that this is a common practice with detrital U-Pb geochronology in common Pb-bearing minerals. We are not the first to discard a significant number of analyses; we have found this practice in many published detrital rutile datasets, although it is not discussed much in the literature. We further expand on this point in our detailed reply and will add this context to the revised manuscript. This manuscript gives precedence for papers to be transparent in data reporting—including the number of grains analyzed and criteria for rejection—as well as examination of the full dataset in Tera-Wasserburg diagrams. Our manuscript provides an opportunity to show the consequence and potential value of the full dataset, which is relevant to others working with this type of data. While the use of common Pb-bearing minerals is common in some labs and geographic settings, the application of these tools is still far behind detrital zircon geochronology, for which there is a well-established global framework. We have encountered many detrital geochronology users who want to add detrital rutile to their toolkits but are still uncertain in how to collect and interpret these data. Here, we present a complicated detrital rutile U-Pb dataset that can serve as an example for how to treat and interpret complex, yet potentially meaningful, discordant data.

Reviewer #1 provided a thoughtful review of our manuscript that highlighted several ways to improve the manuscript that include scaling back statements on 'new workflows' and clarifying the new aspects of the data workflow, clarifying common Pb correction calculations, and expanding the analysis of the detrital rutile trace element geochemistry. The review includes several main suggestions for improvement that we will follow in the revised manuscript. We thank Reviewer #1 for their helpful suggestions, which we plan to incorporate into the revised manuscript.

Reviewer #1's comments are included below in black text, grouped by theme. Below we state how we will implement Reviewer #1's suggestions in a future revision, with our response in purple text and the specific changes highlighted in *bold, italic purple text*.

1. **Comments regarding the data workflow and low-U threshold filtering**

   - Suggestions for improvements including: (a) scaling back on the strong statements about new workflows etc. – because there are papers out there already which identify all the rutiles on a mount (SEM-EDS or Raman), analyse all grains for U-Pb and trace elements, including Cr vs Nb discrimination and Zr-in-rutile temperatures. (b) Keep the U threshold aspect in, but shorten significantly and do not make it a key aspect of the paper as I do not think it is all that common an approach nowadays

   - The use of the phrases "expanded workflow" (title) or "new workflow" (section 7.2 heading). This revised workflow appears to be mainly not applying U concentration thresholding in an initial trace element session. The majority of labs nowadays (as far as I know) are not doing this, so that is not new. It is shown that it is inappropriate, but if it is only undertaken by a small subset of labs then is it all that important ? it certainly doesn't warrant inclusion in the title. e.g. in L26 "We present a new workflow that accounts for low- U rutile..." - I can show you lots of published papers that date all the rutile in the rock and do not undertake U thresholding, five from my lab alone extending back to 2019.

   - L124 (Challenge 1). I dispute the sentence "many detrital rutile methods first analyse trace elements then only collect U-Pb data on rutile above a given U concentration threshold (4-5 ppm).". I have reviewed quite a few studies in the last few years with detrital rutile U-Pb and trace element data in them, and I have never (as far as I remember) encountered this approach. I can see why it may have been applied historically (maybe over a decade ago), where quadrupole-ICP-MS or a slow-scanning sector field MS was used to give the trace elements and U-Pb was subsequently analysed by sector field ICP-MS. But let us talk about the last few years (i.e. what is currently happening). The amount of labs doing this now I feel is very small. A modern quadrupole such as an iCAP or Agilent 7900 can easily produce all the necessary TEs and good U-Pb data simultaneously in the same spot ablation. It may appear that I am making a big deal of this - but then L125-130 then make a big deal of this. I strongly agree it would introduce a bias and this is shown later on. But I feel that such an approach is hardly ever used nowadays and so the authors are arguing against a false premise as a rationale for this paper. I feel challenge #1 needs rewriting and the screening part removed, or convincing demonstration it is still a common approach (e.g. look at all detrital rutile studies published in the last five years and find the % that did U thresholding). I feel this is entirely restricted to sector field labs (a subset of all data produced) and only a subset of those studies would in turn screen by U thresholding.

   - L306-308. Exactly how common is this approach nowadays? To the best of my knowledge I have never reviewed a detrital rutile U-Pb paper that does this. I think nowadays it is a fairly (or even very) uncommon approach. For this reason alone, I am not sure section 5.2 it is worth including in the manuscript, certainly not in so much detail.

   - What exactly is the new workflow - not doing a U-threshold and analysing all grains including those identified by SEM-EDS? There are lots of studies already doing that – I cannot see the justification for "New workflow" in the abstract text or in the heading for section 7.2. For example, Caracciolo et al. (2022) present a large U-Pb detrital rutile (n =712) dataset (along with zircon and apatite), where all rutile grains in the heavy mineral fraction determined by Raman were analysed for U-Pb and trace elements (including Cr/Nb discrimination and Zr-in-rutile temperatures). I do not think the phrase "new workflow" is justified.

   The workflow that we present includes two elements: evaluating the importance of including low-U rutile grains in provenance analysis and considering the effects of U-Pb discordance on provenance interpretations. We agree with Reviewer #1 that most labs that analyze detrital rutile do not apply a U-

threshold filter. While not a global problem, this is a regional problem. There are 4 published detrital rutile U-Pb datasets from Türkiye (including this study), and 2 of them (Okay et al., 2011; Şengün et al., 2020) only analyze U-Pb on grains with uranium concentrations above ca. 4-5 ppm. This is a regional problem and imparts a bias, which we wanted to address in this manuscript as detrital rutile analysis is still an uncommon tool for Anatolia. The 2 studies that do not use a U-threshold filter but instead analyze all detrital rutile grains (Shaanan et al., 2020; this study) have to discard data due to very low uranium signals and must implement a protocol for evaluating discordance because of common Pb incorporation. For example, Shaanan et al. (2020) discard 60% of their detrital rutile U-Pb data due to discordance. Similarly, Reviewer #1 points to the study by Caracciolo and co-authors (2021) that analyzes 712 detrital rutile grains without a U-filter, yet, after discordance filtering, only 347 grains remained (48%) (however we have not been able to examine the data as it is not available online or from the journal or lead author). We agree that automated Raman or automated mineralogy are better suited for identifying polymorphs than handpicking and/or SEM-EDS. Importantly, in the study of Caracciolo and others, there were not enough rutile ages per sample to discuss sample-by-sample provenance interpretations, which we also experienced with our dataset. This points to a larger problem in trying to scale up detrital rutile to large-*n* provenance applications. For this reason, we wanted to confidently include as many U-Pb analyses as possible in our interpretations, which led to the exploration of U-Pb discordance.

To address this concern, ***we will change the text to reduce the discussion of U-threshold filtering. The revised manuscript will clarify that U-threshold filtering is currently not a common practice but is used regionally in Anatolia.***

***Following the comments of Reviewers #1 and #3, we will move away from phrases like 'new workflow.' The revised manuscript will have an updated title and introduction that is oriented toward Türkiye and suture zone settings.***

**2. Comments regarding common Pb corrections**

- Suggestions for improvements including: (c) Clarify the choice of initial age estimate to stick into the 207Pb correction – if an iterative approach has not been used (with at least five iterations) then I am not sure why it is included. But keep the bit on discordance filtering.

- L184 I am confused here. "We explore using an initial date estimate from the uncorrected date (ti) and from the 208Pb-corrected date (t208)." How many iterations are you using after this initial age estimate? It doesn't really matter what the age estimate is if it eventually converges on a solution? That is what is important. Unlike for the 208Pb correction you do not specify the amount of iterations after this initial age estimate?

- L200 How many iterations are used following this initial age estimate. Five was quoted for the 208Pb correction, but the number of iterations is not quoted for the 207Pb correction, and it urgently needs to be. I found in Chew et al. (2011) that it was generally insensitive to the choice of the initial age estimate input into Stacey & Kramers after a few iterations.

- the section on the choice of initial age estimate to stick into the 207Pb correction (uncorrected age [t_initial] versus the 208Pb corrected age [t_208]) is really confusing. I am really puzzled by the large difference between the two approaches for discordant data (Fig. 6). In a 2011 Chemical Geology paper I showed that the final 207Pb-corrected age differs by < 0.05% if an initial age estimate of 1 Ma is used instead of 1 Ga, demonstrating it is not dependent on the choice of initial age after five iterations. As far as I can see after five iterations in a 207Pb correction, you have converged on the answer, regardless of the starting age estimate. So I cannot explain Fig. 6 unless only one iteration of the correction has been undertaken? If that is indeed the case (only one iteration of the correction has been done), then that entire section should be removed as the process has not yet converged on a solution.

- Section 5.1 I found this section really hard to assess when it came to the 207Pb correction using a starting estimate of t_initial or t_208, as the amount of iterations in the 207Pb correction calculation (as far as I could see) was not explicitly specified earlier. I would be somewhat surprised to see any significant variation after a few iterations (say five). It doesn't matter if there is a difference after one iteration – what matters is the variation after the iterative process has been completed. For example, there is a surprising large age difference in Fig 6 for the low concordance grains between an initial age estimate using t_Initial vs an initial age estimate of t_208. If this is after five iterations, then that is a noteworthy result. If it is after one iteration, then it is in my opinion of no significance as you have yet to converge on the solution. Hence I am not sure if the starting age estimate issue is all that important and could be removed (e.g. if Fig.6 is based on one iteration), but it is hard to assess without more information. I found Fig. 6 pretty confusing to be honest and I think the figure cpation needs more information as I am not entirely sure what was being plotted.

The manuscript did not include an iterative approach for the [207]Pb correction. We understand that an iterative approach is recommended (Chew et al., 2011; Thomson et al., 2012; Smye and Stockli, 2014). Below, we display preliminary results from a [207]Pb correction with 5 iterations as compared with the [207]Pb- and [208]Pb-corrected dates from the original manuscript (Figure R1). The preliminary iteration 5 [207]Pb-corrected date spectrum is similar to the [208]Pb corrected spectrum. *The revised manuscript will calculate [207]Pb-corrected ages using an iterative approach with at least 5 iterations. The text and equations explaining the iterative process will be updated (Section 2.3.2). All figures and tables will be updated. Furthermore, interpretations will be updated if there are significant changes in the resulting age spectra (Section 7.1).*

Thank you for the feedback that Section 5.1 is unclear. It will be important to clarify this section as it is the basis for the choice of discordance cutoff. Figure 6 shows the differences between $^{207}Pb_{ti}$- and $^{207}Pb_{t208}$-corrected ages versus the percent concordance. *The revised manuscript will remove the $^{207}Pb_{t208}$ correction as this approach is irrelevant with an iterative $^{207}Pb$ correction. After updating the $^{207}Pb$ corrected ages, we will explore the difference in $^{207}Pb$- and $^{208}Pb$-corrected age in the revised manuscript. The figure captions and relevant text will be updated to better explain this calculation.* In the original manuscript, the ages differed significantly (> 5%) for analyses below 40% concordant, the justification for our 40% concordance filter. The differences in age cannot be explained by grain age or Th concentration. *As needed, the revised manuscript will revisit the concordance cutoff based on the updated $^{207}Pb$-corrected ages, and all text, figures, and tables will be revised. If the significant difference in age remains, we will investigate the potential controls.*

[Figure]

*Figure R1: A preliminary comparison of $^{207}$Pb-corrected dates with 5 iterations (top), $^{207}$Pb-corrected dates in the original manuscript (middle), and $^{208}$Pb-corrected dates with 5 iterations in the original manuscript (bottom). All ages 0 to 1000 Ma that are 100-40% concordant are displayed.*

- The choice of common Pb composition – it is interesting to explore the difference between the 207Pb and 208Pb methods, but they ultimately do not show much of a difference. That is new, but maybe not that significant a result. But I like the general approach to discordance filtering.

- "the various Pb correction methods produce similar age spectra and do not change the final provenance interpretations" – so maybe that section should be scaled a bit as ultimately it does not appear to be that important.

- L190 I would like to see more about the choice of the Pb initial and whether it is appropriate to use the Stacey and Kramers (1975) model. It is well known that the 207Pb/206Pb initial ratio of metamorphic titanite is often significantly lower (i.e. more radiogenic) than the Stacey and Kramers (1975) crustal evolution model, reflecting incorporation of radiogenic Pb from rutile, a

common titanite precursor (see Essex and Gromet, 2000). But rutile replacing titanite is also seen in bedrock samples – have a look at Gumsley et al. (2023, Lithos). In their Figs 11a and 11b you have metamorphic rutile with a 207Pb/206Pb initial with 0.10 -0.12, which can be convincingly linked to breakdown of late Variscan titanite

We chose an initial $^{207/206}$Pb value based on the $^{206}$Pb/$^{238}$U age calculated from the uncorrected $^{206}$Pb/$^{238}$U and $^{207/206}$Pb ratios. However, after 5 iterations, the resulting $^{208}$Pb age is invariant to the choice of initial age and common Pb. To address the review, we checked this by varying the initial age estimate, and therefore the initial common Pb composition, from 1 Ma to 1000 Ma and the resulting $^{208}$Pb-corrected age differs by less than 0.05% for 98% of our unknowns (578 of 592 analyses). As stated by Reviewer #1, Chew et al. (2011) demonstrated a similar result for $^{207}$Pb-corrected ages: the choice of initial age results in a < 0.05% difference in the final $^{207}$Pb-corrected age after 5 iterations. ***The revised manuscript will include an iterative approach to the $^{207}$Pb correction. We expect that the final iteration of the $^{207}$Pb age will be insensitive to the choice of initial common Pb composition. Further, we do not expect significant changes in the resulting age spectra (Figure R1) and do not anticipate major changes to the provenance interpretations. The revised manuscript will clarify the choice of the Pb initial and explain that the resulting Pb-corrected ages are insensitive to this choice, including citation of prior work. The supplementary data files will be updated. Additionally, we will assess the similarities between the resulting $^{207}$Pb- and $^{208}$Pb-corrected ages. Preliminarily, the 208Pb correction appears to differentiate more peaks (Figure R1), which will be explored further in the revised manuscript. We will follow the reviewer's suggestion to scale back our emphasis on Section 5.1.***

This is a good point about whether the Stacey and Kramers (1975) values are appropriate to use as estimates of Pb initial. We agree that it is possible that the $^{207}$Pb/$^{206}$Pb initial for the detrital rutile grains may differ from the Stacey and Kramers (1975) model. As noted in other studies, there are no constraints on the initial Pb composition in detrital samples (e.g., Chew et al., 2011). We attempted to address this in the manuscript:

> *"Most rutile U-Pb dates are expected to be discordant. In-situ studies mitigate this by: (1) regressing discordia lines through co-genetic analyses in Tera-Wasserburg space, where the lower intercept of the discordia with the concordia defines the U-Pb age of Pb diffusion closure (Faure, 1986; Chew et al., 2011; Vermeesch, 2020); or (2) applying a non-radiogenic Pb correction using either an ad hoc Pb model such as that of Stacey and Kramers (1975), or measuring the composition of non-radiogenic Pb in a co-existing phase. However, by nature, the co-genetic grains in detrital samples are unknown." (Lines 142-147)*

We emphasized that, for detrital grains, there are no constraints on the composition of non-radiogenic Pb. This is unlike *in-situ* work where the common Pb composition can be determined by analyzing co-genetic grains or co-genetic, U-free phases (i.e., K-feldspar). In a detrital sample, it is uncertain whether two detrital rutile grains are co-genetic, or if a detrital rutile and a detrital K-feldspar are co-genetic, for example. Therefore, common Pb corrections based in the Stacey and Kramers (1975) Pb evolution model are dominant in the detrital rutile literature (e.g., Thomson et al., 2012; Caracciolo et al., 2021; Odlum et al., 2019; Chew et al., 2020; Najman et al., 2019; Clift et al., 2022; Mark et al., 2016).

As a thought exercise, we take a closer look at one sample for which the grains might be considered co-genetic. Sample 18DMN01 had only 4 rutile grains and the U-Pb results seem to plot in a linear array, suggesting that they may represent one age population from the same source (Figure R2). After 5 iterations, the $^{207/206}$Pb$_c$ initial values used range from 0.8416 to 0.8418, and the $^{207}$Pb-corrected ages range from 92 Ma to 96 Ma. On the other hand, if we assume that the 4 grains from sample 18DMN01 represent one age population (i.e., are co-genetic), we can regress a discordia line through the analyses to assess the initial common Pb value. In this case, the 4 grains from 18DMN01 give a discordia age of 91.5 ± 3.8 Ma and $^{207/206}$Pb$_c$ of 0.802 ± 0.068 (Figure R3). This could suggest that the Stacey and Kramers (1975) values are not the best estimate of initial Pb composition. In this example, the resulting

difference between an age of 91.5 ± 3.8 Ma or 92-96 Ma is within the uncertainty and does not alter the final provenance interpretation.

This approach is difficult to put into practice in provenance studies because it is inherently unclear which analyses should be grouped together (i.e., treated as co-genetic). This problem is well documented in the literature. Should a range of U-Pb ratios be treated as (1) a single age population from one source, (2) a range of ages from one source, or (3) a range of ages from multiple sources? If (1), then analyses should be grouped together and the Pb correction can be performed without an estimate of initial $^{207/206}$Pb (Figure R3). Yet, if it is (2) or (3), it is unclear which analyses should be grouped together and treated as co-genetic. Trace element discrimination may be needed to aid these decisions and provide a way forward.

All of this is to say, we acknowledge that the Stacey and Kramers (1975) model may not be the most accurate initial $^{207/206}$Pb$_c$ value. Yet, we are unaware of a better method for addressing this problem in detrital studies and follow in the footsteps of many studies that have applied $^{208}$Pb and $^{207}$Pb corrections in detrital minerals. There may be other geographic settings where the Pbc composition of sources is well characterized, such that a more appropriate $^{207/206}$Pb$_c$ value may be used for each age population. ***In the revised manuscript, we will add a few lines that explain that the Stacey and Kramers (1975) values may not be the correct common Pb composition but are still an appropriate initial estimate for performing iterative common Pb corrections in detrital grains.***

[Figure]

*Figure R2: Tera-Wasserburg diagram of all detrital rutile U-Pb results. Analyses highlighted in yellow are from sample 18DMN01 and yield $^{207}$Pb-corrected ages from 92 Ma to 96 Ma (after 5 iterations; denoted by yellow circles at intersection with concordia). The common Pb composition ranges from 0.8416 to 0.8418 for these 4 analyses. The black lines show the discordia lines fit between initial Pb$_c$ and each analysis, and their lower intersection with the concordia is the $^{207}$Pb-corrected age.*

[Figure]

*Figure R3: Tera-Wasserburg diagram of analyses from sample 18DMN01. The unanchored discordia age is 91.5 ± 3.8 Ma and the common Pb composition is 0.802 ± 0.068. Figure from IsoplotR (Vermeesch, 2018).*

- L188 "Note that because the correction forces intersection with the concordia, the two dates are identical". I wouldn't mention this at all – you have only one age when doing a 207Pb correction – you report a 207Pb-corrected date.

  Thank you. *We will exclude this in the revised manuscript.*

- L197 presumably this principle about minimum ages also applies to 208Pb corrected data?

  Yes, *we will add this sentence to the 208Pb corrected age section* (Section 2.3.1).

- Section 2.3 Why are 204Pb corrections not discussed?

  Reviewers #1 and #3 both inquired why the $^{204}$Pb correction was not discussed. This was not initially included because (a) $^{204}$(Pb+Hg) and $^{202}$Hg were not measured, so we did not perform a $^{204}$Pb correction (Because of the high Hg background at KU IGL, a $^{204}$Pb correction would be too imprecise. All commercially available UHP He gas options in the midcontinent US have high Hg), and (b) it is reviewed in other publications. Although we will not be able to explore the $^{204}$Pb correction, which is a goal of future work, *the revised manuscript will include a short overview of this correction and its application in rutile.*

- L281. Significant error here – S&K at 1000 Ma is about 0.909?

  *We will fix the typo regarding the $^{207}$Pb/$^{206}$Pb$_{common}$ ratio at 1000 Ma which is 0.909 (line 281).* This is a typo and we've confirmed that the correct ratio values were used in all calculations.

- L299-L300 "However, the similarity in the 207Pb with t208 cumulative date distribution for the 100–40% and 40–0% groups is notable,". I really didn't understand this clause – sorry.

  *The revised manuscript will exclude the t_208 correction as it becomes irrelevant with an iterative $^{207}$Pb correction.* This should help simplify this section. This will remove the confusion with Lines 299-300, which were intending to indicate that the cumulative distribution plot (Fig. 5) shows that the date distributions are similar for the 40-0% concordant and 100-40% concordant groups with $^{207}$Pb$_{t208}$-corrected ages. This could warrant the inclusion of more discordant grains (<40%) in the final interpretations. This does not hold for the $^{207}$Pb$_{ti}$-corrected ages, so *this discussion point will be removed*.

3. **Comments regarding trace element data**

- suggestions for improvements including: (c) Make more of the PCA plot, and also of your own data.

- the PCA approach to exploring the data is interesting, but currently not much is made of it.

- Section 6.3 – PCA – Not quite sure what the main point this paragraph is trying to tell us – it could be expanded on. It does show the Cr vs Nb plot is useful in that Cr (+ V which has similar behaviour) pulls in an opposite direction to Nb (+ Ta which has similar geochemical behaviour). So the Cr vs Nb plot does a good job of separating the fields. If you were to crudely put on mafic vs pelitic fields on the PCA plot (boundary between yellow vs green), then the Hf + Zr vectors would be roughly parallel, showing the mafic vs pelitic distinction is somewhat independent of temperature. But the plot is introduced without significant additional interpretation.

We used the detrital rutile trace element dataset in several ways. We confirmed that analyzed grains were rutile and not other polymorphs using Cr, V and Zr (Appendix). The Cr and Nb concentrations are used to discriminate mafic and pelitic protoliths and Zr concentration is used to determine Zr-in-rutile temperatures. These applications only use the results of 1 or 2 elements. To evaluate the whole suite of trace elements analyzed, we used principal component analysis. Dimensionality reduction methods like PCA and tSNE are useful for evaluating clustering in large, multivariate datasets (we did not discuss tSNE in the manuscript as the results were similar to PCA). We hoped that these methods would help distinguish detrital populations. However, the PCA results show that *"the variance between [grains] can largely be explained by Hf, Zr, Sn, Cr, V, Nb and Ta. Because Cr, Nb and Ta are protolith dependent (PC 2) and Hf and Zr are temperature dependent (PC 1), the variance in detrital rutile trace element chemistry is best explained by both protolith and metamorphic grade, allowing us to track these two properties of source rocks." (Lines 376-379)*

This means that the protolith and Zr-in-rutile sections are already exploring the most interesting aspects of the trace element dataset. We agree that the PCA plot shows that protolith and temperature are independent. ***The revised manuscript will clarify the most salient points of the PCA results. We will emphasize that the protolith and temperature sections capture the most important components of the trace element results. Additionally, we will revisit the trace element data to evaluate trends in samples and/or age populations and consider adding spider plots, bar plots, and/or other clustering algorithms as relevant. The revised manuscript will include updated text, figures and supporting information to support any new findings.***

- Figure 8 – not clear which lines defining fields belong to which paper (Triebold vs Meinhold) on the Cr/Nb plot without reading the text – label them.

Thank you. ***We will update the figure labels in the revised manuscript.***

4. **Additional comments**

- First 8 lines of abstract. I really think there needs to be a caveat here. Very broadly speaking (and there are exceptions), rutile is not particularly common in igneous rocks in the crust and requires reasonably high pressures to crystalize – it is mainly a metamorphic mineral (where again it requires reasonably high pressures to crystallize). It is better than zircon in recording metamorphic events in provenance studies but it too has a relatively restricted paragenesis.

We included the mention of igneous rutile to be complete, but we agree that it is uncommon. ***The revised manuscript will be updated to include this caveat.***

- L46 – I am not sure why this sentence starts with "In convergent margin settings….", as I feel it is applicable to other tectonic settings as well.

Yes, we agree with this point. *We will keep the focus on convergent margin settings following the recommendations of the reviewers. The revised manuscript will have an updated focus that is oriented toward Türkiye and suture zone settings, as discussed in Section 1 above and in our reply to Reviewer #3.*

- L64 – and detrital apatite. In terms of recent publications (i.e. last 5 years) I feel it is more commonly used nowadays than either detrital monazite and detrital muscovite. Please also list the geochronological system applying to the mineral – U-Pb, 40Ar/39Ar, Luf-Hf etc

*The revised manuscript will be updated to include detrital apatite and isotope systems.* The list in the manuscript was meant to only include popular U-Pb minerals.

- Section 2.1 This section needs something on the role of pressure and composition on the stability field of igneous and metamorphic rutile, and also the well documented instability instability in the sub-greenschist to lower greenschist facies (Zack et al., 2004; Yakymchuk et al., 2017).
   - L216. Please also provide pressure estimates. The pressure dependence on the stability of rutile is not getting much attention in this manuscript.

The original manuscript did not include much background on the pressure conditions of rutile formation. Pressure was briefly addressed with the Zr-in-rutile thermometer (Section 2.1). To calculate Zr-in-rutile temperatures, we use an average pressure of 13 kbar with an uncertainty of 5 kbar, which is a reasonable pressure range for rutile in metamorphic rocks (e.g., Zack & Kooijman, 2017). The stability field of rutile has been discussed in some review papers (e.g., Chew et al., 2020, Tropper & Manning 2008), but is not always given as much attention. *The revised manuscript will address the stability of rutile in the background section. Additionally, this will strengthen the interpretations. The dominant 190 Ma age mode is interpreted to be sourced from the Karakaya Complex, which was metamorphosed to blueschist and epidote-amphibolite with minor eclogite facies. In the blueschist facies, rutile is expected to be stable above ~14 kbar and at temperatures ~400-500 $^o$C in metagranitoid compositions (Angiboust and Harlov, 2017), whereas it can coexist with ilmenite from 5-8 kbar and be solely stable above 8 kbar in metapelitic compositions (e.g., Zack & Kooijman, 2017). The 190 Ma rutiles largely have temperatures from 450 to 500 $^o$C. The revised manuscript will include a discussion of rutile stability in the background section and will include further justification for the interpretation of a Karakaya Complex source based on expected rutile stability.*

- Sections 7.2 – Some points below link back to substantive points made at the start of the review:
   - "the 190 Ma population is poorly represented in the detrital zircon record" – but the counterpoint needs to be made that the 90 Ma population is very important in the detrital zircon record and not in the rutile record.

This is a good point. We emphasized that the detrital rutile record contains age populations not present in the detrital zircon record. *The revised manuscript will also emphasize that the zircon record contains prominent age modes at ca. 395 Ma, 110-66 Ma, and 50 Ma that are not well represented in the rutile record and are more likely to represent timespans dominated by magmatism/volcanism. We postulated that the absence of 395 Ma rutile could be a signal of sediment recycling (lines 405-415). We will add to this discussion that there are Late Cretaceous and Eocene magmatic flare ups associated with subduction and syn-collisional magmatism, respectively. Collision-related metamorphism is recorded in the lower plate (the samples are from upper plate basins), so the absence of Late Cretaceous-Eocene detrital rutile could support the absence of sediment transport across the suture zone (which is also shown by Okay & Kylander-Clark (2023) using the detrital zircon record).*

*Although we note there are a few Late Cretaceous detrital rutile grains in our record. We will address this in detail in the revised manuscript.*

**References Cited**

Angiboust, S. and Harlov, D.: Ilmenite breakdown and rutile-titanite stability in metagranitoids: Natural observations and experimental results, Am. Mineral., 102, 1696–1708, https://doi.org/10.2138/am-2017-6064, 2017.

Caracciolo, L., Ravidà, D. C. G., Chew, D., Janßen, M., Lünsdorf, N. K., Heins, W. A., Stephan, T., and Stollhofen, H.: Reconstructing environmental signals across the Permian-Triassic boundary in the SE Germanic Basin: A Quantitative Provenance Analysis (QPA) approach, Glob. Planet. Change, 206, 103631, https://doi.org/10.1016/j.gloplacha.2021.103631, 2021.

Chew, D., O'Sullivan, G., Caracciolo, L., Mark, C., and Tyrrell, S.: Sourcing the sand: Accessory mineral fertility, analytical and other biases in detrital U-Pb provenance analysis, Earth-Sci. Rev., 202, 103093, https://doi.org/10.1016/j.earscirev.2020.103093, 2020.

Chew, D. M., Sylvester, P. J., and Tubrett, M. N.: U–Pb and Th–Pb dating of apatite by LA-ICPMS, Chem. Geol., 280, 200–216, https://doi.org/10.1016/j.chemgeo.2010.11.010, 2011.

Clift, P. D., Mark, C., Alizai, A., Khan, H., and Jan, M. Q.: Detrital U–Pb rutile and zircon data show Indus River sediment dominantly eroded from East Karakoram, not Nanga Parbat, Earth Planet. Sci. Lett., 600, 117873, https://doi.org/10.1016/j.epsl.2022.117873, 2022.

Mark, C., Cogné, N., and Chew, D.: Tracking exhumation and drainage divide migration of the Western Alps: A test of the apatite U-Pb thermochronometer as a detrital provenance tool, GSA Bull., 128, 1439–1460, https://doi.org/10.1130/B31351.1, 2016.

Najman, Y., Mark, C., Barfod, D. N., Carter, A., Parrish, R., Chew, D., and Gemignani, L.: Spatial and temporal trends in exhumation of the Eastern Himalaya and syntaxis as determined from a multitechnique detrital thermochronological study of the Bengal Fan, GSA Bull., 131, 1607–1622, https://doi.org/10.1130/B35031.1, 2019.

Odlum, M. L., Stockli, D. F., Capaldi, T. N., Thomson, K. D., Clark, J., Puigdefàbregas, C., and Fildani, A.: Tectonic and sediment provenance evolution of the South Eastern Pyrenean foreland basins during rift margin inversion and orogenic uplift, Tectonophysics, 765, 226–248, https://doi.org/10.1016/j.tecto.2019.05.008, 2019.

Okay, A. I. and Kylander-Clark, A. R. C.: No sediment transport across the Tethys ocean during the latest Cretaceous: detrital zircon record from the Pontides and the Anatolide–Tauride Block, Int. J. Earth Sci., 112, 999–1022, https://doi.org/10.1007/s00531-022-02275-1, 2023.

Okay, N., Zack, T., Okay, A. I., and Barth, M.: Sinistral transport along the Trans-European Suture Zone: detrital zircon–rutile geochronology and sandstone petrography from the Carboniferous flysch of the Pontides, Geol. Mag., 148, 380–403, https://doi.org/10.1017/S0016756810000804, 2011.

Şengün, F., Zack, T., and Dunkl, I.: Provenance of detrital rutiles from the Jurassic sandstones in the Central Sakarya Zone, NW Turkey: U-Pb ages and trace element geochemistry, Geochemistry, 80, 125667, https://doi.org/10.1016/j.chemer.2020.125667, 2020.

Shaanan, U., Avigad, D., Morag, N., Güngör, T., and Gerdes, A.: Drainage response to Arabia–Eurasia collision: Insights from provenance examination of the Cyprian Kythrea flysch (Eastern Mediterranean Basin), Basin Res., 2020, https://doi.org/10.1111/bre.12452, 2020.

Smye, A. J. and Stockli, D. F.: Rutile U–Pb age depth profiling: A continuous record of lithospheric thermal evolution, Earth Planet. Sci. Lett., 408, 171–182, https://doi.org/10.1016/j.epsl.2014.10.013, 2014.

Thomson, S. N., Gehrels, G. E., Ruiz, J., and Buchwaldt, R.: Routine low-damage apatite U-Pb dating using laser ablation–multicollector–ICPMS, Geochem. Geophys. Geosystems, 13, https://doi.org/10.1029/2011GC003928, 2012.

Tropper, P. and Manning, C.E., 2008. The current status of titanite–rutile thermobarometry in ultrahigh-pressure metamorphic rocks: the influence of titanite activity models on phase equilibrium calculations. *Chemical Geology*, *254*(3-4), pp.123-132.

Vermeesch, P.: IsoplotR: A free and open toolbox for geochronology, Geosci. Front., 9, 1479–1493, https://doi.org/10.1016/j.gsf.2018.04.001, 2018.

Zack, T., and Kooijman. E.: Petrology and Geochronology of Rutile, Reviews in Mineralogy and Geochemistry 2017, 83 (1), 443–467, https://doi.org/10.2138/rmg.2017.83.14.

---

## Author Comment (AC2)

**Response to Reviewer #2's comments on manuscript egusphere-2023-1293**

**"An expanded workflow for detrital rutile provenance studies: An application from the Neotethys Orogen in Anatolia"**

Megan A. Mueller, Alexis Licht, Andreas Möller, Cailey B. Condit, Julie C. Fosdick, Faruk Ocakoğlu, and Clay Campbell

November 15, 2023

We appreciate the thoughtful reviews of the 3 referees. First, we summarize the broad themes from the three reviews before responding in detail to Reviewer #2 below. The reviews critiqued (1) the novelty of the study, (2) the number of U-Pb analyses discarded during data reduction, (3) the potential bias of discarding data and the validity of interpreting discordant U-Pb analyses, and (4) the apparent lack of novelty and complexity in geochemical data interpretations. Regarding these points, the goal of the manuscript was to provide a transparent workflow for detrital rutile geochronology using new data from sedimentary basins in Anatolia. We acknowledge that it was confusing or misleading to present this work as a revised workflow. We intended to emphasize that we found a paucity of methods papers that provide a straightforward approach. Although we did not claim to present new workflows with the geochemical data, this was perhaps unintentionally implied with the title. We have presented as much information as we could squeeze out of our particular geochemical dataset. The revised manuscript will scale back statements on new workflows and instead refocus the title and introduction on suture zone settings and Anatolian geology while maintaining the broad overview of detrital rutile provenance and the details of our methods.

We also acknowledge that it is surprising to see the number of data discarded during data reduction. However, we contend that this is a common practice with detrital U-Pb geochronology in common Pb-bearing minerals. We are not the first to discard a significant number of analyses; we have found this practice in many published detrital rutile datasets, although it is not discussed much in the literature. We further expand on this point in our detailed reply and will add this context to the revised manuscript. This manuscript gives precedence for papers to be transparent in data reporting—including the number of grains analyzed and criteria for rejection—as well as examination of the full dataset in Tera-Wasserburg diagrams. Our manuscript provides an opportunity to show the consequence and potential value of the full dataset, which is relevant to others working with this type of data. While the use of common Pb-bearing minerals is common in some labs and geographic settings, the application of these tools is still far behind detrital zircon geochronology, for which there is a well-established global framework. We have encountered many detrital geochronology users who want to add detrital rutile to their toolkits but are still uncertain in how to collect and interpret these data. Here, we present a complicated detrital rutile U-Pb dataset that can serve as an example for how to treat and interpret complex, yet potentially meaningful, discordant data.

Reviewer #2 provided a critical review of our manuscript with comments that centered around (1) the U-Pb data reduction and handling of the reference materials, (2) rejection of U-Pb analyses, and (3) the validity of provenance interpretations on discordant data. Reviewer #2 included numerous in-text comments that are summarized in their review text. We thank Reviewer #2 for the thorough feedback. Reviewer #2's comments will strengthen the clarity of the revised manuscript, yet we generally disagree with Reviewer #2's characterization of points (2) and (3) and provide a rebuttal.

Reviewer #2 provided comments in the review and line-by-line comments in the text. We address the comments below by theme. Reviewer #2's comments are included below in black text. Our response is in purple text and the specific changes we will implement are highlighted in ***bold, italic purple text***.

1. **Comments regarding U-Pb analytical protocol, data reduction, and data handling of reference materials**

   - the data handling of the reference materials (unreported scatter in the measured and corrected isotopic ratios of the reference materials, inconsistency between uncertainties before and after the correction, lack of representation of the data as Concordia diagrams)

   - Line 145: The Th/U ratio of NIST SRM 612 is expected to be close to 1 (e.g. see Guillong et al 2003 JAAS "U/Th. The comparison of Th and U has the advantage that isotopic abundance, concentrations and first ionisation potentials are similar and a ratio determination should therefore give approx. 1."). This means that the joint tuning of the MS and laser system was not optimal.

   - Line 175: For R10, the uncert. of the uncorrected 6/38 age ("2SE prop abs" from Iolite in table "Uncorrected data output", supplem S2) ranges from 2.2 to 4.1%, for secondary ref mat LI04-08 the uncert. ranges from 2.3 to 4.3%. In contrast, the uncert. of the 6/38 208-corrected dates is 0.4-1.2% and 0.5-1.1%, respectively (208 corrected Table, supplem S2). The uncertainty of the corrected dates appears to be substantially smaller than the uncorrected dates. See also further comments in the U-Pb method section.

   - Line 250: MAJOR COMMENT.

     o The uncorrected R10 data (suppplem table S2, final 7/6 and 6/38 ratios and 2SE prop uncert) are scattered along Concordia and some datapoints are largely discordant. The authors should clarify if any correction has been applied to the primary ref material before normalization of the samples and secondary ref mat. ratios; if not, why.

     o The INDIVIDUAL undertainty of the final UNCORRECTED R10 ratios (table S2, Iolite output: 2SE propagated uncert ) is on average 2.3% for the 7/6 ratio and 3.6% for the 38/6 ratio. In contrast, the reported uncertainty of the CORRECTED R10 data is an avg of 0.7% for the 38/6 ratio or 6/38 age (208 correction, 207 correction, 207 correction with 208 as initial age, 207 then 208 correction, table S2). The average individual uncert for the 7/6 age after the correction is 2.1% (all correction approaches, with ca 50% of the individual uncertainties smaller than the original uncertainty and the rest larger up to 0.5%.) This raises questions about the uncertainty propagation protocol associated with the correction(s).

   - Line 250: MAJOR COMMENT the U-Pb data of the reference materials (and the samples) must be plotted on a Concordia diagram. Additionally, presenting the results as weighted averages values only, without reporting the corresponding MSWD, masks any potential scatter in the data (Fig. A2 and Table "Std reproducibility" in supplem file S2). Additionally, it is unclear how the R10 uncorrected wtd average age reported in Table "Standard reproducibility" (supplem S2) has been calculated., as it differs from the wtd avg calculated (with IsoplotR) using the data reported in Table "uncorrected data output". E.g. the final uncorrected 7/6 age wtd avg (calculated using the 2SE propagated uncert.) is 1088.3 +/- 2.8 Ma, n=215/215, MSWD =0.41, vs the 1091.7 +/-0.5 Ma age reported in in Table "Standard reproducibility"; the final uncorrected 6/38 age wtd avg is 1092.21 + /- 2.32 Ma, n=215/215, MSWD 0.35. This should be clarified also for the secondary reference materials, and averages re-calculated form the presented data when they do not match.

   - Line 250: MAJOR COMMENT. The plot in figure A2 masks significant scatter in the corrected data. E.g. the "208 corrected ages" for R10 range from ca 1060 to 1120 Ma (largely not overlapping at the 2s level), with an average of ca. 1091 Ma however with a MSWD of 5.1, n=210/214 ; this would be apparent if the data had been plotted on a Concordia diagram or weighted avg or linearized probabilty plot. The "207c with 208c as initial age" corrected avg ages

for R10 are: 7/6 corrected age = 1138.6+/- 1.6, n=214, MSWD 8.7; 6/38 corrected age = 1093.7 +/-0.5, 212/214, MSWD 7.6. Such a scatter must be reported and discussed.

- Line 255: MAJOR COMMENT The need for rejection of more than 50% of the analyses due anomalous spiky patterns possibly points to an underlying issue with the analytical protocol, data acquisition or the samples themselves, or their preparation. Were similar spiky patterns observed for the reference materials? Had masses 204-202 been monitored, the occurrence of inclusions or unevenly distributed common Pb would have been identified by simply observing the corresponding signal intensities/pattern in the time resolved data.

- Line 255: On which basis such a 20% 7/6 uncertainty threshold was chosen? Considering that the rutile populations of this study are dominated by 100-300 Ma rutiles (Fig.5), a threshold should have been set on the 6/38 ratio uncertainty instead. (Cf Govin et al 2018 Geology Table DR7; LA U-Pb detrital rutile data similarly collected with a single collector SF-ICPMS)

- Table A2: The choice of 3 passes (instead of 1) for U-Pb analysis is quite unusual. It causes that each set of 3 passes (one "pass" being a sweep across the mass range of interest, 206-238 in this case) is averaged by the Element software into one of the 100 runs, and although a total of 300 sweeps will be measured, the data output will consist of 100 datapoints, not 300. This has implications in terms of counting statistics nd final uncertainty of the ratios. Why this choice, as opposed to measuring the trace elements by means of 120 runs x 1 pass?

The analytical protocol is sound. The same or very similar protocols and data acquisition settings have been in multiple previous studies carried out in the same lab (e.g., Rösel et al., 2019). (1) Anomalous signal patterns were not observed in the reference materials. The rejection of data appears to be a characteristic of this detrital sample set, similar to several other published detrital datasets (e.g., Govin et al. 2018 discarded nearly 40% of data from a much smaller dataset, n=147). See our reply to point #2 below. (2) Regarding the analytical set-up, the tuning was optimized for high sensitivity and low oxide production, while accepting a less than optimal element fractionation. This was intended to allow analyses of low U and Pb rutile. Since the elemental fractionation is corrected during the data reduction process, this does not invalidate or adversely affect the ability to use data, whereas low sensitivity would exclude even more low U and Pb rutile from being able to be used. ***This sentence will be updated in the revised manuscript (line 145).*** (3) The choice of multiple passes is not unusual and has been standard practice for U-Pb dating in multiple labs since before 2007 (e.g., Frei & Gerdes 2009). We are aware of the effects. We hold that the chosen method provides adequate data even for rutile with very low U and Pb concentrations, and fast enough mass scans to identify inclusions. The obvious difference between the U-Th-Pb and trace elements method is that each scan through the masses for the trace element method is much longer than for Pb-Th and U, which does not require settling times of the magnet. A single pass with adequate counting times was therefore chosen for the trace element method and yields a similar number of data points.

Reviewer #2 raises an important point about the reference materials. We address this point by first discussing the primary standard R10, then by clarifying the analytical protocol and discussing the secondary standard results. There is no excessive scatter or discordance in the calibration standard R10. The uncorrected results are displayed in Wetherill space in Figure R1 (using propagated uncertainties). For all analyses (n=210), the concordia age is 1091.7 ± 1.7 Ma with a MSWD of 0.66 and MSWD for concordance and equivalence of 1.2. A common Pb correction was not applied to R10 as Luvizotto et al. (2009) writes, *"LA-ICP-MS data show that the non-radiogenic Pb concentrations (measured as 208 Pb) are very low (average of 0.08 ppm), meaning that common Pb can be neglected for [R10]."* ***The revised manuscript will include concordia diagrams of the reference materials and will clarify that the results are uncorrected for common Pb.***

The U-Pb data was collected in 4 analytical sessions. The analytical protocol was modified from session to session to optimize for the analysis of low U and Pb unknowns and high U and Pb reference

materials. In the first two analytical sessions, 21RtF and 21RtG, Pb and Th isotopes were measured with the secondary electron multiplier (SEM) operating in counting detection mode. The secondary standards Wodgina and Kragerø did not perform well during those analytical sessions: there is an increase in $^{206}$Pb and $^{206}$Pb/$^{238}$U ratio and sharp decrease in $^{207}$Pb/$^{206}$Pb at the beginning of the analysis (Figure R2). During those sessions, Wodgina reached ca. 4 million $^{206}$Pb counts per second, which exceeds the limit of linear behavior in counting detection mode. Kragerø has the second highest counts with a maximum of around 3 million $^{206}$Pb counts per second, and the other standards remained within the linear behavior range. We conclude that the poor performance of Wodgina and Kragerø on those days was due to this counting mode issue and could explain the extreme increase in $^{206}$Pb/$^{238}$U ratio at the start of the analysis (Figure R2). This issue was corrected in the following two sessions, 21RtA and 21RtB, in which the SEM operated in "both" detection mode for Pb and Th isotopes, and Wodgina performed well (Figure R2). Figure R3 displays the Wodgina results from all analytical sessions in Wetherill space: data from the first two sessions are discordant and were rejected, data from the final two sessions are acceptable with a concordia age of 2819.0 ± 3.3 Ma (2s) (less than -1% offset from TIMS age). Note that only one analysis of the unknown came close to the high count rates observed in the Archean high-U rutile Wodgina, so we argue that the issue did not affect the unknowns. We compare the Wodgina results to that of secondary standard 9826J because 9826J was not affected by the SEM detection mode issue (ca. < 750,000 CPS $^{206}$Pb). We compare the performance of 9826J across analytical sessions (Figure R4). In sessions 21RtF and 21RtG, 9826J has a concordia age of 378.1 ± 1.9 Ma (2s) with an MSWD for concordance and equivalence of 2.2, indicating excess scatter. This age is slightly younger than the TIMS age of 381.9 ± 1.1 Ma (Kylander-Clark, 2008), but only offset by about -1%, the long-term reproducibility range of LA-ICP-MS data. From session 21RtA, 9826J has a concordia age of 385.9 ± 3.8 Ma (2s), is within the uncertainty of the TIMS age, and has an MSWD for concordance plus equivalence of 0.88, indicating slight underdispersion. The concordia ages are satisfactory for all analytical sessions, but there is excess scatter in the first two analytical sessions which could be natural.

Next, we investigated whether excess scatter in the secondary standards was a result of the drift correction curve. In Iolite, we compared the effects of weighted linear fit, SplineSmooth5 and SplineSmooth9 curves for drift correction. The choice of spline had little effect on the final age of the reference materials but impacted the MSWD. Scatter in the reference materials is not explained by the drift correction curve. We prefer the weighted linear fit as this model reproduces the secondary standard ages and brings the MSWD closest to 1 for each standard. *The revised manuscript will include the results that were reduced using the weighted linear fit drift correction. All figures and tables in the main text and supplement will be updated. We do not anticipate that this change will produce significant changes to the results or interpretations.*

We maintain that the poor performance of Wodgina and Kragerø in the first analytical sessions was a limitation of the SEM detection mode and does not indicate that the entire analytical sessions should be discarded: (1) the R10 and 9826J results were acceptable for analytical sessions 21RtF and 21RtG, (2) the signal patterns and results were acceptable for Wodgina in analytical sessions 21RtA and 21RtB, (3) in the unknowns, the Pb and Th isotopes signal intensities were within the linear range of the counting mode as evidenced by the data on the other reference materials. *In the revised manuscript, the Wodgina and Kragerø analyses from sessions 21RtF and 21RtG will be excluded. The text and supporting information will be updated to explain the changes in analytical protocol across sessions. Concordia diagrams of reference materials will be added. Here we have shown concordia diagrams of several of the standards, and diagrams of all standards will be included in the revised manuscript. We will discuss in the text and/or supplement the day-by-day performance of the standards.*

Regarding the $^{207}$Pb/$^{206}$Pb ratio threshold, the 20% rejection criteria is following previous studies carried out in the same lab (e.g., Lippert, 2014). In the study suggested by Reviewer #2, Govin et al. (2018), the rejection criterion is based on the uncertainty on the corrected age. The threshold changes based on the corrected age; for example, analyses with uncertainties > 10 % are discarded for $^{207}$Pbcorrected ages > 100 Ma. This age-dependent threshold is similar to approaches used in detrital zircon and can induce bias (Malusà et al., 2013). This rejection criterion depends on how the uncertainty is calculated, including whether uncertainties on the Stacey and Kramers (1975) values are accounted for, for example. For our dataset, Govin et al.'s age uncertainty filter excludes about 20% of the analyses that are included by the $^{207}$Pb/$^{206}$Pb filter. However, Govin et al. (2018) do not exclude any data based on discordance, which, for our dataset, would include more analyses than the protocol we used. ***For the revised manuscript, we will re-examine whether to discard analyses based on the $^{207}$Pb/$^{206}$Pb ratio, $^{206}$Pb/$^{238}$U ratio and/or the uncertainty on the corrected age. If warranted, we will include a brief discussion in the revised manuscript.***

We calculated individual uncertainty in the following way, after Odlum et al. (2019) (see lines 177-178). The uncertainty in percent is calculated using the initial $^{206}$Pb/$^{238}$U ratio and internal uncertainty. The uncertainty on the final age is calculated from the percent uncertainty on the initial ratio. For example, if the initial $^{206}$Pb/$^{238}$U ratio has 2% uncertainty at 2 sigma and the corrected age is 200 Ma, then the corrected age uncertainty is ±4 Ma (2s). We used the internal uncertainty rather than propagated uncertainty, which is likely why the reviewer calculated larger uncertainties. ***The revised manuscript will use the propagated uncertainty. The data tables, figures and text will be updated. The revised manuscript will include Tera-Wasserburg diagrams of the reference materials using the 2s propagated uncertainty. We will display the concordia age and MSWD. We will eliminate Figure A2 and Dataset S2 tab 'Standard Reproducibility' in favor of the new Tera-Wasserburg plots.*** Regarding the uncertainty on the corrected and uncorrected ages of the reference materials, the discrepancy described by Reviewer #2 is likely a result of comparing the internal and propagated uncertainties. ***This should be resolved in the revised manuscript with the use of the propagated uncertainty.***

[Figure]

*Figure R1. All R10 results are displayed in Wetherill space. The ellipses are colored by the analytical session. For all analytical sessions, the concordia age is 1091.7 ± 1.6 Ma with an MSWD of 0.66 and MSWD for concordance and equivalence of 1.2. Uncertainties and ellipses are propagated 2s absolute. The calculated concordia age is shown as a white shaded ellipse. A common Pb correction was not applied. Figure made using IsoplotR (Vermeesch, 2018).*

*Figure R2 (next page). Comparison of secondary standard Wodgina from analytical sessions 1 (21RtF) and 3 (21RtA), representing the switch from operating the secondary electron multiplier (SEM) detection mode in counting to both. The first 6 Wodgina analyses from each session are displayed. Note the increase in $^{206}Pb$ and $^{206}Pb/^{238}U$ ratio and sharp decrease in $^{207}Pb/^{206}Pb$ downhole during session 21RtF (counting mode).*

(a) Wodgina from analytical run 21RtF

(b) Wodgina from analytical run 21RtF

(c) Wodgina from analytical run 21RtA

[Figure]

(d) Wodgina from analytical run 21RtA

[Figure]

[Figure]

*Figure R3. All Wodgina results displayed in Wetherill space. The ellipses are colored by the analytical session. Wodgina from sessions 21RtF (yellow) and 21RtG (red) are excluded from the concordia age calculation. For analytical sessions 21RtA and 21RtB, the concordia age is 2819.0 ± 3.3 Ma with an MSWD for concordance and equivalence of 1.3. Uncertainties and ellipses are propagated 2s absolute. Ellipses that are not shaded are excluded from concordia age calculations (i.e., analyses from sessions 21RtF and 21RtG). The calculated concordia age is shown as a white shaded ellipse. A common Pb correction was not applied. Figure made using IsoplotR (Vermeesch, 2018).*

*Figure R4 (next page). Results of secondary standard 9826J from analytical sessions 21RtF, 21RtG and 21RtA. From sessions 21RtF and 21RtG, 9826J has a concordia age of 378.1 ± 1.9 Ma (2s) with an MSWD for concordance plus equivalence of 2.2, indicating excess scatter. From session 21RtA, 9826J has a concordia age of 385.9 ± 3.8 Ma (2s) with an MSWD for concordance plus equivalence of 0.88, indicating slight underdispersion. Uncertainties and ellipses are propagated 2s absolute. Ellipses that are not shaded are excluded from concordia age calculations. The two calculated concordia ages are shown as white shaded ellipses. A common Pb correction was not applied. Figure made using IsoplotR (Vermeesch, 2018).*

[Figure]

**2. Comments regarding rejected and discordant U-Pb data**

- the sample dataset which should serve as the case study to validate the workflow (rejection of >50% of the initial analysis due to "anomalous spiky patterns", followed by further rejections due to large 7/6 uncertainty, leaving with only 30% of the initial data, with no possibility to carry out provenance interpretation at the sample level due to too small n per sample; last but not least vast majority of these remaining data being largely discordant despite the different common Pb correction approached used

- Line 195: Following the correction, the data from this study are largely discordant.

- Line 280: MAJOR COMMENT. U-Pb data should be plotted as Concordia diagrams with uncertainty represented as ellipses, particularly in a manuscript submitted to a geochronology-focused journal. Alternative ways to plot the data are welcome in addition.

- Table A2: Where are these Concordia diagrams with 2s ellipses?

Reviewers #2 and #3 raised concerns about the number of analyses discarded during data reduction. The workflow that we present includes the analyses of all detrital rutile grains in a sample. *"Rutile grains were handpicked; all rutile grains were picked from most samples, except for samples 16SKY26, 16SKY42 and 17OZK05 for which 260–320 grains were selected"* (lines 238-239). Part of the reason that provenance interpretation is not possible at the sample level is due to the low rutile yield in some samples. For example, sample 18DMN01 only had 5 rutile grains, all of which were analyzed, and 4 ages were obtained. We suspect that this was a feature of local geology (i.e., metamorphic sources were not exposed at the surface at the time the sample was deposited), however, in the future we will collect larger samples to try to increase heavy mineral yield. The second reason is due to the exclusion of

analyses during data reduction, for reasons including poor signal intensity and large uncertainty in the $^{207/206}$Pb ratio. Anomalous signal intensity was likely due to a combination of very low uranium concentrations, "*elemental heterogeneity from ablating into small inclusions and/or lamellae, and inhomogeneities due to micro-cracks with different element/isotope composition*" (lines 257-258).

The exclusion of analyses during data reduction is not unique to this study, but represents a quite common problem with large-*n* detrital rutile provenance work. Many published studies discuss analyzing a larger number of grains than are presented—discarding analyses—due to low U and/or low radiogenic Pb content (Bracciali et al., 2013; Caracciolo et al., 2021). For example, Shaanan et al. (2020) discard 60% of their detrital rutile U-Pb data due to discordance. Similarly, Reviewer #1 points to the study by Caracciolo and co-authors (2021) that analyzes 712 detrital rutile grains without a U-filter, yet, after discordance filtering, only 347 grains remained (48%) (from what we can tell as the data is not available online). Additionally, Caracciolo et al. (2021) did not have enough rutile ages per sample to discuss sample-by-sample provenance interpretations, which we also experienced with our dataset. This points to a larger problem in trying to scale up detrital rutile to large-*n* provenance applications. For this reason, we wanted to confidently include as many U-Pb analyses as possible in our interpretations, which led to the exploration of U-Pb discordance.

To address this comment, ***the revised manuscript will emphasize that the rejection of analyses during data reduction is not a unique limitation of this study. We will clarify that this is common in studies that have attempted large-n detrital rutile U-Pb. The original text emphasized the role of inclusions and lamellae in potentially contributing to poor signal intensity patterns and did not clearly emphasize that, in addition, low signal intensity could be from low U and/or low radiogenic Pb contents. The revised manuscript will also include low U and low Pb as a potential cause of poor raw signal intensity. As suggested by Reviewer #3, we will also include a data treatment section of the Appendix that includes a figure with examples of signal intensity patterns in the unknowns.***

We are unclear about Reviewer #2's criticism (1) of the data as discordant after common Pb correction and (2) the apparent absence of concordia diagrams. We suspect that (1) is due to a lack of clarity in what is being shown in the figures. The uncorrected results are displayed in Tera-Wasserburg space (Figures 4, 7b, 8c, 10). The common Pb corrections force concordance, so all analyses are concordant after correction. The corrected data are shown as histograms, KDEs and CDFs. The figures with data displayed in Tera-Wasserburg space are the uncorrected ratios, whereas the data displayed as histograms, KDEs and CDFs are the common Pb-corrected data (Figures 5, 7c, 12, A3); although one KDE in Figure 5 shows the full uncorrected dataset. Regarding (2), Tera-Wasserburg diagrams are concordia diagrams, but perhaps the reviewer is referring to Wetherill diagrams. However, it is generally more useful to display data from common Pb-bearing minerals in Tera-Wasserburg space as it is easy to see the linear discordia arrays between the common and radiogenic ratios (many examples, including Chew et al., 2014). ***The revised manuscript text and figure captions will clarify whether the uncorrected or common Pb-corrected data are displayed. The revised manuscript will include the figures displaying the reference materials in Tera-Wasserburg diagrams with 2s error ellipses.*** The current Figure 4 displays the unknowns in Tera-Wasserburg space with error bars so that the figure is less crowded, ***and the revised manuscript appendix will include the same figure with 2s error ellipses.***

3. **Comments regarding provenance interpretations in common Pb-bearing minerals**

   ● attempted provenance analysis based on such a largely discordant dataset, with age modes identified in the KDEs distributions of up to 60% discordant data and unsupported claim that the KDEs distributions of variably discordant data are similar hence meaningful, and can be used to constrain provenance

- Line 85: The conclusions of this study are not supported by the data; following the application of the common Pb correction methods the sample data remain largely discordant, rising questions about the effectiveness of the correction approaches and or assumptions. Importantly, no common Pb correction appears to have been applied to the primary reference material R10

- Line 130: In this study a very large portion of the "common Pb corrected" data are rejected, which makes the authors' approach as biased as the U-based filtering used by others.

- Line 140: Published studies have shown detrital rutile samples where a proportion (of the uncorrected data) is concordant, in addition to (typically one) array of discordant dates intercepting the cluster of concordant dates (Bracciali et al 2013, 2015, 2016).

- Line 255: rejection of 70% of the sample data is remarkably high, inevitably biasing the dataset and hindering any provenance interpretation (even if the remaining data were close to concordant, which are not)

- Line 285: MAJOR COMMENT. Final sample sizes are small following rejection of 70% of the initial data. The inability to assess the approach presented in this at the individual sample level is a major limitation.

- Line 290: MAIN COMMENT. The authors aim to use a set of natural samples to test their approach. Not only i) the original dataset is largely biased by rejection of ca 70% of the collected data (due to "anomalous -spiky- signals") and 7/6 uncert >20%, lines 253-256, but ii) following (207 or 208) correction, the data are largely discordant. Identifying "age modes" in KDEs generated from largely discordant data is pointless and such practice should be avoided as the modes or peaks of discordant data are geologically meaningless. The final aim of any detrital provenance study is to identify the timing of real geological events which can be tracked back to the rock sources. Age peaks or modes derived from distributions of largely discordant U-Pb data are geologically meaningless. Because of i) and ii) the results of this study cannot support any robust provenance interpretation.

- Line 300: The 100-40% concordant group includes in the first place largely discordant dates which are geologically meaningless. The similarity between the 0-40% conc and the 100-40% conc distribution does not justify inclusion of discordant dates.

- Line 300: I am afraid I have to disagree here. The comparison of the KDEs distributions of Fig 5, bottom panel indicates significant differences. A representation of the same data at a smaller scale (e.g. < 500 Ma) would enhance such differences (presence or absence of peaks, position of youngest peak). Such an overinterpretation is misleading and should not be encouraged as an acceptable practice.

Again, we suspect that the comment here is based on a lack of clarity about what is being shown in the figures. The uncorrected results are displayed in Tera-Wasserburg space (Figures 4, 7b, 8c, 10). The common Pb corrections force concordance, so all analyses are concordant after correction. The corrected data are shown as histograms, KDEs and CDFs. The figures with data displayed in Tera-Wasserburg space are the uncorrected ratios, whereas the data displayed as histograms, KDEs and CDFs are the common Pb-corrected data (Figures 5, 7c, 12, A3); although one KDE in Figure 5 shows the full uncorrected dataset. *This was not clear in the manuscript and will be clarified in the revised version.*

We suspect that this clarification addresses the concerns of Reviewer #2. Reviewer #2 argues that *"Identifying "age modes" in KDEs generated from largely discordant data is pointless and such practice should be avoided as the modes or peaks of discordant data are geologically meaningless. The final aim of any detrital provenance study is to identify the timing of real geological events which can be traced back to the rock sources. Age peaks or modes derived from distributions of largely discordant U-Pb data are geologically meaningless."* We disagree with this argument. In case the above clarification has not

resolved the discrepancy, we provide an explanation below for why geologically meaningful interpretations can be made from initially discordant data.

We maintain that geologically meaningful interpretations can be made from initially discordant data when appropriate common Pb corrections are applied. We again emphasize that the initially discordant data are concordant after common Pb correction; our interpretations are based on concordant data. U-Pb discordance in common Pb-bearing minerals is well documented in published reference materials and unknowns (e.g., Chew et al., 2011, 2014). Practically everyone using detrital rutile data is using data that is discordant before correction. Many publications focus on common Pb corrections and how to treat discordance—so that accurate interpretations can be made from initially discordant data (e.g., Faure, 1986; Williams, 1997; Ludwig, 1998; Andersen, 2002; Chew et al., 2011, 2014; McLean et al., 2011; Thomson et al., 2012; Smye and Stockli, 2014; Vermeesch, 2020, 2021). We note that many *in-situ* studies with common Pb-bearing minerals fit discordia arrays to co-genetic grains to derive a [207]Pb-corrected age. In these instances, individual analyses can be nearly 100% discordant and still interpreted confidently within the population of co-genetic grains (e.g., Poulaki et al., 2023). We are unsure why Reviewer #2 dismisses discordant rutile data, especially when discordant data are present and interpreted in their own papers that they referenced.

Reviewer #2 argues that *"Published studies have shown detrital rutile samples where a proportion (of the uncorrected data) is concordant, in addition to (typically one) array of discordant dates intercepting the cluster of concordant dates (Bracciali et al 2013, 2015, 2016)."* We agree that rutile U-Pb analyses can be concordant (e.g., Rösel et al., 2019, Kooijman et al. 2010, and many others), which is also the case with some of our analyses (10% of the analyses are >90% concordant), but whether a detrital population has only one age mode is dependent on the particular sedimentary system, so it is not universally true that there will be only one array of discordant analyses. We do not agree that detrital data ought to conform to one discordia array, when multiple age populations are expected in detrital samples from most tectonic settings (i.e., Govin et al., 2018 and most studies on detrital U-Pb geochronology). In Rösel et al. (2019), the detrital rutile grains are concordant and interpreted to be sourced from high grade metamorphic rocks, which could indicate a lithologic control on discordance.

Finally, with regards to our study, we used the U-Pb dates combined with protolith information and temperature from trace elements to inform provenance interpretations. In Section 7, we demonstrate that the resulting U-Pb age peaks correspond to plausible sedimentary sources that match the timing of metamorphism, protolith, and temperature of those sources. Therefore, we maintain that our dataset provides meaningful provenance information, and that the conclusions are supported by the data.

From the premise that meaningful interpretations can be made from initially discordant data, we wanted to explore, *"do different discordance filters influence the resulting age spectra and provenance interpretations or not?"* (lines 148-149), in other words, *how discordant is 'too discordant'?* Answering this question requires identifying a metric or threshold for 'too discordant.' We approached this by comparing the uncorrected, [207]Pb-corrected, and [208]Pb-corrected dates. The unfiltered, uncorrected dates yielded a large, unimodal age peak that is useless for interpretation (Figure 5). In the comparison of [207]Pb corrected dates, the age difference *"is less than 1% for analyses less than 60% concordant and less than 5% for analyses 60–40% concordant"* (line 295) (Figure 6). This is why we chose to include all analyses up to 60% discordant (above 40% concordant). In a separate comparison, we binned the [207]Pb- and [208]Pb-corrected dates by their percent discordance (Figure 5, see also Figure 4). Rather than zooming in on the < 500 Ma ages, here we show a modified Figure 5 (Figure R5) with labeled KDE peaks (using detritalPy; Sharman et al., 2018). There is variability in age modes and their amplitude between some of the discordance bins (which we discussed in lines 293-304). For example, the ~190 Ma peak is slightly older at 200 Ma in the 60-40% concordant bin. Between all of the concordance bins, the main age modes are generally at 95 Ma, 190 Ma, 310 Ma, and 580 Ma. We disagree with the Reviewer's characterization that *"the comparison of the KDEs distributions of Fig 5, bottom panel indicates significant differences."* The similarity between the [207]Pb-corrected, and [208]Pb-corrected dates for grains 100-40% concordant is shown

in Figure 5 as KDEs and CDFs and in the statistical comparisons in supporting information (Dataset S3 in Mueller et al., 2023). We view the similarity between the [207]Pb-corrected and [208]Pb-corrected dates for grains 100-40% concordant as a positive check on the discordance filter chosen here. Therefore, we maintain that reliable provenance interpretations can be made from data above 40% concordant in our specific dataset (Section 5.1).

Regarding potential bias in data reduction and filtering, it is possible that the rutile analyses excluded due to low U, low Pb, inclusions, or high 207/206 uncertainty could impart a bias on the age distributions. It is difficult to demonstrate this one way or another as we cannot know the age distribution of excluded grains. They were excluded precisely because their age information was not usable. To reduce bias, we aimed to include as many analyses as possible by including discordant data. The effects of a strict discordance filter are shown in Figure 4. If we had applied a 20% discordance filter, as is common in detrital zircon work and less common in detrital rutile (Shaanan et al., 2020), the results would be only the 100-80% concordance group (which constitutes only 65 analyses). Although, we define concordance here as distance to concordia along the discordia (after Vermeesch, 2021) rather than the relative age difference between the $^{206}$Pb/$^{238}$U and $^{207}$Pb/$^{206}$Pb dates. The 100-80% concordance group has similar dominant age modes as our preferred 100-40% concordance group; however, the KDE amplitudes are different, likely related to $n$. This does not change the resulting interpretations in our dataset, but this may be important for other datasets for which a stricter concordance filter alters date distributions and/or peak amplitude is important (see also lines 293-304).

*Figure R5 (next page). Distributions of U-Pb dates of all samples together displayed as kernel density estimates (KDEs) with labeled age peaks, and cumulative distribution functions (detritalPy; Sharman et al., 2018). Both uncorrected and corrected data are displayed. The $^{207}$Pb-corrected data are separated into concordance bins. Note that the KDEs are not relative plots. The main age modes present throughout the date distributions are ca. 95 Ma, 190 Ma, 310 Ma, and 580 Ma.*

[Figure]

**4. Additional line by line comments, excluding typographic comments**

- Line 40: Detrital rutile mineralogy and geochemistry have been routinely applied to constrain provenance for almost two decades, rutile U-Pb chron. for a decade.

We agree and included detrital rutile in our summary of detrital geochronometers in lines 62-65.

- Line 40: I suggest to rewrite the introduction focusing on detrital rutile as a provenance tracker and a complementary proxy to other single grain and/or bulk techniques. Reference to the theory of plate tectonics or the evolution of sedimentary provenance studies since the 70s is off topic.

*The revised manuscript will change from a focus on new workflows to detrital rutile provenance, suture zone settings, and Anatolia geology.*

- Line 45: Sedimentary provenance includes detrital geo- and thermo-chronology of a wide range of accessory minerals as established techniques, not only mineralogy and geochemistry of sediments.

We agree and included a summary of detrital geochronometers in lines 62-65.

- Line 50: In 2023, detrital geochron (especially of zircon) is by far an established provenance tool. Non relevant terminology (classic as opposed to... modern?) should be avoided. Single mineral approaches are not elevated relative to bulk methods. In fact, single and bulk methods can be complementary in constraining provenance. The choice of method(s) depends on the scientific question.

We agree and did not mean to imply that bulk methods are outdated, as we use these methods ourselves. *We will rephrase.*

- Line 55: Zircon is an ideal mineral provenance proxy due to i) its widespread occurrence in crustal rocks (certainly not a limitation) ii) its chemical and mechanical resistance to geologic and sedimentary processes. Please avoid referrring to "the zircon problem", which is unjustified and misleading. Rutile occurs in a narrower range of crustal rocks compared to zircon, but that does not mean that a "rutile problem" should be invoked.

  - Line 60: Sedimentary provenance based on one technique (any technique) can be incomplete, or not. It depends on the scientific question(s) being addressed. An interpretation based on rutile only could also be incomplete. Please delete or rewrite.

  - Line 65: There is no such "zircon problem". Avoid using such terminology. We could likewise refer to "any mineral" problem.

  - Line 70: Please delete. Detrital rutile is an established proxy and there is no such problem to overcome.

*We will clarify that zircon is an excellent provenance proxy due to the reasons stated.* The rise in the application of other detrital phases has revealed quite well that zircon does a good job in reconstructing provenance, but often does not capture the entire provenance picture (e.g., Moecher and Samson, 2006; Hietpas et al. 2010; Gaschnig, 2019). We don't find it misleading to characterize this as a "zircon problem" as many studies are reporting to reconstruct provenance with zircon alone even though zircon is primarily sourced from felsic to intermediate igneous rocks, thereby underrepresenting mafic rock types in the source, for example. Pereira and Storey (2023) refer to this as "zircon's blind spots" (pgs. 3, 4). However, we did not mean to imply that rutile should replace zircon as the only provenance tool used, but rather used as a complementary dataset (line 20). We agree that bulk and single mineral approaches ought to be used together to reconstruct a full provenance history.

- Line 70: Mention the metamorphic (and igneous) rocks where rutile can form, with references, and why it can be commonly found in sedimentary rocks. Metamafic and metapelic (rutile) are terms introduced by Meinhold (2010, ESR) to broadly categorize rutile provenance.

We believe that we have already included this information in our short introductory review: *"Rutile can form in metamafic and metapelitic rocks across a range of P-T conditions, therefore, detrital rutile is especially advantageous when tracking sediment input from greenschist to eclogite facies sources (e.g., Zack and Kooijman, 2017)."* (lines 69-72).

- Line 70: MAIN COMMENT "U-Pb closure temperature" is not correct as the notion of closure refers to volume diffusion of Pb.  Please cite the work by Cherniak, Contrib Min Pet 2000 and other studies showing how the open-system vs close-system behaviour of radiogenic Pb is better

described by the notion of partial retention zone and on which natural parameters is dependant. The review by Smye et al 2018, Chem Geol is a good source of references.

- o Line 75: This statement is not entirely correct. First cycle detrital rutile can track cooling following any thermal event capable of resetting the U-Pb clock in rutile. Please note that: i) any (high-grade) metamorphic unit must be exposed to surface before being eroded; ii) when cooling is "slow" the detrital rutile U-Pb date can substantially post-date the "age" of the metamorphic event. Please amend the introduction to include examples (e.g Flowers et al 2005 Geology, fig.2 where zircon is 110 Ma, rutile 75 Ma in basement crystalline rocks; other examples in Fig. 4 of Bracciali 2019, Geosci.)

The idea of closure temperature is an oversimplification, but is still a widely used term in the thermochronology community that will be familiar to most readers of this manuscript. The original concept of the closure temperature concept by Dodson (1973) clearly includes the existence of a partial retention zone (Dodson 1973, Fig. 1). The closure temperature defined there is a construct, whose dependencies and limitations are widely known. We argue that it is still convenient and acceptable to use this term and concept to compare the general temperature sensitivity of different chronometers. As pointed out by the reviewer, a given chronometer is sensitive (among other things) to a temperature range that is determined by the diffusion kinetics of the radiogenic isotope, in this case Pb. In this way, thermochronometers have a temperature range where child products are partially retained, the partial retention zone, which is sensitive to factors such as grain size and cooling rate. The U-Pb date is reflective of the interplay between the kinetics of diffusion and/or annealing and accumulation rates. We wrote, *"rutile U-Pb dates correspond cooling from the most recent medium to high-temperature metamorphic event that exceeded the closure temperature"* (lines 72-73); **the revised manuscript will clarify that slow cooling rates can produce U-Pb dates significantly younger than the timing of metamorphism (e.g., Möller et al. 2000). This is relevant to our dataset as rutile U-Pb dates around 190 Ma are 10-25 Myr younger than estimates of peak metamorphism in the Karakaya Complex. We will mention this in the discussion.**

- Line 80: Alternative correction approaches are not acknowledged in this manuscript. Bracciali et al (2013 Chem Geol; 2016 ESR) corrected only the discordant data using the terrestrial Pb-evolution model of Stacey and Kramers (1975) and no Pbc correction applied to the primary reference material (containing negligibile common Pb). Please include

We discuss the use of the Stacey and Kramers (1975) terrestrial Pb evolution model to correct discordant data in the $^{208}$Pb correction and $^{207}$Pb correction sections (Section 2.3.1 and 2.3.2, respectively) and have not applied any corrections to the reference materials.

- Line 90: MAJOR COMMENT. This detrital rutile synopsis is only based on geochemistry of rutile (and derived thermometry). No mention to U-Pb dating of rutile (by LA and ID-TIMS), in a manuscript submitted to G-Chron. Please rewrite and inlcude relevant literature.

This is a good point. We discussed rutile geochemistry in the overview of Section 2.1 and common Pb corrections in Section 2.3 but did not include a discussion of U-Pb dating methods. **The revised manuscript will include a short overview of rutile U-Pb.**

- Line 120: This is not necessarily the case. A rutile with a low U content, but "old" (e.g. Ga), having accumulated enough radiogenic Pb could be easier to date than a "young" rutile (e.g. few Ma) richer in U. Additionally, the lowest measurable radiogenic Pb signal intensities (affecting the precision of the isotope ratios) will vary depending on the LA technique of choice (quadrupole, vs SC-SF- vs MC-SF ICP-MS).

Yes, **the revised manuscript can include statements that both low U and young rutile can be challenging to date.** We already discussed how instrumentation can affect signal intensity in our discussion of U-threshold filtering (lines 305-320).

- Line 120: this is not correct: high U rutiles could be discordant because of of a high relative Pbc content

We show in Figure 7 that rutile above 4 ppm U, the high-U rutile, have a higher proportion of concordant analyses (n=60/68, 88%) than low U rutile. Yes, there are several high U rutile that are discordant (n=8/68, 12%) due to high Pbc (Figure 7b). We would expect that this proportion might vary across datasets. ***We will clarify in line 120 that screening low-U rutile produces a higher proportion of analyses with acceptably high U and Pb signal intensities and seems to result in a higher proportion of concordant analyses.***

- Line 125: The omission or inclusion of low-U rutile does not "make sense" depending on the geological setting: either it is deemed a valid data handling approach, or not. Filtering on U content (in essence rejecting analyses which are expected to fail) is equivalent to discarding analyses where the very low-intensity measured signals would result in poorly determined isotopic ratios with an extremely large uncertainty (e.g. tens %, 2s), since this would make the corresponding dates hardly usable for any provenance interpretation. The application of such filtering before any common Pb correction (as appropriate) must be discussed.

We are uncertain of the exact reason why the published studies used a U threshold and gave them the benefit of the doubt by assuming that they had a valid reason for doing so. The fact is that such datasets exist, and for the region of our study constitute half of the publications (2 of 4). One idea is that they were not expecting metamafic sources and therefore expected the majority of rutile to have high U. Here, we show that the U thresholds used are too high, as rutile with U < 4-5 ppm can yield dates with acceptable precision and concordance (see Section 5.2, Figure 7). And setting a specific ppm level as threshold is not good practice as a general recommendation because of the aforementioned dependence of LOD on instrumentation and analytical parameter choices. Therefore, filtering on U content is not "rejecting analyses which are expected to fail" as we demonstrate that they do not all fail. In fact, excluding rutile below 4 ppm U biases the dataset (discussed in Section 5.2). Regarding the removal of analyses with large uncertainties in the isotopic ratio: *"214 analyses were excluded for 207Pb/206Pb error above 20%"* (line 255). The analyses that were removed generally had low uranium concentrations: maximum of 8.9 ppm, average of 0.5 ppm, minimum of 0.02 ppm (Dataset S1). See also reply to point #1 above.

- Line 255: clarify which control goals were used

This is stated in the same sentence: *"Even with the modified protocol, a significant number of analyses did not meet quality control goals: 686 of 1,278 analyses were excluded due to anomalous (spiky) patterns in raw signal intensity and a further 214 analyses were excluded for 207Pb/206Pb error above 20%, leaving 378 analyses remaining (30%)."* (lines 253-256). The quality control goals are acceptable signal intensity, both high enough U and Pb for data reduction (***we will add this statement***) and smooth, non-spiky, signal with 207/206 uncertainty below 20% (2s). ***We will clarify that the 20% uncertainty is at the 2-sigma level.***

- Line 270: Trace elements in rutile can easily sum-up to 1-2%

The TiO2 internal standardization varies across the literature, with TiO2 normalization commonly 100 mass-% (e.g., Rösel et al., 2019; Plavsa et al., 2018), 99 mass-% (e.g., Ewing et al., 2013), or 98 mass-% (e.g., Hart et al., 2018). The TiO2 internal standardization value and reproducibility of standards are not consistently reported, so it is hard to assess. The choice of TiO2 normalization to values between 100 and 98 mass-% is below the reproducibility of reference materials and does not significantly influence the calculated trace element results.

Figure R6 displays the trace element results of the three reference materials for the four analytical sessions. The trace element results are within 10% for all of the main elements discussed in the paper: Cr, Nb, and Zr. For the secondary standard GSC-1G, all elements are within 10% of the published values

except for Sn and Ga. The R10 rutile reference material displayed internal heterogeneity in trace element composition, which we noticed when comparing the trace element results with the laser ablation spot coordinates. This is also reflected in the range of trace element compositions reported for R10 in the GeoReM database (http://georem.mpch-mainz.gwdg.de/). The R10 results are within 10% for many elements but the offset is much higher for some elements. We note that all of the R10 results are within the range of reported values from the GeoReM database. Reproducibility of around 10% is consistent with the literature. For example, Ewing et al. (2013) report that trace element measurements generally reproduce reference materials to within 11%, but there are other reports of concentrations varying up to 30% for some elements (Plavsa et al., 2018). ***The revised manuscript will include a discussion of reproducibility in the main text and appendix.***

[Figure]

*Figure R6 (previous page). Plots of percent deviation of the trace elements for reference materials GSD-1G, GSC-1G and R10. The data were calibrated using GSD-1G and Ti as an internal standard element. Error bars show the standard deviation.*

- Line 315: It is expected that when a minimum signal intensity (CPS or V) has to be set to filter the data, such a threshold will vary depending on the sensitivity of the technique (Q-ICP-MS, SF-SC, SF-MC).

Yes, we believe we stated this in lines 305-320, that the U concentration filter does not make sense as it is instrument and parameter dependent. To clarify again, we are not advocating for such a threshold, but it exists in the published literature on detrital rutile and its potential effects need to be discussed. *We will check that this is clear in the revised manuscript.*

**References Cited**

Andersen, T.: Correction of common lead in U–Pb analyses that do not report 204Pb, Chem. Geol., 192, 59–79, https://doi.org/10.1016/S0009-2541(02)00195-X, 2002.

Bracciali, L., Parrish, R. R., Horstwood, M. S. A., Condon, D. J., and Najman, Y.: UPb LA-(MC)-ICP-MS dating of rutile: New reference materials and applications to sedimentary provenance, Chem. Geol., 347, 82–101, https://doi.org/10.1016/j.chemgeo.2013.03.013, 2013.

Caracciolo, L., Ravidà, D. C. G., Chew, D., Janßen, M., Lünsdorf, N. K., Heins, W. A., Stephan, T., and Stollhofen, H.: Reconstructing environmental signals across the Permian-Triassic boundary in the SE Germanic Basin: A Quantitative Provenance Analysis (QPA) approach, Glob. Planet. Change, 206, 103631, https://doi.org/10.1016/j.gloplacha.2021.103631, 2021.

Chew, D. M., Sylvester, P. J., and Tubrett, M. N.: U–Pb and Th–Pb dating of apatite by LA-ICPMS, Chem. Geol., 280, 200–216, https://doi.org/10.1016/j.chemgeo.2010.11.010, 2011.

Chew, D. M., Petrus, J. A., and Kamber, B. S.: U-Pb LA-ICPMS dating using accessory mineral standards with variable common Pb, Chem. Geol., 363, 185–199, https://doi.org/10.1016/j.chemgeo.2013.11.006, 2014.

Dodson, M. H.: Closure Temperature in Cooling Geochronological and Petrological Systems, Contributions to mineralogy and petrology, 40, 259–274, 1973.

Ewing, T. A., Hermann, J., and Rubatto, D.: The robustness of the Zr-in-rutile and Ti-in-zircon thermometers during high-temperature metamorphism (Ivrea-Verbano Zone, northern Italy), Contrib. Mineral. Petrol., 165, 757–779, https://doi.org/10.1007/s00410-012-0834-5, 2013.

Faure, G.: Principles of Isotope Geology, 2nd Edition., Wiley & Sons, Inc., 608 pp., 1986.

Frei, D. and Gerdes, A.: Precise and accurate in situ U–Pb dating of zircon with high sample throughput by automated LA-SF-ICP-MS, Chemical Geology, 261, 261–270, https://doi.org/10.1016/j.chemgeo.2008.07.025, 2009.

[revised manuscript text omitted]

---

## Author Comment (AC3)

Response to Reviewer #3's comments on manuscript egusphere-2023-1293

"An expanded workflow for detrital rutile provenance studies: An application from the Neotethys Orogen in Anatolia"

Megan A. Mueller, Alexis Licht, Andreas Möller, Cailey B. Condit, Julie C. Fosdick, Faruk Ocakoğlu, and Clay Campbell

November 15, 2023

We appreciate the thoughtful reviews of the 3 referees. First, we summarize the broad themes from the three reviews before responding in detail to Reviewer #3 below. The reviews critiqued (1) the novelty of the study, (2) the number of U-Pb analyses discarded during data reduction, (3) the potential bias of discarding data and the validity of interpreting discordant U-Pb analyses, and (4) the apparent lack of novelty and complexity in geochemical data interpretations. Regarding these points, the goal of the manuscript was to provide a transparent workflow for detrital rutile geochronology using new data from sedimentary basins in Anatolia. We acknowledge that it was confusing or misleading to present this work as a revised workflow. We intended to emphasize that we found a paucity of methods papers that provide a straightforward approach. Although we did not claim to present new workflows with the geochemical data, this was perhaps unintentionally implied with the title. We have presented as much information as we could squeeze out of our particular geochemical dataset. The revised manuscript will scale back statements on new workflows and instead refocus the title and introduction on suture zone settings and Anatolian geology while maintaining the broad overview of detrital rutile provenance and the details of our methods.

We also acknowledge that it is surprising to see the number of data discarded during data reduction. However, we contend that this is a common practice with detrital U-Pb geochronology in common Pb-bearing minerals. We are not the first to discard a significant number of analyses; we have found this practice in many published detrital rutile datasets, although it is not discussed much in the literature. We further expand on this point in our detailed reply and will add this context to the revised manuscript. This manuscript gives precedence for papers to be transparent in data reporting—including the number of grains analyzed and criteria for rejection—as well as examination of the full dataset in Tera-Wasserburg diagrams. Our manuscript provides an opportunity to show the consequence and potential value of the full dataset, which is relevant to others working with this type of data. While the use of common Pb-bearing minerals is common in some labs and geographic settings, the application of these tools is still far behind detrital zircon geochronology, for which there is a well-established global framework. We have encountered many detrital geochronology users who want to add detrital rutile to their toolkits but are still uncertain in how to collect and interpret these data. Here, we present a complicated detrital rutile U-Pb dataset that can serve as an example for how to treat and interpret complex, yet potentially meaningful, discordant data.

Reviewer #3 provided a thoughtful review of our manuscript that highlighted several ways to improve the manuscript that include scaling back statements on 'new workflows' and altering the framing of the manuscript, clarifying the U-Pb data reduction and rejection of U-Pb analyses, and expanding the analysis of the detrital rutile trace element geochemistry. The review includes several main suggestions for improvement that we will implement in the revised manuscript. Reviewer #3 included many in-text typographic and citation comments that will be incorporated into the revised manuscript.

Reviewer #3 provided comments in the review and line-by-line comments in the text. We address the comments below by theme. Reviewer #3's comments are included below in black text. Our response is in purple text and the specific changes we will implement are highlighted in *bold, italic purple text*.

1. **Comments regarding the framing of the manuscript**

- My suggestion, if you will, is that you make a full twist and turn of your manuscript. Place the focus on using detrital rutile in reconstructing convergent margin evolution, using Anatolia and your sample set. I recommend you acquire more data (more grains) to overcome the reduced number of dates you have from each individual sample. Focus first on your case study, then move on into the rutile dating challenges affecting provenance, and how your approach can enhance the level of information we obtain from U-Pb rutile dating and from convergent settings. This requires major rewriting, but you can reuse most of your very nice figures. Unfortunately, I do not recommend publication in its current form, but you should feel encouraged to a re-submission.

- Title: After reading the manuscript a few times, I have my doubts about the suitability of this title. You have explored a couple things in your geochronology that improve using rutile ages in provenance studies.

- Line 25: I think that the main focus should be this one, and rutile data treatment would be a second-level objective.

The goal of the manuscript was to provide a transparent workflow for detrital rutile geochronology using new data from sedimentary basins in Anatolia. We acknowledge that it was confusing or misleading to present this work as a revised workflow. We intended to emphasize that we found a paucity of methods papers that provide a straightforward approach. ***The revised manuscript will scale back statements on new workflows and instead refocus the title and introduction on suture zone settings and Anatolian geology while maintaining the broad overview of detrital rutile provenance and the details of our methods.***

***Following the comments of Reviewers #1 and #3, we will move away from phrases like 'new workflow.' The revised manuscript will have an updated title and introduction that is oriented toward Türkiye and suture zone settings. We will emphasize how detrital rutile is a useful complement to detrital zircon in accretionary collisional settings like Anatolia, where many sediment sources are obducted ophiolites and metamorphic terranes. Additionally, we will emphasize how detrital rutile can disentangle signatures of sediment recycling. This was not discussed much in the manuscript but can be expanded in the revision with the more nuanced geologic setting. We will keep the discussion of Pb corrections and uranium concentration.***

2. **Comments regarding U-Pb analytical protocol, data reduction, and rejection of data**

- My main concern related to U-Pb data acquisition/treatment came from the rather large number of discarded rutile grains, due to inclusions, lamellae, etc. In the end, they discarded many rutile grains. How did it affect or biased the final dataset? In these circumstances, how robust is it to average all samples and plot them together, especially in a dynamic sedimentary environment? What about source variability through time weighted on each of your sample populations (n)? Figure A3 is clear in illustrating the issue here.

  o Figure A3: very few grains per sample. Can we really average all these data into one KDE? You showed they have very differing ages... I am afraid that you have a major issue here and you need a bit more data. Another challenge in detrital rutile geochronology...

Reviewers #2 and #3 raised concerns about the number of analyses discarded during data reduction. The workflow that we present includes the analyses of all detrital rutile grains in a sample. *"Rutile grains were handpicked; all rutile grains were picked from most samples, except for samples 16SKY26, 16SKY42 and 17OZK05 for which 260–320 grains were selected"* (lines 238-239). Part of the

reason that provenance interpretation is not possible at the sample level is because of the low rutile yield in some samples. For example, sample 18DMN01 only had 5 rutile grains, all of which were analyzed and 4 satisfactory dates were obtained. First, we suspect that this could be a feature of local geology (i.e., metamorphic sources were not exposed at the surface at the time the sample was deposited), however, in the future we will collect larger samples to try to increase heavy mineral yield. Second, low yield could be due to the exclusion of analyses during data reduction, for reasons including poor signal intensity and large uncertainty in the $^{207/206}$Pb ratio. Anomalous signal intensity was likely due to a combination of very low uranium concentrations, "*elemental heterogeneity from ablating into small inclusions and/or lamellae, and inhomogeneities due to micro-cracks with different element/isotope composition*" (lines 257-258).

The exclusion of analyses during data reduction is not unique to this study but represents a larger problem with large-*n* detrital rutile provenance work. Many published studies discuss analyzing a larger number of grains than are presented—discarding analyses—due to low U and/or low radiogenic Pb content (e.g., Bracciali et al., 2013; Caracciolo et al., 2021). For example, Shaanan et al. (2020) discard 60% of their detrital rutile U-Pb data due to discordance. Similarly, Reviewer #1 points to the study by Caracciolo and co-authors (2021) that analyzes 712 detrital rutile grains without a U-filter, yet, after discordance filtering, only 347 grains remained (48%) (from what we can tell as the data is not available online). Additionally, Caracciolo et al. (2021) did not have enough rutile ages per sample to discuss sample-by-sample provenance interpretations, which we also experienced with our dataset. This points to a larger problem in trying to scale up detrital rutile to large-*n* provenance applications. For this reason, we wanted to confidently include as many U-Pb analyses as possible in our interpretations, which led to the exploration of U-Pb discordance.

To address this comment, *the revised manuscript will emphasize that the rejection of analyses during data reduction is not a unique limitation of this study, but typical of many detrital rutile studies. We will clarify that this seems to be common in studies that have attempted large-n detrital rutile U-Pb. The original text emphasized the role of inclusions and lamellae in potentially contributing to poor signal intensity patterns and did not clearly emphasize that, in addition, low signal intensity could be from low U and/or low radiogenic Pb contents. The revised manuscript will also include low U and low Pb as a potential cause of poor raw signal intensity. As discussed below, we will also include a data treatment section of the Appendix that includes a figure with examples of signal intensity patterns in the unknowns.*

- Line 235: we assume from the non-magnetic fraction of the heaviest fraction, but you should detail it here, please.

    o Line 235: random HM minerals? Did you also hand-picked for those or you just decided you wouldn't pick for these two? Why?

    o Line 240: if you hand-picked rutile grains except for three samples, did you do this for all samples? Or did you image all your grains to analyse their textures, the occurrence of inclusions and double check their mineral id? It would be appropriate to include your SEM imaging conditions as well, as you would do for LA-ICPMS or EPMA.

The samples analyzed were previously processed for detrital zircon U-Pb and Hf analysis (Mueller et al., 2019, 2022; Campbell et al., 2023). In order to extract detrital rutile, all heavy mineral fractions from post-water table separation steps were recombined and reprocessed in the following manner. Samples were first separated in methylene iodide to collect the dense fraction. The Frantz magnetic separator was set to 20° side slope and 20° forward slope such that rutile grains were separated into the 0.3 to 0.7 amp. fraction (Rosenblum and Brownfield, 2000). Rutile grains were handpicked with a Leica M205C binocular microscope using transmitted and polarized light. Rutile grains are red-brown-yellow colored in reflected light, red to opaque in plane polarized light, and displayed a resinous to vitreous luster; grains are well rounded to euhedral and many display twinning characteristic of rutile's

tetragonal crystal system and striations parallel to the long axis. Three samples—16SKY26, 16SKY42 and 17OZK05—yielded hundreds of rutile grains and we handpicked 260–320 rutile grains from each sample. For samples with a small quantity of heavy mineral grains, rutile was picked from all 0.3 to >1.5 amp. magnetic fractions. The low yield of rutile partially contributes to the low-*n* of the samples. Then, the grains were mounted in epoxy, polished and imaged in the SEM. We used EDS to confirm that the grains that were handpicked were in fact $TiO_2$. ***We will clarify this methodology in the revised manuscript.***

- Line 245: I find this rather harsh, when you have a fine laser. This will increase your DHF. Why not being more gentle, ablating at a lower frequency? You also went too depth unnecessarily

This comment is regarding the 10 Hz rep rate. The laser system used has a fast washout: 4-5 magnitudes of signal in <1 second. Using a lower frequency would result in a sawtooth signal pattern and lead to aliasing problems. We decided against signal smoothing and lower frequency to avoid "smearing" out the effect of inclusions or exacerbating the potential effect of surface common Pb.  For the relatively large spot used, the pit depth is moderate, also due to the moderate laser energy used.  10 Hz frequency has been widely used for U-Pb geochronology of accessory minerals, including rutile (e.g., Kooijman et al. 2010; Zack et al. 2011). The conditions were chosen to allow analyses of relatively low U and Pb rutile, as anticipated for relatively young rutile received from low to medium grade metamorphic rocks with a high likelihood of mafic protolith.

- Line 245: I understand why, but in most detrital rutile samples this may be a very large spot, where you most likely will hit micro-inclusions. For a better workflow, in my pov, we should aim for 35-40 um, even though we lose signal. This should be discussed, perhaps not here, but in the discussion section

Yes, a larger spot size gives a higher signal, which is better for grains with potentially low U or low Pb concentrations. The tradeoff is potentially hitting more inclusions, we are aware of this. See previous comment, we aimed to optimize for relatively young, low U and Pb  rutile, because it was obvious that many source rocks would be ophiolite units with largely mafic rock types.  In areas with higher input of older or higher grade metamorphic rocks conditions may be chosen differently.

- Line 250: include the published or accepted ages for all these RM here. Also, you need to state if you used a sample bracketing approach, and of how many unknowns interspersed with how many primary and secondary RM.

***The revised manuscript will include the published ages for the reference materials:*** R10 (1091.6 ± 3.5 Ma by TIMS, 2s abs.; Luvizotto et al., 2009), Wodgina (2845.8 ± 7.8 Ma by TIMS; Ewing, 2011), Kragerø (1085.7 ± 7.9 Ma by TIMS; Kellett et al., 2018), 9826J (381.9 ± 1.1 Ma by TIMS; Kylander-Clark, 2008), and LJ04-08 (498 ± 3 Ma by LA-ICP-MS; Apen et al., 2020). ***The revised manuscript will clarify our sample-standard bracketing approach. This is stated briefly in Table A2 but will be expanded.*** For U-Pb analyses, the analysis of 5-8 unknowns was followed by 2 standards, the primary standard R10 and one of the secondary standards. For trace element analysis, the analysis of 5-10 unknowns was followed by analysis of 2 standards, the primary standard GSD-1G and one of the secondary standards.

- Line 250: what splines did your use?

For the drift correction, we used iolite's SplineSmooth5 for all analytical sessions. However, in response to Reviewer #2's comments regarding standard reproducibility, we compared the effects of weighted linear fit, SplineSmooth5 and SplineSmooth9 curves for drift correction. The choice of spline had little effect on the final age of the reference materials but impacted the MSWD. We prefer the weighted linear fit as this curve reproduces the secondary standard ages and brings the MSWD closest to 1 for each standard. ***The revised manuscript will include the results that were reduced using the weighted linear fit drift correction. All figures and tables in the main text and supplement will be***

*updated. We do not anticipate that this change will produce significant changes to the results or interpretations.*

- Line 255: it is not clear how it changed from the previous one, as you don't highlight it.

*We will remove this statement and only focus on the protocols used here.*

- Line 255: which signal? in all channels? It is natural for rutile to have small mineral inclusions. For how long, in s, did those spikes affected your signals? The entire ablation duration? At the start, end? We all discard a few analyses every now and again, but I was a bit surprised with these numbers. I think that this deserves further consideration, so you should include a "data treatment" section in the appendix, where you show print screens of your signals and give examples of analyses you excluded due to these spikes…

The U-Pb data reduction was performed in *iolite* (Paton et al., 2011). We monitored signal intensity by examining $^{206}$Pb, $^{207}$Pb, $^{238}$U, $^{232}$Th, 206/238 and 207/206 ratio channels. It was easy to spot inclusions by monitoring these channels. In some instances, the signal of an inclusion was short enough that the integration window could be shortened to exclude the inclusion. In other cases, the non-inclusion signal could not be isolated and the entire analysis was discarded. Some grains were excluded due to too low U or Pb signal. Anomalous patterns in 207/206 ratio resulted in either shortening the integration window or excluding the analysis. ***The revised manuscript will include an appendix section that details the data treatment. We will follow Reviewer #3's suggestion to include a figure with a few examples of good and bad signals.***

- Line 270: which one was used as primary and which ones as QC ? You should provide Ti concentration of your primary material

  o  Line 270: how did your secondary rm behaved as QC? Which elements are within 10% error and which ones fall out?

For trace element analysis, GSD-1G was used as the primary standard. Figure R1 displays the trace element results of the three reference materials for the four analytical sessions. The trace element results are within 10% for all of the main elements discussed in the paper: Cr, Nb, and Zr. For the secondary standard GSC-1G, all elements are within 10% of the published values except for Sn and Ga. The R10 rutile reference material displayed internal heterogeneity in trace element composition, which we noticed when comparing the trace element results with the laser ablation spot coordinates. This is also reflected in the range of trace element compositions reported for R10 in the GeoReM database (http://georem.mpch-mainz.gwdg.de/). The R10 results are within 10% for many elements but the offset is much higher for some elements. We note that all of the R10 results are within the range of reported values from the GeoReM database. ***The revised manuscript will clarify the primary and secondary reference materials and include a discussion of reproducibility in the main text and appendix.***

[Figure]

*Figure R1. Plots of percent deviation of the trace elements for reference materials GSD-1G, GSC-1G and R10. The data were calibrated using GSD-1G and Ti as an internal standard element. Error bars show the standard deviation.*

**3. Comments regarding the trace element dataset**

- The geochemistry is used, but PCA is not really adding anything new or rather interesting. I do not see the purpose of keeping it in its current form. I am afraid that by stating strongly very early on that you would be "expanding the workflow" of rutile in provenance studies, but then not combining it well with TE systematics is misleading.

- Section 2.1. Provenance: Don't place all the focus on Nb-Cr and Zr thermometry. You can be more concise in your sentences and ideas, and then you could highlight how other more recent tools can be effectively used in provenance studies (see my comments in the pdf). I also find that this subsection's title is not suitable. You should re-think better how 2.1 related to the other subsection under section 2 and propose an improved structure and matching headings. You have provenance in 2, and then a "synopsis", followed by "challenges" that actually just relate to U-Pb. So, in my view you are emphasising the application of detrital rutile geochronology as a provenance tool. This is fine, but the subheadings need to clarify this.

- Section 6.2. Careful with rushing into conclusions. While it may be true that in your dataset your U<4ppm is vastly metamafic, this does not necessarily imply lower T. For a robust assessment of lower Ts, you need to rely on your metapelitic detrital grains. Rephrase.

- Line 355: hmmm if they are metamafic, you increase the probability of not having the boundary conditions to effectively use Zr as a thermometer. So lower Zr in a mafic rutile does not equal lower T...

- Section 6.3 If you are not exploring your data further, I would remove it, as it is not bringing anything really new, and it does not add to any relevant information to your main objective. Since they are detrital grains, I would have difficulty to saying much more.

  - If you decide to follow my suggestion of major rewriting, you could do a bit more work on detrital rutile trace element geochemistry, try to interpret their chemistries a bit more, and then it would be fine to use the PCA as a starting point of such a subsection.

As stated above, although we did not claim to present new workflows with the geochemical data, this was perhaps unintentionally implied with the title. We have presented as much information as we could squeeze out of our particular geochemical dataset.

Reviewers #1 and #3 raised points about the use of our geochemical dataset. We used the detrital rutile trace element dataset in several ways. We confirmed that analyzed grains were rutile using Cr, V and Zr (Appendix A). The Cr and Nb concentrations are used to discriminate mafic and pelitic protoliths and Zr concentration is used to determine Zr-in-rutile temperatures, with the caveat that this assumes the necessary buffering assemblage. The aforementioned applications only use 1 or 2 elements each. To evaluate the suite of trace elements, we used principal component analysis. Dimensionality reduction methods like PCA and tSNE are useful for evaluating clustering in large, multivariate datasets (we did not discuss tSNE in the manuscript as the results were similar to PCA). We hoped that these methods would help distinguish detrital populations. However, the PCA results show that *"the variance between [grains] can largely be explained by Hf, Zr, Sn, Cr, V, Nb and Ta. Because Cr, Nb and Ta are protolith dependent (PC 2) and Hf and Zr are temperature dependent (PC 1), the variance in detrital rutile trace element chemistry is best explained by both protolith and metamorphic grade, allowing us to track these two properties of source rocks." (Lines 376-379).* This is valuable information, since we are not aware of other publications trying this approach. The PCA results show that the protolith and Zr-in-rutile sections are already exploring the most salient aspects of the trace element dataset. ***The revised manuscript will clarify the most salient points of the PCA results. We will emphasize that the protolith and temperature sections capture the most important components of the trace element results. Additionally, we will***

*revisit the trace element data to evaluate trends in samples and/or age populations and consider adding spider plots, bar plots, and/or other clustering algorithms as relevant. The revised manuscript will include updated text, figures and supporting information to support any new findings.*

Regarding the Zr-in-rutile temperatures of metamafic grains, we show that the majority of mafic-classified grains have lower temperatures, around 400-500 °C (Figure 9). We did not mean to imply that all metamafic grains have low Zr-in-rutile temperatures. *We will clarify the text.* On the comments about Zr-in-rutile temperatures in mafic grains that point to the requirement for zircon, quartz and rutile to be in equilibrium to use the Zr-in-rutile thermometer, we are aware that this assumption may not hold in mafic rocks. In the absence of zircon and/or quartz, the concentration of Zr is not buffered by the reaction and the calculated temperature is not reliable. In their review, Pereira and Storey (2023) discuss how inclusions in rutile can be used to determine whether rutile grew in equilibrium. We suspect that only a workflow of automated mineralogy with a very small spot size would be able to systematically study all inclusions in a high-*n* detrital study. *We can add a statement in Section 2.1 about the requirement that zircon, quartz and rutile are in equilibrium, which may not hold and is hard to determine in a detrital context.*

**4. Additional line by line comments, excluding typographic comments**

- Sections 2.3.1 and 2.3.2: You should provide at least a short review on the $^{204}$Pb-based correction of common Pb. You can discuss why it is sometimes very difficult to apply it, but omitting it is not satisfactory in a paper meant to cover common Pb corrections.

  - Line 150: what about 204Pb correction? Even if you do not include a section, I think you should add a sentence to explain how it can be done.

Reviewers #1 and #3 both inquired why the $^{204}$Pb correction was not discussed. This was not initially included because (a) $^{204}$(Pb+Hg) and $^{202}$Hg were not measured (high Hg in UHP He gas in the midcontinent US), so we did not perform a $^{204}$Pb correction, and (b) it is reviewed in other publications. Although we will not be able to explore the $^{204}$Pb correction in our dataset, it is a goal of future work, *the revised manuscript will include a short overview of this correction and its application in rutile.*

- Sections 2.3.1 and 2.3.2: I suggest you include a figure with two panels, in one showing a common Pb-bearing analysis and the corresponding corrected age using the $^{208}$Pb correction and in the second the 207. These can the theoretical if you will, but you can add arrows and annotations illustrating the assumptions.

  - Line 190 / Figure 1: actually, it would be beneficial to see the lines of 2s +/- regressing the age in TW space from 6/7 ratios. equally relevant is the impact of these choices in the resulting date.

*Following the comments of Reviewer #2, the revised manuscript will modify the discussion of the $^{207}$Pb correction. We will use an iterative approach to the $^{207}$Pb correction and will include the equations for how to do so in the revised manuscript. This eliminates the original purpose of Figure 1, which shows how to graphically perform a $^{207}$Pb correction and how to calculate discordance. It might make sense to keep Figure 1 in the revised manuscript to demonstrate the effect of common Pb incorporation in pulling an analysis from concordia toward the Pbc composition. We will consider this point in the revised manuscript.*

- Line 70: I think this is a very simplistic sentence for the complexity about the age significance. Since this is partly the focus of your work, why not elaborate a bit more? In recent papers you find discussion about rutile ages all over.

Reviewers #2 and #3 commented on the statement, *"With a U-Pb closure temperature of 490–640°C (Kooijman et al., 2010), rutile U-Pb dates correspond to cooling from the most recent medium to*

*high-temperature metamorphic event that exceeded the closure temperature (Zack et al., 2004b; Zack and Kooijman, 2017)"* (line 72-74). ***The revised manuscript will expand on this statement to better reflect the meaning of a rutile U-Pb date. The revised manuscript will clarify that the U-Pb age is a function of grain size, diffusion kinetics, and cooling rate, and potentially of fluid presence. High peak temperature and/or slow cooling rates can produce U-Pb dates significantly younger than the timing of metamorphism (e.g., Möller et al. 2000). This is relevant to our dataset as rutile U-Pb dates around 190 Ma are 10-25 Myr younger than estimates of peak metamorphism in the Karakaya Complex. We will mention this in the discussion.***

- Section 2.1: I think that the work by Hart et al., 2016 should be acknowledged here as well, with a brief outlook detailed in Pereira and Storey 2023, on the applications of such an approach in rutile provenance studies.

*We will add mention of these papers in our revised Section 2.1 overview of detrital rutile.*

- Line 95: Potentially you could add potential use of rutile TE to constrain oxygen fugacity (using V and H).

*We will add mention of this application of rutile trace elements to the overview.*

- Section 2.2: In all truth, while many of us have been saying that this can generate a bias, no systematic study has been done to actually quantify and test the effect of screening for high U rutile only. I think that you may want to strengthen this point in a few parts of the manuscript, including here.

*We will strengthen the point that our dataset clearly shows the bias of screening out rutile with low uranium concentrations.*

- Line 135-140: in my opinion, this is irrelevant here. Focus on rutile, people reading should know about U-Pb geochronology or should go somewhere else to learn the fundamentals.

*We will consider shortening or removing this overview of U-Pb geochronology of rutile.* We originally included these few lines because we have encountered many detrital geochronology users who want to add detrital rutile to their toolkits but are still uncertain in how to collect and interpret these data. We have seen detrital rutile papers that treat the data like detrital zircon in how they treat discordant data (i.e., applying a very strict discordance filter). It is important to emphasize to the detrital geochronology community the differences in why zircon versus common Pb-bearing minerals are discordant and how to treat these datasets.

- Line 190: what do you consider to be a stricter filter?

*We will change the language of this sentence.* Here we define discordance following the "Stacey-Kramers discordance filter" from Vermeesch (2021). At any discordance threshold, this filter includes more young dates than old (> 1000 Ma) dates due to the change in concordia slope around 1000 Ma.

- Section 5: This is a bit odd. Usually you would have a separate Results section and then you discriminate each type of result as a separate subsection...

*We will reconsider the organization of the Results section in the revised manuscript.*

- Line 350: since this is a results section, you need to explain this data a bit further. Which pressure have you used for these calculations and why? It is not clear if you followed what you reviewed earlier, as no uncertainties in your Ts were reported.
  - Figure 9: calculated for what P?

We calculated Zr-in-rutile temperatures with an average pressure of 13 kbar and an uncertainty of 5 kbar (i.e., 8 kbar to 18 kbar). The uncertainty is calculated as the difference between the Zr-in-rutile

temperature at 13 kbar versus 8 kbar or 18 kbar. This calculated uncertainty is already included in the supplementary file spreadsheet. ***We will clarify this in the methods section.***

- Line 350: arghhh you will find plenty of provenance studies or detrital rutile studies where no such strategy is employed... missing some literature here…

Reviewers #1 and #3 commented on whether 'low-U filtering' is prevalent in the literature. We agree that most labs that analyze detrital rutile do not apply a U-threshold filter. While not a global problem, this is a regional problem. There are 4 published detrital rutile U-Pb datasets from Türkiye (including this study), and 2 of them (Okay et al., 2011; Şengün et al., 2020) only analyze U-Pb on grains with uranium concentrations above ca. 4-5 ppm. This is a regional problem and imparts a bias, which we wanted to address in this manuscript as detrital rutile is still uncommon in Anatolia. The 2 studies that do not use a U-threshold filter but instead analyze all detrital rutile grains (Shaanan et al., 2020; this study) have to discard data due to very low uranium signals (below LOD) and must implement a protocol for evaluating discordance because of common Pb incorporation. For example, Shaanan et al. (2020) discard 60% of their detrital rutile U-Pb data due to discordance. Similarly, Reviewer #1 points to the study by Caracciolo and co-authors (2021) that analyzes 712 detrital rutile grains without a U-filter, yet, after discordance filtering, only 347 grains remained (48%) (from what we can tell as the data is not available online). Additionally, there were not enough rutile ages per sample to discuss sample-by-sample provenance interpretations, which we experienced with our dataset. This points to a larger problem in trying to scale up detrital rutile to large-*n* provenance applications. For this reason, we wanted to confidently include as many U-Pb analyses as possible in our interpretations, which led to the exploration of U-Pb discordance. To address this concern, ***we will change the text to reduce the discussion of U-threshold filtering. The revised manuscript will clarify that U-threshold filtering is currently not a common practice but is used regionally in Anatolia.***

- Line 415: are there KDEs from these units that you could use to compare them with?

Yes, the detrital zircon results are displayed alongside the detrital rutile results as KDEs in Figure 12. ***We will refer the reader to Figure 12 in this sentence.***

- Figure 7: in terms of age, you data actually seems to support that no significant differences are perceptible between populations.  In your 190-200 population you may have a higher proportion of lower U grains, so that is nice, but it seems that there are no missing ages when you compare it with the > 4ppm U grains.

We agree that for our dataset there is not a significant difference in age modes between the U-threshold and concordance filtered data. It would be interesting to see if this holds in other datasets. We want to alert readers to the possibility and encourage them to test this. We addressed this in the text: *"The [U-threshold and concordance] filtering methods produce date spectra with the same dominant modes, yet the amplitude of peaks varies between methods. For example, the 190 Ma mode is more prominent with the concordance filter than with the U-threshold filter. Furthermore, the predominant date modes contain rutile of both metapelitic and metamafic origin (cf. next section and Figure 8). Even though the two filters do not yield different provenance interpretations in this case, most mafic-classified grains have U contents below 4 ppm and are in the 190 Ma population. Hence, the U-threshold filter is likely biasing results toward pelitic sources."* (lines 330-335).

- Figure 7: not statically robust to say much about the data points here...

We agree and have not made any statements or interpretations on the U-Pb dates > 1000 Ma.

- Figure 7: in the green and pink you should state here in the legend no U concentration filtering

***We will update the figure caption and/or legend.***

- Figure 8: include the 2se as error lines or just plot the data as ellipses

      ○  Figure 10: include the 2se as error lines or, even better, as ellipses

We chose not to include the ellipses or error bars in these figures in order to simplify. The U-Pb data is shown with 2s error in Figure 4. ***We will consider updating this in the revised manuscript.***

- Figure 8: you have different dashed lines, so you should clearly attribute each to an author

We included the citations in the figure caption, but acknowledge it is unclear in the figure. ***We will change this in the revised manuscript.***

- Figure 12: to help the reader, you could add bars or arrows to known events in the area. This would simplify our work as we try to interpret the plot and follow your discussion\

***This is a good suggestion, and we will update the figure in the revised manuscript to show the main orogenic events that correspond to the main age populations, Variscan, Cimmerian and Alpine.***

**References Cited**

[revised manuscript text omitted]

---

## Author Response (AR1)

February 19, 2024

To the Editors and Associate Editors:

On behalf of my co-authors, I send you the revised manuscript titled "Navigating the complexity of detrital rutile provenance: Methodological insights from the Neotethys Orogen in Anatolia."

The manuscript investigates U-Pb and trace element data reduction, processing, and common Pb correction workflows using new detrital rutile U-Pb geochronology and trace element geochemistry results from the Late Cretaceous to Eocene Central Sakarya and Sarıcakaya Basins in Anatolia. We use our dataset to demonstrate how to navigate the complexities of natural datasets. We provide recommendations for common Pb correction, discordance calculation and data filtering that are applicable to detrital rutile and other common Pb-bearing detrital minerals. Additionally, to facilitate the standardization of data reporting approaches, we provide open access code as Jupyter Notebooks for data processing and analysis steps, including common Pb corrections, uncertainty filters, discordance calculations, and trace element analysis.

Reviews from the first version of this manuscript (manuscript number egusphere-2023-1293) indicated substantial revisions were needed before acceptance. Three referees and the Associate Editor provided constructive comments that enabled us to significantly clarify and strengthen our original manuscript. The three reviewers critiqued the novelty of the study, the number of U-Pb analyses discarded during data reduction, the potential bias of discarding data and the validity of interpreting discordant U-Pb analyses, and the apparent lack of novelty and complexity in geochemical data interpretations. The reviewers' comments were addressed in our previously submitted responses to reviewers. The Associate Editor's comments are addressed below. The revised manuscript includes all of the changes and revisions indicated in our responses.

The manuscript has been completely rewritten and reorganized. The major changes are summarized here. (1) All reviews questioned the large number of U-Pb data rejected. We contextualize our U-Pb data within the published literature. We demonstrate that the rejection of data during U-Pb data reduction is under-discussed, and the rejection of data during filtering (U-Pb ratio and/or discordance) is a seemingly universal limitation in detrital rutile work (Section 7.1). As far as we are aware, we are the first paper to systematically step through the data reduction process and explore the effects of various choices on resulting U-Pb date distributions. From this, we provide recommendations to the community for data treatment as well as opportunities based on current limitations. (2) Because of these new recommendations, we provide, as a companion to this manuscript, the Jupyter Notebooks used for data reduction and visualization. We have encountered many detrital geochronology users who want to use detrital rutile but are uncertain how to collect, reduce, and interpret these data. We hope that the provided open access code will be a helpful resource. (3) We added an overview and discussion of data filtering based on U-Pb ratio uncertainty and discordance. Following the Associate Editor's comments, we added an overview and discussion of various discordance metrics in U-Pb data from common Pb-bearing minerals. From the comments of Reviewer 1, all $^{207}$Pb correction results were re-calculated following an iterative approach, and, based on the updated Pb-corrected results and discordance calculations, we no longer recommend using a percent discordance cutoff. Instead, we recommend a filter based on U-Pb ratio uncertainty (i.e., power law filter) and the inclusion of grains of all concordance levels. (4) Several reviews indicated that more could be done with the geochemical data but did not provide specific suggestions. The geochemical data does not show any trends by stratigraphic age, sample location, or age population. We redid the PCA analysis using compositional PCA and largely found the same results: the protolith and temperature sections capture the most important components of the trace element results. We use the geochemical data, specifically Zr-in-rutile temperature and mafic-pelitic categories (Cr, Nb), to interrogate potential bias in U-Pb data rejection and uncertainty filtering (Section 7.4). In this way, we use the geochemical data to demonstrate that the data rejection does not significantly bias the U-Pb results.

Additional changes based on reviewer comments include: (1) An updated title and text that moves away from claiming 'new analytical workflows.' (2) The introduction and background are updated following reviewer comments to better address rutile stability fields, U-Pb geochronology literature, and U-Pb closure temperature. (3) The common Pb correction section includes an overview of $^{204}$Pb corrections and is updated to include an iterative approach for the $^{207}$Pb correction. All figures and results are updated with the iterative $^{207}$Pb correction approach. (4) Several reviewer comments indicated that the data reduction methods were unclear, therefore, we added additional text in the methods section (i.e., Section 4.4). Additionally, we include our data reduction workflow code as open access Jupyter notebooks. (5) The results and discussion sections have been separated. The results section now includes text on the raw U-Pb data quality with examples of accepted and rejected analyses, common Pb correction results, and a comparison of discordance metrics and uncertainty filters. The discussion section includes recommendations for U-Pb data rejection, correction, and filtering. (6) The reviewers were divided on the importance of discussing the implications of including/excluding low-U rutile grains. We significantly shortened this point. (7) We've expanded the supporting information text to reflect the reviewers' comments on U-Pb and trace element standard reproducibility. (8) All of the data repository files have been updated with the U-Pb data that was newly reduced with a weighted linear spline in iolite, iterative common Pb corrections, discordance metrics, and uncertainty filters.

In addition to the first round reviewers, the following reviewers are suggested:
Sarah George, University of Oklahoma, sarahgeorge@ou.edu
Emily Finzel, University of Iowa, emily-finzel@uiowa.edu
Will Jackson, University of Memphis, wtjckson@memphis.edu
Gary O'Sullivan, Trinity College Dublin, osullig3@tcd.ie
Andrew Kylander-Clark, University of California Santa Barbara, kylander@geol.ucsb.edu

Thank you for your guidance during the submission and review process and for consideration of the revised manuscript.

Sincerely,
Megan Mueller

**Response to the Associate Editor's comments on manuscript egusphere-2023-1293**

*1. You use the percentage of common Pb (using the Stacey-Kramers model) as a discordance filter. As mentioned on line 195 of your manuscript, this filter skews the age distribution towards younger ages. Have you tried the logratio definition of discordance? You may find that this has less of an effect on the shape of the age distributions.*

Thank you for this suggestion. In the revised manuscript we compared the Stacey-Kramers discordance model to the Aitchison (logratio) model (Section 2.3.5 and Section 5.3 and Figure 6). In addition to this study, many detrital rutile U-Pb datasets include analyses that are highly discordant with some analyses plotting close to common Pb compositions (i.e., Govin et al., 2018 Geology). The logratio distances are smallest for analyses closest to the concordia at both the upper and lower intercepts; therefore, a discordance filter based on this metric would exclude the 'middle space' analyses. The Aitchison distance metric does not seem to be applicable to datasets with high discordance analyses, but may be better suited for datasets with little discordance. On the other hand, the Stacey-Kramers distance seems to be a more representative metric of U-Pb systematics in common Pb bearing minerals, where grains closest to the common Pb composition have the highest discordance.

*2. Please remove the use of similarity, likeness and cross-correlation from your study. As a rule of thumb, it is best to avoid dissimilarity measures that were invented by geologists, not statisticians.*

We have removed this section.

*3. Please make a cleaner separation between results and interpretation.*

The updated manuscript has separated the text into results (Sections 5 and 6) and discussion (Section 7).

*4. I agree with the reviewers that PCA is potentially powerful, but hasn't been used to its full potential in your study. However, there is another issue that needs to be fixed first. According to line 372, you used OriginPro to do these calculations. I haven't used Origin before, but I understand that it is a general purpose statistics package. Therefore, I suspect that it ignores the fact that trace element concentrations are compositional data. The important difference between 'regular' PCA and 'compositional' PCA is explained by Aitchison (1983, doi:10.1093/biomet/70.1.57). There are lots of computer programs that implement compositional PCA (using logratio statistcs), including CoDaPack (in Excel) and the compositions, robCompositions and provenance packages in R. The interpretation of compositional biplots is succinctly described in Raimon Tolosana-Delgado's "CoDa in a nutshell" document: http://www.sediment.uni-goettingen.de/staff/tolosana/extra/CoDaNutshell.pdf.*

Thank you for this comment. We were not aware of this limitation in OriginPro and changed our PCA approach to use the robCompositions package in R. The results are broadly similar, where the variance in detrital rutile trace element chemistry is best explained by both protolith and metamorphic grade.

*5. Your response to reviewer 2 attributes the poor performance of the Kragero secondary standard to a pulse-analog conversion issue. Does the summary table of Figure A2 represent all the data or only the 'good' data?*

Following the comments of Reviewer 2, we now show the U-Pb reference material results in concordia diagrams in the supplement which include all of the 'good' and 'bad' data. We have added an extensive discussion of standard reproducibility in the supplemental text, following our response to reviewers.

*6. I have never used DetritalPy, but the cumulative distributions of Figure 5 look wrong. Cumulative distributions should not be smooth but should consist of discrete steps.*

The figures with cumulative distributions look smooth because they are cumulative KDE distributions. The figure captions have been updated to explain this.

*7. Lines 261-262 of your paper claims that "Detrital rutile U-Pb raw data are given in the data repository". However, when I go to the data repository, I get a spreadsheet with isotopic ratios and concentrations. This is not what I understand as raw data. It would be great if you could share your raw mass spectrometer data. Your*

*response to Reviewer 2 contains some interesting plots showing the raw mass spectrometer data to illustrate what you mean with 'spiky' data. I would be interested to see some of this detail incorporated in the paper.*

This is a good point and we have modified Figure 3 to include representative signal intensity patterns. We show examples of acceptable analyses and rejected analyses. We acknowledge that a large number of analyses were rejected and this modified figure helps illustrate and justify the situation. We are unaware of any published papers with raw data to compare our results with, so this serves as an important first step toward a more standardized community understanding of detrital rutile U-Pb data reduction workflows. Additionally, the supplemental material includes the plot from our response to Reviewer 2 within our discussion of standard reproducibility. The raw U-Pb and trace element data are now included in the data repository (https://doi.org/10.17605/OSF.IO/A4YE5).

---

## Author Response (AR2)

April 16, 2024

To the Editors and Associate Editors:

On behalf of me and my co-authors, I send you the revised manuscript titled "Navigating the complexity of detrital rutile provenance: Methodological insights from the Neotethys Orogen in Anatolia."

The manuscript investigates U-Pb and trace element data reduction, processing, and common Pb correction workflows using new detrital rutile U-Pb geochronology and trace element geochemistry results from the Late Cretaceous to Eocene Central Sakarya and Sarıcakaya Basins in Anatolia. We use our dataset to demonstrate how to navigate the complexities of natural datasets. We provide recommendations for common Pb correction, discordance calculation and data filtering that are applicable to detrital rutile and other common Pb-bearing detrital minerals. Additionally, to facilitate the standardization of data reporting approaches, we provide open access code as Jupyter Notebooks for data processing and analysis steps, including common Pb corrections, uncertainty filters, discordance calculations, and trace element analysis.

The second round of reviews of this manuscript (manuscript number egusphere-2023-1293) indicated minor revisions were needed before acceptance. One referee and the Associate Editor provided constructive comments that enabled us to clarify and strengthen the manuscript. The reviewer critiqued the number of U-Pb analyses discarded during data reduction and interpretations of the trace element data. The reviewer's comments are addressed below. The revised manuscript includes all of the changes and revisions indicated in our responses.

We thank the Associate Editor for comments, which mainly highlighted the reviewer's points, so we address them here. (1) We use the daily instrument tuning data of NIST612 glass and U-Pb precision to demonstrate that there are no analytical issues. The first-round revision manuscript was edited to emphasize that the rejection of analyses during data reduction is not a unique limitation of this study, but typical of many detrital rutile studies (see Discussion section). We reiterate that the number of discarded analyses is surprising, but is the result of a natural dataset and rather common in the literature. Our reply to Reviewer #4 provides further evidence that the number of rejected analyses is not due to analytical or data reduction errors. (2) Cumulative KDE distributions are commonly used and the difference between CKDEs and CDFs are not the focus of the manuscript. We updated the figures to CDFs. (3) To shorten the manuscript, we moved a significant portion of the discussion on discordance to the supplemental material, including the log-ratio method. Additionally, Reviewers 1, 2, and 4 commented on whether PCA is adding anything "new or interesting." In the manuscript text and reply to reviewers, we tried to demonstrate the many ways in which we have looked at the trace element data. In the end, the most insights are gained from Cr, Nb, and Zr values. In light of this assessment and the recommendation to shorten the manuscript, we moved the PCA text and figure to the supplemental material. (4) Regarding manuscript length, we shortened the manuscript by moving several sections of text and 3 figures to the supplemental material.

Thank you again for consideration of the revised manuscript.

Sincerely,
Megan Mueller

**Response to the Reviewer #4's comments on manuscript egusphere-2023-1293**

*This paper from Mueller et al. details complexities around the dating of detrital rutile in rocks from Anatolia. It provides some information that are commonly not provided in geochronological datasets, to provoke discussion about how we can best treat these kinds of data and materials (i.e. detrital Pbc–bearing minerals that contain provenance-diagnostic trace elements and that can be dated by the U–Pb method). As this paper has already been reviewed, I have few comments about the abstract, introduction and "synopsis" (section 2), which have been altered quite heavily from the initial submitted manuscript and which appear to me to be a useful and concise discussion of relevant topics. The Geological context (section 3) is also perfectly followable in its current format, and the methodology (section 4) can also be easily parsed and appears robust to me. And, overall, the paper provides several interesting diagrams and discussion on topics related to dating of Pbc bearing phases.*

*Section 5.1 has my first comment. Here you state that 665 of your 1,277 analyses have been rejected (line 395). I am familiar with detrital rutile dating, and this seems to be uncommonly high. In general, however, I think that anyone would agree that methodologies that screen data must be applied lightly and applied only to improve the accuracy of the resulting dataset, and also that any screen that is too onerous can run counter to that purpose. As this is the most significant screen in your data processing workflow, it thus needs to be rigorously and extensively justified. Figure 3 attempts to do this, but it is a blunt tool for this task. Qualitative descriptions of unsuitable raw ICPMS signals are given, but reproducible thresholds to include or not to include a grain aren't provided. You identify three problems that are cause for you to discard grains. These are "spikiness", inclusions, and low-signals. Firstly, which of these three reasons is most significant, and in what proportion do these three categories occur? Secondly, a 'spiky' raw ICPMS signal can be a sign of poor gas flow in the analytical set-up, cones that need to changed, or other build-up in the system's cells and tubing. For completeness, you should consider and discuss this. Thirdly, and most seriously – screening grains on the basis of low raw counts on U and Pb is philosophically just as incorrect as screening grains on the basis of low U in ppm, similar to the authors whose methodology you disagree with (i.e. Okay et al., 2011). The only difference is that, instead of discarding grains on the processed signal, you now reject grains on the basis of the raw signal. On those grounds, I fundamentally disagree with this approach. Why not use the Chew et al. power-law filter for all grains, regardless of a screen for raw background corrected counts on 238, 206, 207 etc.? Such an approach would at least treat all grains in the same way.*

       We reiterate that the number of discarded analyses is surprising, but is the result of a natural dataset and not analytical or data reduction error.

       Rejecting/filtering data is common practice, whether due to high uncertainty or discordance, etc. However, using a filter based on element abundance (i.e., 4-5 ppm U threshold) is only valid for a certain abundance sensitivity (cps/ppm), which depends on instrument (laser and ICP-MS type) and instrument settings (see also Section 7.2 text). A threshold/filter based on the raw data (basically on cps) on the other hand is directly linked to counting statistics, which is a fundamental statistical limitation and not instrument or setting specific. This is not at all the same as the reviewer claims. Basing it on the U ppm makes the most sense for our dataset of mostly Phanerozoic rutiles, because we are mostly interested in the grains with high U/Pb, not the ones with high Pb counts and low U/Pb. For datasets with largely Precambrian rutile another approach may be needed.

The U (ppm) versus 206Pb/238U uncertainty plot (Figure R1) shows that the main issue is the very low U and therefore Pb concentrations, followed by inclusions. We demonstrate that the very, very low concentration grains have corresponding low counts therefore have high uncertainty. We reiterate that the rutile material we are working with has significantly lower U concentrations than many other studies. There is a good reason to reject data with high uncertainty, because they do not allow geologically significant dates to be calculated. At some point a threshold has to be set. We include grains down to $10^{-3}$ ppm U, which is three orders of magnitude below the 4-5 ppm threshold in the literature, therefore we argue that excluding low U and low Pb signals is not the same as screening low U grains at 4-5 ppm. Further, the U filter used here is based on the CPS and therefore the limits posed by pure counting statistics, whereas a "ppm filter" is instrument and setting dependent.

The "spiky" signals are mainly a result of very low count rates due to the exceptionally low U concentrations encountered in many of the analyzed rutile grains (Fig. R1). We provide a few comments on instrument set-up by addressing instrument tuning and the precision of results. A summary of daily tuning results is given in Table R1. QA/QC for tuning is aimed at stable signal, high count rates and low oxide production monitored by measuring NIST 612 glass. U238 for 50 micron spot size and 3.3 J/cm2 fluence is at 2.4- 4 million cps. Oxide rate is below 0.2% 254UO as percentage of 238U. The cps yield per ppm U was 64,000-107,000 cps/ppm for NIST612 for 50 micron spots, and 63,000-75,000 cps/ppm for 50 micron spots on rutile with 3.0 J/cm2 fluence (Table R1; now added to method table S2 in supplement). The NIST 612 cps/ppm values are in the high range of what is typically achieved on our Element2 ICP-MS since 2009 (This is monitored every day). We do not have data from other laboratories since this is data that is rarely published, but we contend that this is more than adequate in comparison to any other laboratory with a single collector ICP-MS and similar detector setup. We are not aware that other labs with a similar setup (Element2 without xcones setup) can achieve significantly higher cps yield unless they are using much higher rep rates or laser energy, which produces deeper pits and can have a detrimental effect on downhole fractionation. It is noted that some publications report much higher fluence values, but from discussion with other colleagues and laser company engineers, many of these are not properly calibrated and are overestimates.

Further, we compare the precision of our U-Pb results (single collector HR-ICP-MS) to those of two published studies using an HR-ICP-MS (Odlum et al., 2024), MC-ICP-MS (Bracciali et al., 2015) and Q-ICP-MS (Jenkins et al., 2023) (Figure R1). We achieve lower uncertainties on rutile with U concentrations in the parts per million range (> 1 ppm U) compared with the unknowns analyzed on a multi-collector. Compared with the reference materials analyzed on a Q-ICP-MS, we achieve similar precision on our unknowns in the parts per million U range (> 1 ppm U). Our rutile range extends to 100x or less U than the rutile analyzed by quadrupole and multi-collector instruments, and the rutile we had high uncertainty on is in the lower U range (less than 1 ppm U).

Table R1: Summary of daily instrument tuning results.

| NIST612 ppm U: | 37.38 NIST 612 | | | | | Th/U | R10 ppm U | 44.1 | | | GSD | | |
|---|---|---|---|---|---|---|---|---|---|---|---|---|---|
| | cps/ppm | 238U | 206Pb | spot (µm circle) | J/cm2 | | cps/ppm | 238U | spot (µm circle) | J/cm2 | Ti49 | spot (µm circle) | J/cm2 |
| 7/12/2021 | 107009 | 4.00E+06 | 750,000 | 50 | 3.3 | 0.7 | 40816 | 1.80E+06 | 35 | 3.0 | | | |
| 7/13/2021 | 101659 | 3.80E+06 | 800,000 | 50 | 3.3 | 0.78 | 74830 | 3.30E+06 | 50 | 3.0 | | | |
| 7/13/21 trace | 101659 | 3.80E+06 | 730,000 | 50 | 3.3 | 0.7 | | | 35 | 3.0 | | | |
| 7/14/2021 | 66881 | 2.50E+06 | 500,000 | 50 | 3.3 | 0.75 | 68027 | 3.00E+06 | 50 | 3.0 | | | |
| 7/14/2021 trace | 66881 | 2.50E+06 | 500,000 | 50 | 3.3 | 0.75 | 68027 | 3.00E+06 | 50 | | 1.80E+06 | 25 | 3 |
| 7/15/2021 | 66881 | 2.50E+06 | 500,000 | 50 | 3.3 | 0.7 | 63492 | 2.80E+06 | 50 | 3.0 | | | |
| 7/16/2021 | 64205 | 2.40E+06 | 480,000 | 50 | 3.3 | 0.75 | 68027 | 3.00E+06 | 50 | 3.0 | 1.70E+06 | 25 | 3 |
| 7/19/2021 trace | 66881 | 2.50E+06 | 520,000 | 50 | 3.3 | 0.8 | | | | | 1.80E+06 | 25 | 3 |

[Figure]

Figure R1: Comparison of 206Pb/238U uncertainty (2s %) versus uranium concentration from this study (detrital rutile unknowns; analyzed on single collector HR-ICP-MS), Odlum et al. (2024) (detrital rutile unknowns; single collector HR-ICP-MS) Bracciali et al. (2015) (detrital rutile unknowns and rutile secondary standard; MC-ICP-MS), and Jenkins et al. (2023) (rutile reference materials; Q-ICP-MS).

*My second comment relates to interpretations of the trace elements.*

- *Firstly, point 4) in the abstract could be due to an artefact in your detrital dataset. In particular, figure 10 does not convince me that you have sufficient data to determine whether metapelitic or metamafic rutile in general contains proportionally more U. It could be that the metamafic rutile have lower average U due to random chance due to low numbers of analysed grains. Additionally, your finding only holds within the confines of your dataset, which is not globally representative.*

We show that "mafic classified grains are dominantly low U (95%, n=106/112 below 4 ppm). The majority of rutile with U contents above 4 ppm are classified as pelitic (85%, n=34/40)" (Section 6.2). Figure 10 does not show all of the rutile grains with measured U ppm, but rather shows the smaller subset of grains with both U-Pb and trace element data, as noted in the figure caption. In any case, we agree that our dataset is not a universal representation of detrital rutile, however, it is documented that "uranium concentration in rutile varies among metamorphic protoliths: for example, rutile from mafic

eclogites tend to have, 134 on average, 75% less U than those from metapelites (i.e., 5 ppm vs. 21 ppm; Meinhold, 2010)" (Section 2.2). Our dataset affirms this trend.

- *Secondly, exploration of the trace element data is underdeveloped. PCA is an extremely useful tool for exploratory geochemical data analysis, but it inherently results in loss of information, as all geochemical variation is condensed into a 2-dimensional space – it is possible that scatterplots etc. may reveal useful information not shown on PCA diagram. In section 6.3, you make the interpretation that the trace element data derives from protolith (Cr, Nb etc.) and temperature (Zr, Hf) factors. However, the vectors on figure 12 demonstrate that PC1 is dominated by Tungsten (W). Why is this? What is the significance of that finding? And why is it not discussed? Additionally, why not colour the points by their metapelitic/metamafic categorisation, and/or Zr-in-rutile T? This would indicate whether PC2 really is discriminating on the basis of protolith, or PC1 on the basis of T.*

The main trace elements discussed in the rutile literature are Cr, Nb and Zr (citations), with additional attention given to the combination of Cr, Nb, Zr, V and Fe for discriminating $TiO_2$ polymorphs (i.e., rutile, anatase, brookite see our supplemental Figure S2) (Triebold et al., 2012) and to Nb and Ta as tracers of subduction zone fluids and continental crust formation (Figure R2; Rudnick et al., 2000; Xiao et al., 2006). We include the commonly used scatterplots of these elements in the main text (e.g., Figure 9). Additionally, we display the results on Tera-Wasserburg diagrams to show the distribution of scatter plot discrimination fields by age (e.g., Figures 9, 11), which is not commonly done.

There is little literature on W in rutile and how it can be used in detrital rutile datasets. The elements W, Sb and Sn can be used in mineral exploration (Plavsa et al., 2018; Agangi et al., 2019). We show our trace element results with rutile from orogenic gold deposits, which were used to roughly define ore (Au mineralized), metamorphic and granitoid fields (Agangi et al., 2019). Most of our data plot within the metamorphic field and there is no correlation by rutile age (Figure R3) or by protolith.

Reviewers 1, 2, and 4 have commented on whether the PCA is adding anything "new or interesting." In the manuscript text and reply to reviewers, we have tried to demonstrate the many ways we have looked at the trace element data. In the end, the most insights are gained from Cr, Nb, and Zr. In light of this and the Associate Editor's recommendation to shorten the manuscript, we removed the PCA text and figure from the revised manuscript.

[Figure]

Figure R2: Nb/Ta versus Nb diagram after Xiao et al. (2006). The rutile grains are colored by their mafic-peitic classification (Cr vs Nb; see main text). The mafic and pelitic grains group together, which is expected as protolith is classified by Nb contents. The grains are scattered between continental crust, chondritic or eclogite values and there is no clustering by grain date.

[Figure]

Figure R3: Trace element data plots of Sb, Sn, W and V used to delimit ore-related (Au mineralized) rutile (after Agangi et al., 2019). Rutile from this study are shown with the dataset of orogenic gold deposit rutile from Agangi et al. (2019). White circles are rutile grains without U-Pb dates.

● *Thirdly, plots of PCA loadings are discussed in section 6.3, but not such plots are shown.*

We moved the PCA from the main text to supplement, and briefly address this comment. The loadings are shown in the PCA figure as lines with arrows. PCA scores represent the transformed data points in the new coordinate system defined by the principal components, where each data point in the original dataset is projected onto the principal component axes, resulting in a set of scores that describe the position of the data points in the new coordinate system. Loadings represent the correlations between the original variables and the principal components, thus describing the relationships between the original variables and the principal components. The loadings vectors indicate the direction and magnitude of the contribution of each original variable to the principal components.

● *Fourthly, and lastly, consideration is not given to the fact that some rutile derive from igneous rocks. It is possible that some rutile labelled as metamafic or metapelitic using the scheme of Triebold et al. (2012) may derive from a plutonic rocks (e.g. Huang et al., 2024 – igneous rutile dated from an Archean pluton; Pe-Piper et al., 2019 – igneous rutile dated from a syenite; Janousek and Gerdes, 2003 – igneous rutile dated from a granitoid pluton). Indeed, on line 76 it is stated that rutile is a common accessory mineral in metamorphic \*and igneous\* rocks. How might this affect interpretations on the basis of a metamorphic/igneous division, especially WRT high-T concordant "metapelitic" grains on figure 11. Might these be igneous?*

Distinguishing igneous versus metamorphic rutile is not straightforward in detrital samples, and is reviewed in Pereira and Storey (2023). The composition of heavy mineral assemblage (e.g., counting the total number of rutile and zircon grains in a sample) can be used as an index for igneous or metamorphic sources (Morton and Hallsworth, 1994). Another proposed method is using the ore discrimination diagrams discussed above (Figure R3). In those diagrams, the majority of grains plot in the metamorphic field rather than granitoid. Additionally, we suggested that igneous rutile could potentially be identified if their date overlaps with detrital zircon populations. For example, "we interpret the 90 Ma rutile population as either igneous or metamorphic rutile derived from Late Cretaceous magmatism and associated contact metamorphism on the Pontides" (Section 8). This is because the "rutile grains that (poorly) define the ca. 90 Ma population [...] include some of the highest Zr-in-rutile temperatures" and because the "zircon record has abundant Late Cretaceous and Eocene populations [...] associated with magmatic flare-ups" (Section 8). We suspect that the 90 Ma rutiles are either igneous or formed during syn-magmatic, high-T metamorphism.

*My third and last comment relates to the discussion. From line 599 several examples are given of papers where a significant proportion of rutile ages were rejected from published studies (Caracciolo et al., 2021; Govin et al., 2021; Shannan et al., 2020). Firstly, it is not clear from this section whether these rates of rejection are typical of detrital rutile studies, or whether perhaps they are instead related to the specific source regions of these rutile. And secondly, it is noted that the study of Shannan et al. (2020) uses a discordance filter, which is a method that is discounted by the present study and thus not particularly useful for comparative purposes. Consequently, I don't find this section convincing as an argument to support the interpretation from line 607: "Together these studies illustrate Together these*

*studies illustrate that there is a formidable methodological hurdle in trying to scale up detrital rutile U-Pb to large-n provenance applications".*

We are uncertain how representative these 4 studies are for the rates of U-Pb data rejection in detrital rutile. It is rarely reported and these 3 published studies plus this manuscript are what we found available. We are unaware of large-n detrital rutile datasets (> 300 analyses / sample), which is likely limited by rutile fertility and data rejection. As far as we are aware, this is the first manuscript to discuss the rejection of data, criteria for rejection, and potential bias. These 4 datasets evidence that data rejection does occur. It is hard to speculate how prevalent it is without more data reporting on data inclusion/rejection. For this reason, "more rigorous data reporting and standardizing metrics used for evaluating 'acceptable' U-Pb analyses. We recommend that the criteria for data rejection be explicitly discussed in all detrital rutile studies."

*Figure comments*
*Figures 5, 8 – a cumulative age distribution plot would be more useful than a cumulative KDE plot, which is simply a repetition of the data below it in a more awkward form.*
We changed the figures, thank you.

*Figure 11 – near-concordant grains are often very high T grains. Is it possible that these are primary igneous rutile?*
Yes, it is possible. Please see above discussion.

*Figure 12 – colour by protolith according to the scheme of Triebold et al., 2012.*
We removed this figure. See above discussion.

References Cited

Agangi, A., Reddy, S. M., Plavsa, D., Fougerouse, D., Clark, C., Roberts, M., and Johnson, T. E.: Antimony in rutile as a pathfinder for orogenic gold deposits, Ore Geol. Rev., 106, 1–11, https://doi.org/10.1016/j.oregeorev.2019.01.018, 2019.

Bracciali, L., Najman, Y., Parrish, R. R., Akhter, S. H., and Millar, I.: The Brahmaputra tale of tectonics and erosion: Early Miocene river capture in the Eastern Himalaya, Earth Planet. Sci. Lett., 415, 25–37, https://doi.org/10.1016/j.epsl.2015.01.022, 2015.

Jenkins, K., Goemann, K., Belousov, I., Morissette, M., and Danyushevsky, L.: Investigation of the Ablation Behaviour of Andradite-Grossular Garnets and Rutile with Implications for U-Pb Geochronology, Geostand. Geoanalytical Res., 47, 267–295, https://doi.org/10.1111/ggr.12478, 2023.

Odlum, M. L., Capaldi, T. N., Thomson, K. D., and Stockli, D. F.: Tracking cycles of Phanerozoic opening and closing of ocean basins using detrital rutile and zircon geochronology and geochemistry, Geology, https://doi.org/10.1130/G51826.1, 2024.

Pereira, I. and Storey, C. D.: Detrital rutile: Records of the deep crust, ores and fluids, Lithos, 107010, https://doi.org/10.1016/j.lithos.2022.107010, 2023.

Plavsa, D., Reddy, S. M., Agangi, A., Clark, C., Kylander-Clark, A., and Tiddy, C. J.: Microstructural, trace element and geochronological characterization of TiO2 polymorphs and implications for mineral exploration, Chem. Geol., 476, 130–149, https://doi.org/10.1016/j.chemgeo.2017.11.011, 2018.

Rudnick, R., Barth, M., Horn, I., and McDonough, W. F.: Rutile-Bearing Refractory Eclogites: Missing Link Between Continents and Depleted Mantle, Science, 287, 278–281, https://doi.org/10.1126/science.287.5451.278, 2000.

Triebold, S., von Eynatten, H., and Zack, T.: A recipe for the use of rutile in sedimentary provenance analysis, Sediment. Geol., 282, 268–275, https://doi.org/10.1016/j.sedgeo.2012.09.008, 2012.

Xiao, Y., Sun, W., Hoefs, J., Simon, K., Zhang, Z., Li, S., and Hofmann, A. W.: Making continental crust through slab melting: Constraints from niobium–tantalum fractionation in UHP metamorphic rutile, Geochim. Cosmochim. Acta, 70, 4770–4782, https://doi.org/10.1016/j.gca.2006.07.010, 2006.